# BMP4 initiates and patterns ventral-caudal structures in zebrafish and human pluripotent stem cell aggregates

Yan-Yi Xing [1,5], Ying Huang [2,5], Tao Cheng [1,5✉], Yi-Meng Tian [2], Yang Dong [1], Zi-Xin Jin [2], Hai-Rong Pu [2], Tao Luo [3], Xiang Liu [2], Hao-Nan Shen [2], Jing Mo [4], Jun Ma [2], Jun-Feng Ji [3] & Peng-Fei Xu [1,2✉]

## Abstract

**The formation of body axes is a key developmental milestone in vertebrate embryos and is guided by specialized groups of cells known as organizers. The molecular nature of organizers has been extensively investigated across the vertebrate kingdom; however, the minimal conditions and factors sufficient to guide embryogenesis and organogenesis—particularly in humans—remain incompletely understood. Here, we show that BMP4 alone, when administered at an appropriate dosage, is sufficient to induce the formation of an organizer for ventral–caudal-like structure (VCLS) formation. This organizer directs endoderm-deficient ventral–caudal cell fate specification and morphogenesis in zebrafish embryos. In 3D human pluripotent stem cell (hPSC) aggregates, BMP4 can induce an elongated embryonic structure that is characterized by ventral–caudal cell fates. Importantly, hPSCs instructed by BMP4 are sufficient to induce a secondary posterior axis when grafted into the animal pole of the zebrafish embryo. Our study thus uncovers BMP4 as the inducer for the formation of a ventral–caudal organizer in the vertebrate embryo.**

**Keywords** BMP4; Ventral-caudal Organizer; Morphogenesis; Zebrafish Explant; Human Pluripotent Stem Cells
**Subject Categories** Development; Stem Cells & Regenerative Medicine

## Introduction

During early vertebrate embryogenesis, body axis elongation occurs in two principal stages. Initially, lineage specification during gastrulation results in the formation of three germ layers, where processes like internalization, convergence, and extension rearrange and elongate these layers. This leads to the development of the head and trunk regions, with mesodermal precursors contributing to the axial, paraxial, and lateral plate mesoderm (LPM) (Wilson et al, 2009). Subsequently, a structure known as the tailbud emerges at the posterior embryo end, harboring progenitors vital for cell lineage specification and tail elongation, which forms the caudal body axis. In zebrafish, tailbud formation occurs as the ventral and dorsal parts of the blastoderm margin slip over the yolk and fuse at the vegetal pole at the end of gastrulation (Beck, 2015; Gonzalez et al, 2000; Kanki and Ho, 1997; Kimmel et al, 1995; Kimmel et al, 1990; Myers et al, 2002). Although loss-of-function studies have identified numerous genes and signaling pathways essential for tailbud development (Agathon et al, 2003; Aires et al, 2019; Esterberg et al, 2008; Goto et al, 2017; Gouti et al, 2017; Kinney et al, 2020; Koch et al, 2017; Ye and Kimelman, 2020), our understanding of the minimal factors required to direct caudal tissue formation remains limited, particularly in humans.

A century ago, Mangold and Spemann discovered that the dorsal blastopore lip of an amphibian gastrula can induce a secondary axis (mainly the head and trunk region)(Martinez Arias and Steventon, 2018; Spemann and Mangold, 1924). After that, the organizer concept has been well recognized in different developmental processes and stages, across different species, and the molecular natures of those organizers have also been well characterized (Agathon et al, 2003; Anderson and Stern, 2016; Fauny et al, 2009; Martinez Arias and Steventon, 2018). However, the minimal conditions that can induce an organizer capable of inducing and patterning the caudal tissues of the vertebrate embryo have not been fully addressed. Stem cell-based synthetic embryology has created new research models and provided significant insights into developmental and regenerative biology (Schlissel and Li, 2020). By applying essential morphogens globally or locally, key aspects of early development have recently been recapitulated in vitro with pluripotent stem cells. For instance, activation of the WNT signaling pathway by its agonist-treated homogenously can result in the formation of gastruloids (Beccari et al, 2018; Hamazaki

[1]Women's Hospital, and Institute of Genetics, Zhejiang University School of Medicine, 310006 Hangzhou, Zhejiang, China. [2]Center for Genetic Medicine, the Fourth Affiliated Hospital of School of Medicine, and International School of Medicine, Zhejiang University, 322000 Yiwu, Zhejiang, China. [3]Center of Stem Cell and Regenerative Medicine, and Bone Marrow Transplantation Center of the First Affiliated Hospital, Zhejiang University School of Medicine, 310058 Hangzhou, Zhejiang, China. [4]Department of Immunology, School of Basic Medicine, Guizhou Medical University, 561113 Guiyang, Guizhou, China. [5]These authors contributed equally: Yan-Yi Xing, Ying Huang, Tao Cheng. ✉E-mail: chengtao2ldo@zju.edu.cn; pengfei_xu@zju.edu.cn

et al, 2024; Turner et al, 2017; Umemura et al, 2022; van den Brink et al, 2020), trunk-like structures (Veenvliet et al, 2020), or somitoids (Qin et al, 2022), depending on the culture conditions. Similarly, the development of A-P patterned head-like structures can be induced by a single gradient of Nodal (Cheng et al, 2023). Treatment with BMP4 in micropatterned human stem cells can recapitulate early embryonic spatial patterning (Warmflash et al, 2014), and BMP4-induced morphogen signaling center could result in the development of a mammalian embryo model (Xu et al, 2021). Moreover, human stem cell colonies treated with Wnt and Activin can function as "organizer" (Martyn et al, 2018). These models have effectively mimicked germ layer specification, body patterning, and somitogenesis.

BMP4, a member of the TGF-β superfamily, has been shown to play critical roles in body plan establishment and tailbud development across various organisms such as mice (Sharma et al, 2017), frog (Beck et al, 2001), chick (Reshef et al, 1998), and zebrafish (Pyati et al, 2005). Mutation of genes that repress BMP signaling, such as *swirl/bmp2b* (Kishimoto et al, 1997), *snailhouse/bmp7a* (Dick et al, 2000), *somitabun/smad5* (Hild et al, 1999), and *lost-a-fin/alk8* (Mintzer et al, 2001), leads to the defect in tail development and varying degrees of posterior tissue truncation in zebrafish. It was also shown that elevated BMP signaling promotes epibolic cell migration into the tailbud, enhancing tail formation while suppressing the development of head and trunk structures (Myers et al, 2002). Although the majority of *Bmp4*-deficient mice fail to complete gastrulation, a small number of embryos that develop to later stages exhibit defects in posterior structures development (Winnier et al, 1995). In this study, we demonstrate that moderate BMP4 signaling can induce the formation of a secondary axis in zebrafish that contains caudal structures yet lacks axial mesoderm and endoderm. Similar results can be obtained from the BMP4-induced zebrafish explant and 3D aggregates of human pluripotent stem cells. By using single-cell RNAseq and live imaging, we systematically analyzed the dynamics of cell fate specification and cell movements of key cell types during those processes. Additionally, the organizing potential of BMP4-treated hPSCs was further validated through xenograft experiments.

# Results

## Bmp4 induces a secondary axis with caudal-like tissues in zebrafish embryos

We first employed zebrafish animal pole blastomeres as a model to explore the induction capacity of Bmp4. The pluripotent, naïve, and homogeneous nature of the animal pole cells in zebrafish embryos makes them an ideal system for studying germ layer induction and morphogenesis (Ho and Kimmel, 1993).

By injecting 10 pg of zebrafish *bmp4* mRNA into one animal-pole blastomere at the 128-cell stage, we established a BMP signaling gradient (Fig. EV1A). No obvious morphological changes were observed in injected embryos from the blastula stage until the beginning of gastrulation (6 hpf, hours post fertilization). Interestingly, a protrusion progressively bulges at the injection site and becomes evident at mid-gastrulation (8 hpf, Fig. 1A). At the beginning of the segmentation stage (12 hpf), a distinct secondary axis became apparent (Fig. 1A). This axis extends from the head

region of the primary axis to a separate protrusive end that resembles the tailbud. By 24 hpf, the *bmp4*-injected embryo developed a secondary axis that diverged from the main axis at the head region, featuring multiple somites but lacking a visible notochord, indicating the absence of axial mesoderm (Figs. 1A and EV2B).

Molecular analysis of the induction process revealed that Bmp4 activates key morphogenetic signals pivotal for early development, including Nodal, Wnt, and FGF, as evidenced by the expression of *ndr2*, *wnt8a*, and *fgf8a* (Figs. 1D and EV1G). At the same time, the naive animal pole cells are transformed into ventral mesoderm in response to *bmp4*, as shown by the expression of *eve1* (Fig. EV1G). However, we failed to detect the expression of endoderm marker *sox17* near the *bmp4*-injected region (Fig. EV1G).

A secondary tailbud-like structure induced by *bmp4* is evident at 12 hpf, marked by the expression of *tbxta* (Fig. EV1I). At 24 hpf, markers of ventral-lateral mesoderm derivatives such as pronephric duct (*cldn3d*) and somites (*tnnt2c, myod1*) can be detected in the secondary axis (Fig. EV1J). However, specification of axial mesoderm is not observed, as indicated by the absence of *shha* expression (Fig. EV1J). In addition, a neural structure—marked by *sox19a* expression—was observed to extend to the tip of the secondary tail, and it lacked ventral neural identity, as indicated by the absence of *olig2* expression (Fig. EV1J). Notably, no specification of a head region is detected in the secondary axis, as indicated by the lack of *egr2b* expression (Fig. EV1J).

## In vitro induction of a ventral–caudal-like structure by moderate Bmp4 signaling

To further explore the induction potential of Bmp4 in vitro, we explanted the animal pole cells from embryos injected with *bmp4* mRNA at 512-cell stage (hereafter referred to as 'Bmp4 explant'). We then investigated the morphological and cellular events within these explants.

Upon excision, the explant rapidly became a spherical shape and established a BMP signaling gradient (Figs. 1B and EV1B). Beginning from the onset of segmentation, the explants exposed to a moderate amount (8–10 pg of mRNA) of *bmp4* exhibited a remarkable elongation (about 50% of injected explants, Figs. 1C and EV2C,G; Movie EV1), whereas both insufficient and excessive levels of *bmp4* failed to stimulate this elongation (Figs. 1A,C and EV2A,C–M). In elongated explants, BMP signaling—measured by the expression of its direct target *id1* (Katagiri et al, 2002)—forms a gradient that is high in the *tbxta*-expressing tailbud-like region and low in the anterior pre-somatic mesoderm-like region expressing *tbx6* (Fig. EV1C), recapitulating in vivo patterns (Row and Kimelman, 2009). Interestingly, the elongated explants exhibited sequential somite formation (Fig. 1G; Movie EV11).

Consistent with the in vivo results, the expression of *wnt8a*, *ndr2*, and *eve1* in the Bmp4 explants was also activated during gastrulation (Fig. 1D). At the segmentation stages, we found that *tbxta* was expressed at one end of Bmp4 explant, which we designated as the posterior (Fig. 1E,F). In addition, a neural domain that expresses *sox2* was found to partially overlap with a tailbud-like mesodermal domain expressing *tbxta*. This overlapping region exhibits the molecular features of neuromesodermal progenitors (NMP, *tbxta*[+], and *sox2*[+], Fig. 1E,F), which we defined as NMP-like cells.

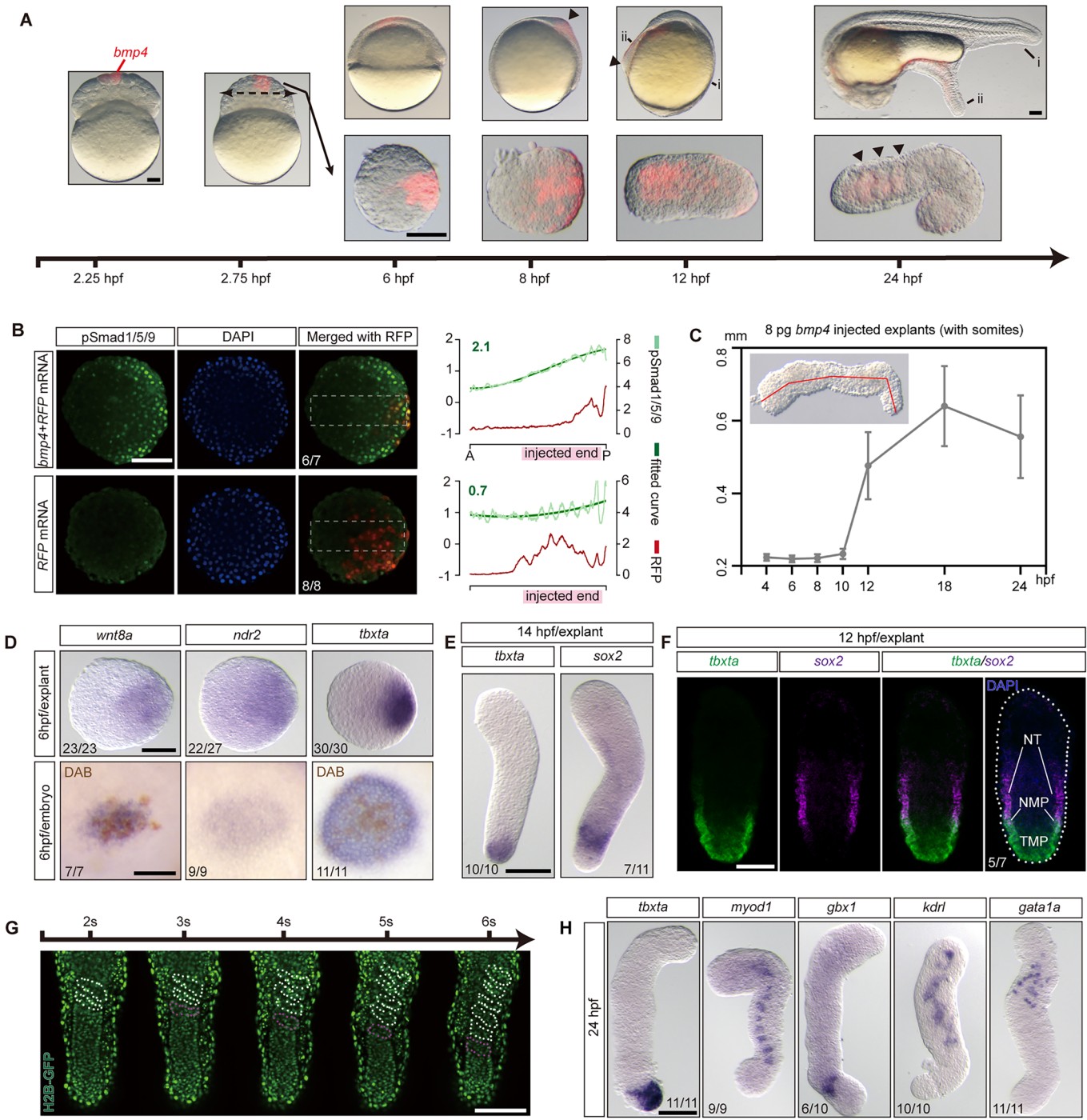

By 24 hpf, we detected the specification of the ventral-lateral mesodermal derivatives such as erythrocyte (*gata1a*) and vascular endothelium (*kdrl*). And the Bmp4 explants eventually formed 8–10 somites, as indicated by *myod1/fn1b* expression (Figs. 1H and EV1K). However, we did not detect the formation of notochord or head region (*tbxta*, *shha*, *egr2b*, Figs. 1H and EV1K). Although a neural domain expressing *gbx1* was found in the explant (Fig. 1H), it was not well organized compared to the in vivo neural tube. Taken together, these results indicate that Bmp4 explant closely resemble the caudal region of the zebrafish embryo but lack axial

mesoderm and endoderm. Thus, we defined this structure as ventral–caudal-like structure (VCLS).

We then explored why *bmp4* alone is sufficient to induce a VCLS in zebrafish. Previous studies have indicated that *bmp2b*, in combination with Nodal ligand, can instruct the development of tail tissue, which is mediated by a tail organizer specified through triple stimulation of Nodal, Wnt, and BMP (Fauny et al, 2009; Xu et al, 2014). However, explants constructed by injection of various dosages of *bmp2b* alone never underwent extension, and were absent of Nodal, Wnt signaling, and mesoderm specification (*tbxta*)

**Figure 1. Testing the induction capability of Bmp4 in zebrafish embryo and explant.**

(A) Representative images of *bmp4*-injected zebrafish embryos (upper) and explants (lower, with injection site towards the right) from 2.25 to 24 hpf. Red fluorescence marks the descendants of the *bmp4*-injected blastomere. Arrowheads at 8 hpf and 10 hpf embryo point to the protrusions induced by Bmp4 in embryos, and at 24 hpf in explants to the segmented somites. Labels indicate the primary (i) and secondary axis (ii). (B) Immunofluorescence imaging of Phospho-Smad1/5/9 in *bmp4* + *RFP* injected explant (upper) and *RFP* injected explant (lower) at 4 hpf, with the injection site towards the right. Signal measurements along the dotted rectangles are shown on the right. Dark Green curves represent the nonlinear fit of actual data curves (light green), with Dark Green numbers indicating the mean slopes. Red curves depict the distributions of RFP signaling intensity along the dotted rectangles. (C) Quantification of the elongation of Bmp4 explants from 4 to 24 hpf. The Midlines along anterior–posterior axis of Bmp4 explants are measured (red line), as displayed in the upper left. Data are presented as mean ± SD. (D) *bmp4*-injected explants (upper, posterior towards the right) and embryos (lower, animal pole view) at 6 hpf stained by whole mount in situ hybridization (WISH) with the indicated probes. DAB staining of GFP (brown spots) indicates the clones of *bmp4&GFP* mRNA injected blastomere at 2.25 hpf. (E) Bmp4 explants at 14 hpf stained by WISH with indicated probes. (F) Whole mount in situ hybridization chain reaction (HCR) staining of Bmp4 explants at 12 hpf for *tbxta* and *sox2*, with co-staining of DAPI. NT neural tube, MP neuromesodermal progenitor, TMP tailbud mesodermal progenitor. (G) Time-lapse imaging of Bmp4 explants displaying the sequential formation of somites, as per Movie EV11. White dotted lines indicate existing somites, and purple dotted lines indicate newly formed somites. Nuclei were labeled by H2B-GFP (green fluorescence). (H) Bmp4 explants at 24 hpf stained by WISH with the indicated probes. For all explants displayed in this article, anterior (opposite to the injection site) is toward the top unless otherwise stated. Each experiment was performed for at least three independent replicates (technical replicates). Scale bars: 100 μm. Source data are available online for this figure.

(Figs. EV2A,N–P and EV1F). In addition, although uninjected explants express BMP ligands (Xu et al, 2021), they fail to induce Nodal, Wnt signaling, or mesoderm formation, even upon the addition of low doses of *bmp4* (Fig. EV1F). BMP4 has been shown to act upstream of Nodal during the blastula stage(Winnier et al, 1995) and maintain Wnt signaling levels in the tailbud in mice (Sharma et al, 2017), and induce both Nodal and Wnt in 2D micropatterned gastruloids derived from human PSCs containing all three germ layers (Martyn et al, 2018). We found that zebrafish *bmp4* alone is sufficient to induce Nodal, Wnt signaling, thereby generating a tailbud-like region both in vivo and in vitro (Figs. 1D and EV1D,G). Inhibition of Nodal signaling prevented *bmp4* from inducing the tailbud-like region and caudal mesoderm, and also blocked explant elongation. Inhibition of Wnt signaling attenuated the mesodermal induction capacity of *bmp4* and abolished the explant elongation. Inhibition of FGF signaling did not significantly impact *bmp4*-induced caudal mesoderm but still impaired explant elongation, possibly due to disruption of the segmentation clock during somitogenesis (Fig. EV1E). Notably, high dosage of zebrafish *bmp4* could induce the endoderm specification (Fig. EV1H). Taken together, these results indicate that zebrafish *bmp4* alone—at appropriate dosages and without exogenous Nodal—is sufficient to simultaneously activate Nodal, Wnt, and BMP signaling, leading to the formation of a tailbud-like region, a capability not shared by *bmp2b* alone. And cells in this tailbud-like region were further organized and developed into a VCLS.

## Emergence of the anterior–posterior axis of Bmp4 explant

To systematically analyze the cell fate specification of Bmp4 explant, we performed single-cell sequencing analysis. We collected 35,830 single-cell transcriptomes from 6 developmental stages of Bmp4 explants over 24 h (Figs. 2A and EV3A; Datasets EV1 and EV2), and 28 cell clusters were identified and annotated based on the marker gene expression. These clusters could be categorized into somitogenesis-related clusters, lateral plate mesoderm and its derivatives, neural ectoderm and its derivatives, epidermal ectoderm and its derivatives, and other cell clusters (Fig. 2A).

It is noteworthy that lineages such as paraxial mesoderm-related progenitors, including NMP-like, tailbud mesoderm progenitor (TMP (Martin and Kimelman, 2010)), and pre-somitic mesoderm (PSM) were specified during the gastrulation stage, and gradually diminished during the somitogenesis stage. In contrast, tissues such as somites, pronephric duct (PD), vasculature, and erythrocytes emerged and expanded during the organogenesis stage (Fig. 2B), which is in accordance with the in vivo situation. Besides, these cell fates cannot be found in uninjected control explants, which are composed of anterior neural and epidermal cell identities at 10 hpf (Fig. EV4I–K). Bmp4 explant showed transcriptional similarity to wild-type embryos at each of the six developmental time points (Fig. 2C; Dataset EV1), suggesting a comparable developmental trajectory up to 24 hpf. The developmental time points of Bmp4 explant can be divided into three major stages based on transcriptional similarity: onset-Gastrulation (6 hpf), Mid-Gastrulation (8–10 hpf), and Somitogenesis (12–24 hpf) (Fig. EV4B).

To elucidate the spatial organization of cell types and their developmental trajectories, we performed hybridization chain reaction (HCR) analysis of key cell clusters at six developmental stages. The Bmp4 explant exhibits anterior–posterior (AP) polarity at the onset of gastrulation, with a distinct mesodermal region assigned to the posterior end, which could be further divided into a central domain expressing *aldh1a2* (This cell cluster is referred to as non-dorsal margin involuted, NDMI) and a surrounding domain expressing *tbxta* (This cell cluster is refer to as non-dorsal margin, NDM, Fig. 2D,E). Notably, NDMI and NDM express markers of the cells located in the lateral (*aldh1a2*) and ventral (*eve1*) margins, respectively—populations previously shown to possess organizing capacity (Fauny et al, 2009)—while markers of dorsal margin identity, such as *noto*, are absent (Fig. EV4L,M). By the end of gastrulation (10 hpf), the *aldh1a2*-expressing cell cluster has shifted completely anterior to the *tbxta*-expressing domain, resulting in the formation of anterior–posterior axis of Bmp4 explant (Fig. 2F,G). Single-cell trajectory analysis indicates that the *aldh1a2*-expressing cell cluster exhibits signatures of paraxial and lateral plate mesoderm (Fig. EV3H). Additionally, the neural progenitors, specified during the gastrulation stage, flank the entire mesodermal region (Fig. 2F,G). It is noteworthy that the mesodermal and neural domains specified in Bmp4 explant are radially symmetrical viewed from the sagittal section, suggesting the lack of left—right axis (Fig. 2F,G).

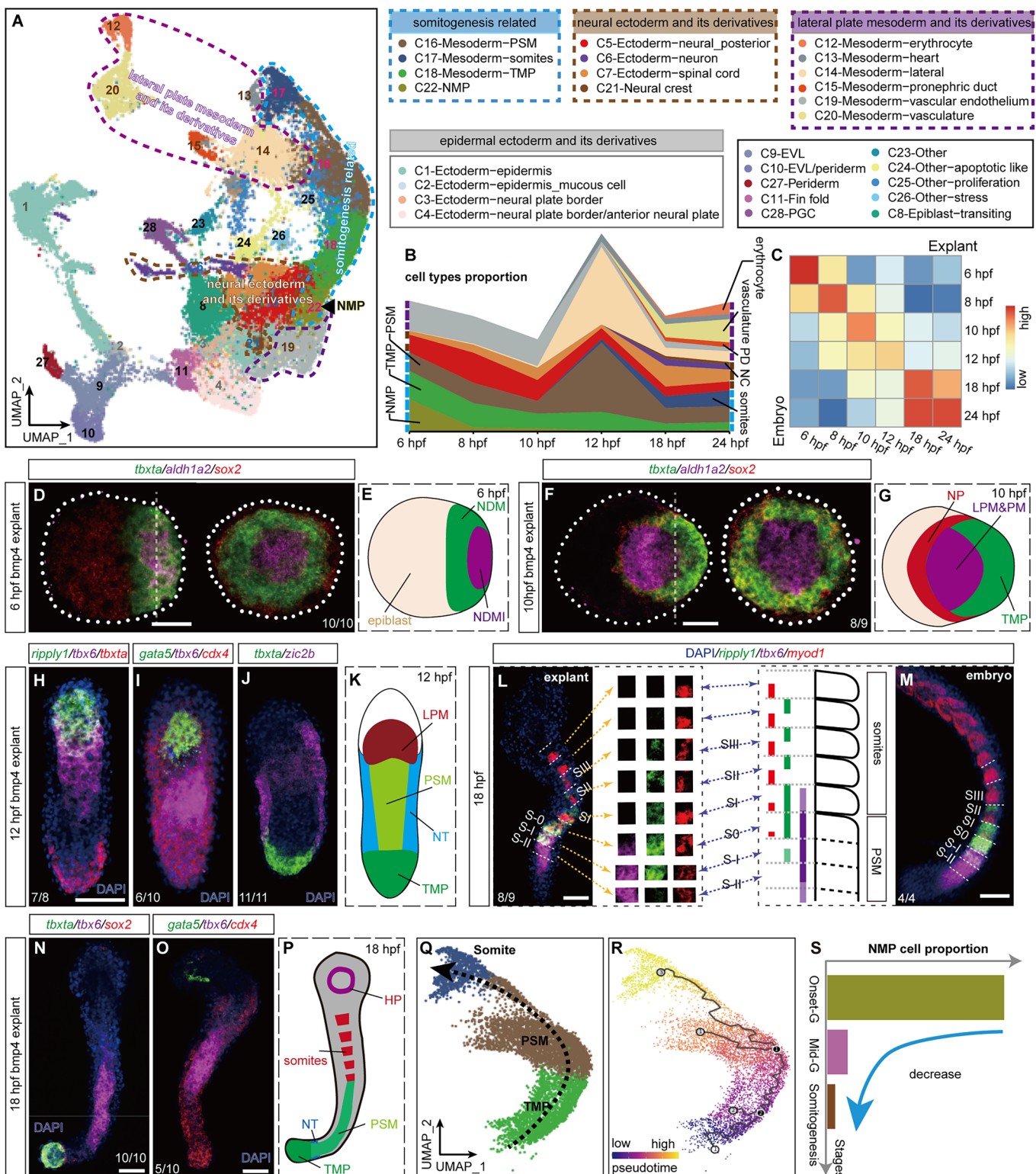

## Recapitulation of somitogenesis in Bmp4 explant

Somitogenesis is a highly conserved developmental process across vertebrates, characterized by a sequential progression of differentiation from head to tail (Aulehla and Pourquié, 2010). During this process,

new somites bud off from the anterior part of the pre-somitic mesoderm (PSM), while a continuous influx of cells from the tailbud replenishes and sustains the posterior PSM. These formed somites then differentiate into various tissues, including vertebrae, muscles, and dermis at later stages of development (Oates et al, 2012).

**Figure 2. Analysis of cellular composition and pattern formation dynamics in Bmp4 explants.**

(**A**) Uniform manifold approximation and projection (UMAP) of integrated scRNA-seq datasets for Bmp4 explants at 6, 8, 10, 12, 18, and 24 hpf, colored by cell clusters. Blue, brown, and purple dotted circles indicate clusters related to somitogenesis, neural ectoderm/derivatives, and lateral plate mesoderm/derivatives, respectively. Arrowhead marks the neural mesodermal progenitors (NMP). (**B**) Ribbon plot showing the temporal proportions of cell clusters related to somitogenesis and lateral plate mesoderm/derivatives in Bmp4 explants. (**C**) Heatmap showing the Pearson correlation between zebrafish embryos and Bmp4 explants at the indicated developmental stages, showcasing the similarity in gene expression patterns. (**D, F**) Left: HCR co-staining of Bmp4 explants at 6 hpf (**D**) and 10 hpf (**F**) for *tbxta*, *aldh1a2*, and *sox2*, with posterior to the right. Right: Sagittal views (from the posterior end) of the same stained explant in the left panel along the dashed lines. Dashed circles demarcate the boundaries of explants. (**H–J**) HCR staining for *ripply1*, *tbx6*, and *tbxta* (**H**), *gata5*, *tbx6*, and *cdx4* (**I**), *tbxta* and *zic2b* (**J**) of Bmp4 explants at 12 hpf. (**L, M**) HCR staining of Bmp4 explant (**L**) or zebrafish embryo (**M**) at 18 hpf for *ripply1*, *tbx6*, and *myod1*, oriented with anterior towards the top. Somites are segmented by white dashed lines. Expression patterns of three genes in each somite of Bmp4 explant are displayed as insets (**L**, right) and schematically for the embryo (**M**, left). (**N, O**) HCR co-staining for *tbxta*, *tbx6*, and *sox2* (**N**), and *gata5*, *tbx6*, and *cdx4* (**O**) in Bmp4 explants at 18 hpf. (**E, G, K, P**) Schematic summaries illustrating patterns of cell clusters in Bmp4 explants at 6 hpf (**E**), 10 hpf (**G**), 12 hpf (**K**), 18 hpf (**P**). NDM non-dorsal margin, NDMI non-dorsal margin involuted, LPM lateral plate mesoderm, PM paraxial mesoderm, PSM pre-somitic mesoderm, HP heart progenitor. All samples in (**H–J, L–O**) were co-stained with DAPI. (**Q, R**) Cell state trajectory analysis of somitogenesis-related cell clusters. Cells are colored by cell types (**Q**) and pseudotime (**R**). (**S**) Time-course analysis of the proportion of NMP-like cells in Bmp4 explants. Each experiment was performed for at least three independent replicates (technical replicates). Scale bars: 50 μm. Source data are available online for this figure.

In Bmp4 explant at the early segmentation stage (12–14 hpf), a distinct PSM region (marked by *tbx6* and *her1*) can be observed located posterior to the LPM (marked by *gata5*) and anterior to the TMP (*tbxta⁺sox2⁻zic2b⁻*) (Figs. 2H–K and EV4F). Notably, midline progenitor cells—which typically give rise to the notochord, hypochord, and floor plate in the tail region (Row et al, 2016)—are absent in Bmp4 explants (Figs. 2J,K and EV4F). Instead, the progenitor zone in Bmp4 explant (TMP) resembles the mesodermal progenitor niche (Martin and Kimelman, 2010) or tail organizer (Agathon et al, 2003) that gives rise to somitic mesoderm (Fig. 2H–K). By 18 hpf, Bmp4 explant exhibited distinctive molecular characteristics that made it easily distinguishable between mature, newly formed, and prospective somites (Windner et al, 2015). *tbx6* is highly expressed throughout the anterior PSM (S0, -I, -II), decreases in the newly formed somite (SI), and becomes undetectable beyond SII. The expression of *ripply1* initiates at the posterior of S-I, upregulated throughout S0 and S1, restricted to the anterior half of SII, SIII, and becomes barely detectable beyond SIII. In most cases, *myod1* expression is restricted to the posterior half of the somite in Bmp4 explant, indicating the onset of myogenesis and the proper establishment of anterior–posterior polarity in the somites (Fig. 2L,M). In addition, *cdx4* expression extends from the TMP to the *gata5⁺* domain, suggesting that the majority of Bmp4 explant exhibits ventral and posterior identities (Fig. 2N–P).

Single-cell trajectory analysis also revealed a clear differentiation trajectory that initiates from the tailbud cell state, progresses though PSM, and ultimately ends at somites state (Fig. 2Q,R). Along this trajectory, we identified many genes that remain poorly studied regarding their roles in somitogenesis, in addition to those genes previously reported to play roles in the differentiation from TMP to PSM and from PSM to somites. Thus, Bmp4 explant holds great potential as a valuable model for future research in somitogenesis (Figs. EV3C–G,I–K).

NMP cells in zebrafish have been demonstrated to be bipotential (Row et al, 2016). However, a single NMP cell rarely gives rise to both neural and mesodermal descendants, due to its low proliferation frequency (Martin and Steventon, 2022). We also identified the presence of NMP-like cells expressing *tbxta* and *sox2* in Bmp4 explant and observed similar biological characteristics of NMP cells in zebrafish (Figs. 1F and EV4A,C,D). The proportion of NMP-like cells decreased quickly from gastrulation to segmentation stages (Figs. 2N,P,S, 3G, and EV3B), and they tended to stay in G1 phase rather than advancing to S phase, in contrast to TMP and neural cells (Fig. EV4E), suggesting the low proliferating frequency of the NMP-like cells.

Collectively, these findings demonstrate that Bmp4 explants undergo a somitogenesis progression akin to that observed in vivo.

## Caudal organ primordia are orderly organized in the Bmp4 explant at 24 hpf

By the end of 24 hpf, the zebrafish body plan is well established, with clearly apparent organ primordia (Kimmel et al, 1995). To systematically compare Bmp4 explant with the embryo at this stage, we integrated our single-cell datasets from Bmp4 explant with a published embryonic single-cell dataset (Wagner et al, 2018) at 24 hpf (Fig. 3A–C; Dataset EV3). Unsupervised clustering revealed 30 distinct cell clusters. We observed that the majority of cells in Bmp4 explant are ventrolateral mesoderm derivatives, including somites, endothelium, pronephric duct, erythroid and other related cell types (Fig. 3D). In contrast, brain cells (forebrain, midbrain, hindbrain, etc.) were minimally present in Bmp4 explant, with only a few posterior neural cells (such as the spinal cord and neural crest) being detectable (Fig. 3D,E). Notably, no endoderm or axial mesoderm derivatives (pharyngeal pouch, hatching gland, and notochord) were detected in the Bmp4 explant (Fig. 3F).

To determine whether the organ primordia in Bmp4 explant share similar developmental trajectories with their in vivo counterparts, we employed two computational methods to reconstruct the single-cell developmental trajectories of Bmp4 explant. First, we utilized a simulated diffusion-based computational approach (URD) (Farrell et al, 2018), which traced the origins of the tailbud mesoderm progenitor (TMP), pre-somitic mesoderm (PSM), and somites back to a shared cluster of progenitors at an earlier stage. Similarly, vasculature and erythrocytes, the pronephric duct (PD), heart, spinal cord, and neural crest were observed to originate from distinct progenitor clusters (Fig. 3O). Subsequently, we employed the K-Nearest Neighbor algorithm (KNN) on our time-series single-cell datasets of Bmp4 explant (see "Methods"). During the segmentation stage, we observed progressive differentiation of the neural plate into neural crest cells and Rohon-Beard neurons (12–24 hpf). Notably, the NMP-like cells, which have the potential to generate both mesoderm and neural ectoderm cells, have already disappeared by the onset of segmentation. This observation may explain the diminishment of the spinal cord during this stage. In addition, clear progression was observed as the paraxial mesoderm

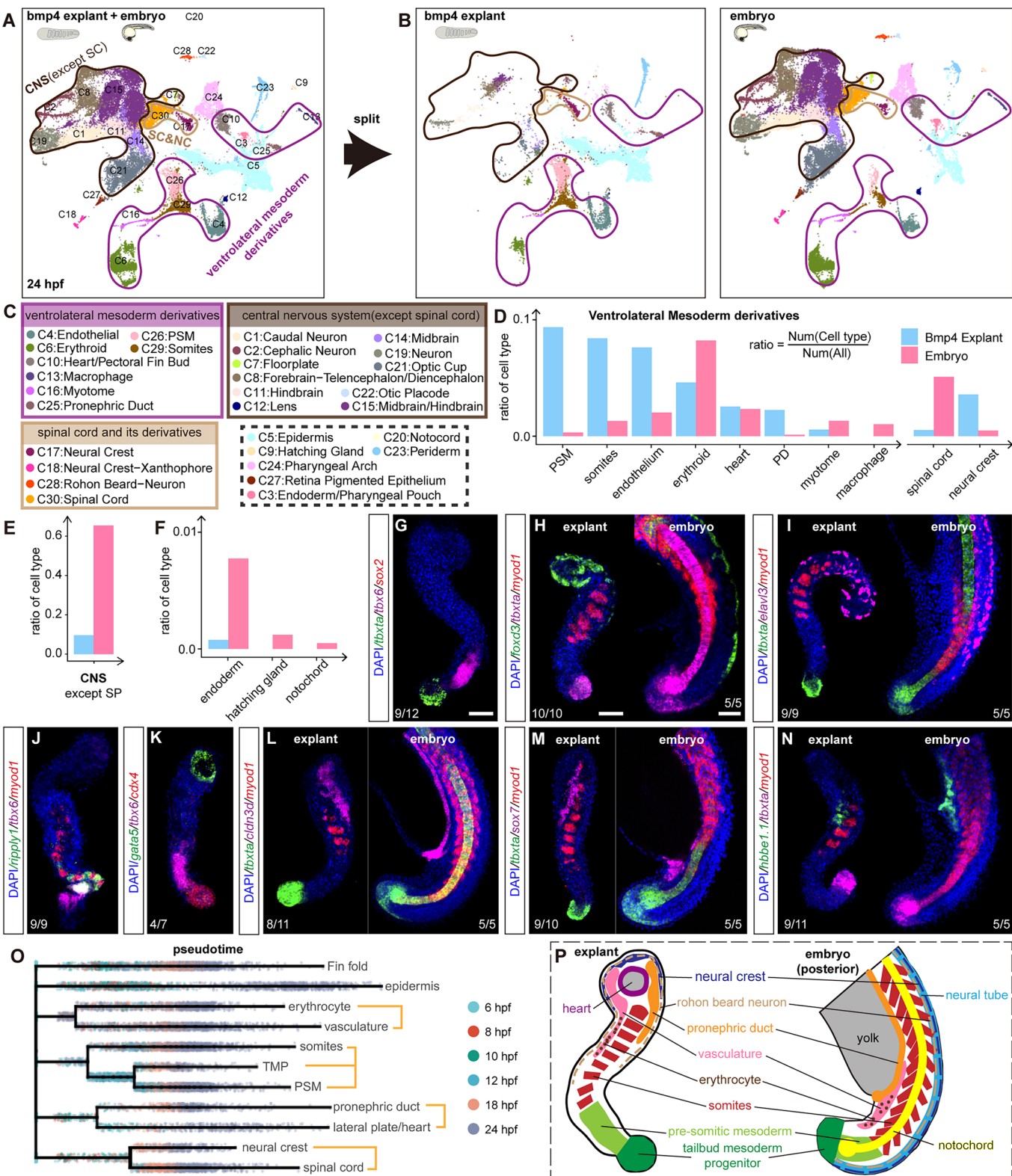

The EMBO Journal Volume 45 | Issue 1 | January 2026 | 210 – 242

**Figure 3. Characterization of Bmp4 explants at the organogenesis stage.**

(A, B) UMAP plots showing integrated single-cell RNA sequencing (scRNA-seq) datasets of zebrafish embryos and Bmp4 explants at 24 hpf. (A) Displays the combined datasets with cells colored by the cell clusters listed in (C). (B) Shows separated datasets for Bmp4 explants (left) and zebrafish embryos (right). Black, purple, and yellow circles highlight cell clusters of the central nervous system (excluding spinal cord), ventrolateral mesoderm derivatives, and spinal cord & its derivatives, respectively. (D–F) Comparisons of the proportions of indicated cell clusters between Bmp4 explants and zebrafish embryos at 24 hpf. PD pronephric duct, SP spinal cord, NC neural crest, CNS central nervous system. (G, J, K) HCR co-staining for *tbxta*, *tbx6*, and *sox2* (G), *ripply1*, *tbx6* and *myod1* (J), *gata5*, *tbx6*, and *cdx4* (K) in Bmp4 explants at 24 hpf. (H, I, L–N) HCR co-staining for *foxd3*, *tbxta*, and *myod1*(H), *tbxta*, *elavl3*, and *myod1* (I), *tbxta*, *cldn3d*, and *myod1*(L), *tbxta*, *sox7*, and *myod1* (M), *hbbe1.1*, *tbxta*, and *myod1* (N) in Bmp4 explants (left) at 24 hpf and zebrafish embryos (right, only the posterior part is displayed) at 20–24 hpf. Anterior is toward the top for both the explants and embryos. Samples in (G–N) were all co-stained with DAPI. (O) Force-directed layout of Bmp4 explant scRNA-seq data. Cells are color-coded by developmental stages, with terminal cell clusters labeled by cell identities. (P) Schematic summary illustrating structures of Bmp4 explant at 24 hpf, compared to the caudal part of the zebrafish embryo. Each experiment was performed for at least three independent replicates (technical replicates). Scale bars: 50 μm. Source data are available online for this figure.

differentiates into PSM, which further differentiates into somites. Furthermore, the TMP contributed concurrently to the PSM during this period (Fig. EV3B). These results indicate that both trunk and tail somites are specified in the Bmp4 explant, with trunk somites originating from the paraxial mesoderm formed during gastrulation, while tail somites are specified from cells in the TMP (Attardi et al, 2018; Holley, 2006; Row et al, 2016).

To investigate the arrangement of organ primordia within the Bmp4 explant, we performed HCR co-staining for key makers of each organ primordium. We observed that cell types related to somitogenesis display a pronounced spatial distribution along the anterior–posterior (AP) axis: the tailbud mesoderm progenitor is located at the most posterior end of the Bmp4 explant, while the PSM is positioned anterior to the TMP yet posterior to the somites (Fig. 3G,J), mirroring patterns observed in the embryo. Furthermore, ventrolateral mesoderm derivatives are located at the anterior part of the Bmp4 explant: the heart primordium is found at the very anterior end; the pronephric duct, vascular endothelium, and erythrocytes are located between the somites and the heart region (Fig. 3K–N). The pronephric duct is in the trunk region of the embryo. It is noteworthy that the posterior part of the PD overlaps with 2–3 anterior somites in the Bmp4 explant, which are thus morphologically identified as trunk somites (Fig. 3L). The spinal cord, which is located anterior to the TMP, is barely detectable, but its related derivatives, such as neural crest and Rohon-beard neuron, are found at the anterior of the Bmp4 explant, surrounding the ventrolateral mesoderm derivatives (Fig. 3G–I). The normal anterior limits of expression of central/posterior *hox* genes (*hoxd10a/hoxd12a*, Fig. EV4G) and central/anterior *hox* genes (*hoxc6b/hoxb3a*, Fig. EV4H) further demonstrate that the Bmp4 explant is well organized anterior–posteriorly (Hayward et al, 2015).

In the zebrafish embryo, the anatomical trunk—comprising the heart, pronephric duct, and trunk somites—extends from the posterior of the head to the anus, while the tail, which includes tissues such as tailbud, tail somites, and vasculature, is located posterior to the anal opening. Comparison of Bmp4 explant with the embryo reveals that the majority of tissues in Bmp4 explant—TMP, PSM, posterior somites, vasculature and erythrocyte—compose a tail-like structure. Other tissues, such as the pronephric duct, anterior trunk somites, and spinal cord derivatives, are located near the tail in both Bmp4 explant and embryo. Only the heart is an exclusively anterior structure in the Bmp4 explant (Fig. 3P).

Taken together, these results further demonstrate that the Bmp4 explant develops into a VCLS lacking axial mesoderm and endoderm at the organogenesis stage.

## Characteristics of gastrulation and tail elongation cell movements in Bmp4-induced VCLS

During embryonic development, coordinated cell movements coupled with cell fate specification dynamically sculpt the evolving morphology of the embryo (Kanki and Ho, 1997) (Fig. 4A). To evaluate how cell movements shape this VCLS and to what extent it recapitulates embryonic development, we conducted timelapse confocal imaging and analyzed cell movements using Imaris. Morphogenesis of Bmp4 explant can be divided into two consecutive phases, the gastrulation and segmentation stages (Fig. 4D,M), consistent with in vivo development. During the gastrulation stage, cells near the injection site move toward the anterior of the explant and internalized outside-in very quickly (Fig. 4B,B'; Movies EV2 and EV3), leading to the formation of a transient hole reminiscent of the blastopore. This hole eventually closed near the injection site where the TMP cell cluster resides, reminiscent of the fusion of blastoderm margin in vivo (Figs. 4C,D and 2G). Analysis of the movement direction and the changes in neighboring cells of internalizing cells at different time points reveals that most cells move in the same direction. At early stages, a significant number of cells move together, while at later stages, they move more discretely, suggesting a group of cells showed collective movement at early stages and individual movement at later stages (Fig. 4E–G). This is consistent with in vivo studies in zebrafish, where the majority of margin cells internalize collectively, although some cells ingress individually (Pinheiro and Heisenberg, 2020). Similar processes can also be observed on the animal pole of the embryo that is injected with *bmp4* (Figs. EV5A–F; Movies EV4 and EV5). These results indicate that the posterior region of early VCLS recapitulates both molecular and morphogenetic features of the blastoderm margin.

To investigate which cluster of cells internalized, we tracked the cell movements of a Bmp4 explant constructed from a transgenic line Tg (*aldh1a2*:H2B-RFP). Cells near the injection site exhibited red fluorescence before the onset of internalization, suggesting that cell fate specification preceded cell movement for this cluster (Figs. EV5G,H; Movies EV6 and EV7). During gastrulation, *aldh1a2*-expressing cells (LPM and PM) progressively internalize, resulting in the formation of a red fluorescent region anterior to the non-internalizing, non-red fluorescent region by the end of gastrulation (Fig. EV5I). It is noteworthy that some clones of *bmp4* injected blastomere at the 128-cell stage internalize at the leading edge, expressing *aldh1a2* and are specified as LPM. The remaining cells stay at the injection site, and are specified as enveloping layer as indicated by the expression of *krt18* (Fig. EV5R–V).

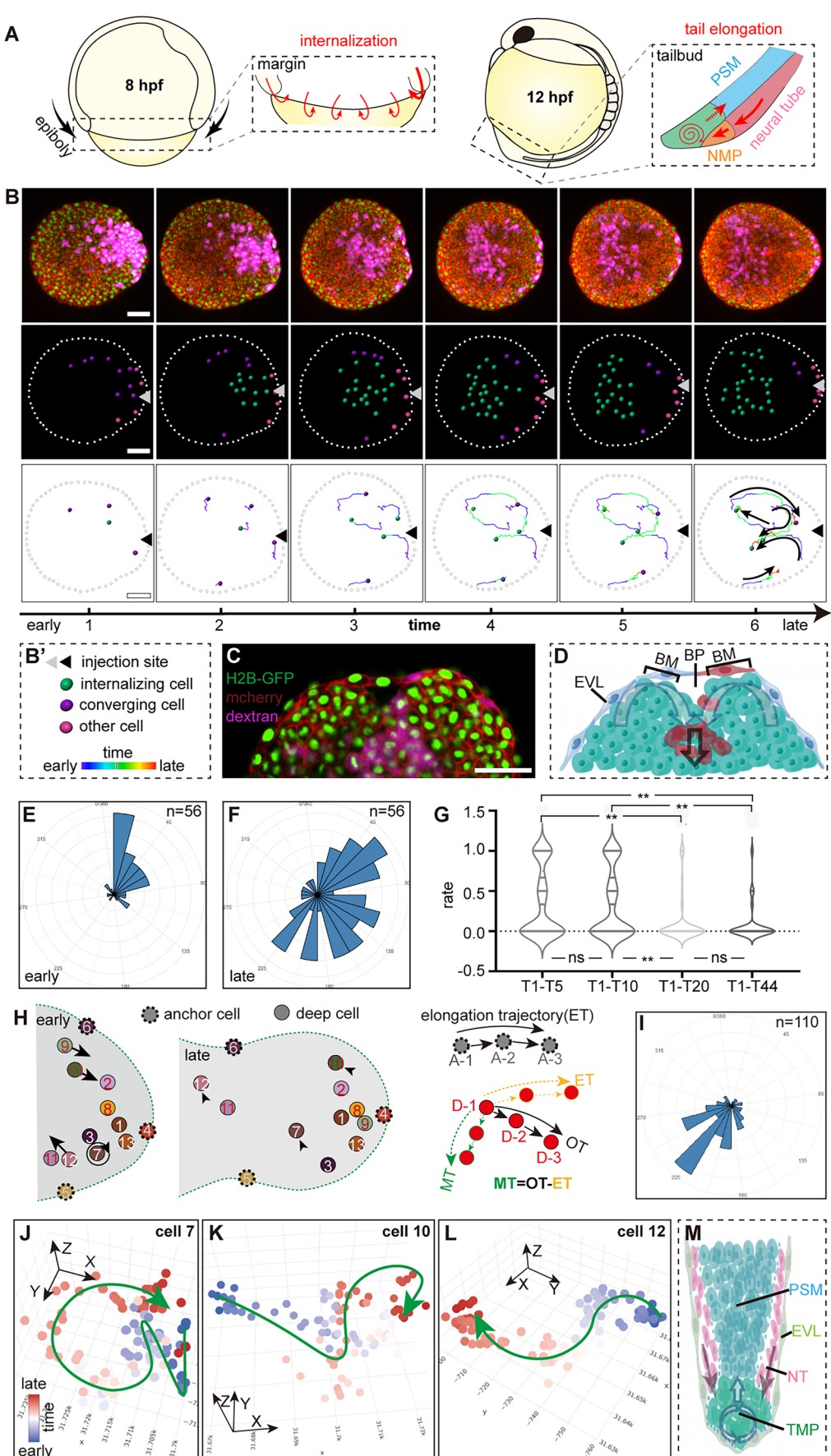

**Figure 4. Morphogenetic dynamics of Bmp4 explants.**

(A) Schematic representation of gastrulation and tail elongation cell movements in the zebrafish embryo. PSM, pre-somitic mesoderm. (B) Time-lapse imaging of Bmp4 explants during gastrulation, as detailed in Movies EV2 and EV3. Upper panel: Representative images of Bmp4 explants with cell nuclei, cell membrane, and descendants of *bmp4*-injected blastomeres labeled with H2B-GFP, mCherry, and Dextran Fluorescein (purple), respectively. Middle panel: Images with selected cells colored by states of cell movements as displayed in (B'). Lower panel: Images showing movement trajectories with only five representative trajectories displayed and summarized by white arrows for clarity. The time course of trajectories is color-coded from blue to red as displayed in (B'). (C) Representative image of Bmp4 explants during gastrulation, referencing Movie EV2. Labeling details are as described in (B). (D) Schematic representation of gastrulation cell movements in Bmp4 explants. EVL, enveloping layer; BM, blastoderm margin; BP, blastopore. A black hollow arrow indicates the direction of involution movement. (E, F) Rose plot of cell movement directions of internalizing cells of Bmp4 explants at early (E) and late (F) timepoints (within 3 min). (G) Statistical analysis of the numbers of the adjacent cells of internalizing cells. Significance was calculated by one-way ANOVA. **$P < 0.001$. Exact *P* values: T1-T5 vs T1-T10, >0.9999; T1-T5 vs. T1-T20, 0.0032; T1-T5 vs. T1-T444, 0.0016; T1-T10 vs. T1-T20, 0.0032; T1-T10 vs. T1-T44, 0.0016; T1-T20 vs. T1-T44, 0.9978. (H) Schematic representation of positions of tracked cells at early (left) and late (middle) time points. About 110 cells were tracked, and the 13 cells in the diagram were labeled as representatives. Dashed circles indicate the EVL cells used as anchor cells. Solid circles indicate deep cells. The strategy for analyzing migration trajectories is outlined to the right, including calculation of the elongation trajectory using three anchor cells and subsequent adjustment using the observed trajectories of deep cells to obtain migration trajectories (see "Methods"). A anchor cell, D deep cell, ET elongation trajectory, OT observed trajectory, MT migration trajectory. (I) Rose plot of cell movement directions of cells in the TMP within a period of about 1 h. (J–L) Movement trajectories of cell 7, 10, and 12, respectively, with dots representing the 3D positions of the tracked cells and green lines with arrows manually delineated to indicate the movement trajectories. (M) Schematic of tail elongation cell movements in Bmp4 explants. Each experiment was performed for at least three independent replicates (technical replicates). Scale bars: 50 μm. Source data are available online for this figure.

Then, to ascertain the primary driver of axis elongation during the segmentation stage (Fig. EV5J,K; Movies EV8 and EV9), we first compared the elongation rates of the anterior and posterior parts of the Bmp4 explant, respectively. We found that the posterior part elongated much faster than the anterior part (Fig. EV5L–N), suggesting that the TMP which is located at the posterior end contribute significantly to the elongation of the entire Bmp4 explant during the segmentation stage. During embryonic tail elongation, cells in the tailbud exhibit local rotational yet forward movements, contributing to the posterior of PSM, and thereby driving the axis elongation (Das et al, 2019; Lawton et al, 2013). We analyzed trajectories of more than 100 different cells in posterior end of Bmp4 explant, which exhibited different kinds of movement trajectories (several representative cells were chosen to shown their movement trajectories): EVL cells (cell 4, 5, 6) and deep cells (cell 1, 2, 3, 7–13) within or near the TMP (Fig. 4H). Given that EVL cells are interconnected by tight junctions and typically do not migrate, the consistent, straightforward trajectories observed for all EVL cells (observed trajectories, OT; Figs. 4H and EV5O,P,Q; Movie EV10) reflect overall explant elongation. Thus, the three EVL cells were defined as anchor cells, with their trajectories representing the explant elongation, which is referred to as the "elongation trajectory (ET)". Subsequently, we calculated the "migration trajectories (MT)" for other cells by subtracting the ET from OT (Fig. 4H). It turned out that most cells exhibited stereotypical rotating trajectories (cells 7, 10, 12, and Figs. 4J–L and EV5O–Q) and cell movement was random at different timepoints (Fig. 4I), resembling the local rotational cell movements observed in the embryonic tailbud in vivo (Lawton et al, 2013). Some cells in the TMP migrated anteriorly in a zigzag pattern and integrated into the PSM (cell 12, Figs. 4L and EV5P). Some cells adjacent to the TMP migrated posteriorly into the TMP (cell 7, 10 and Figs. 4J,K and EV5O,Q).

## iBMP4 human pluripotent stem cells function as an organizer of VCLS

The term 'organizer' is functionally defined as a cluster of cells capable of inducing a secondary axis when transplanted ectopically into host embryos (Spemann and Mangold, 2001). The result that

the posterior end of Bmp4 explant displays molecular and morphogenetic characteristics of non-dorsal blastoderm margin (Fig. EV4L,M) which can organize trunk and tail structures in vivo prompts us to hypothesize that *bmp4* may induce a "ventral–caudal organizer" capable of organizing the formation of caudal structures that lack axial mesoderm and endoderm. The cross-species transplantation strategy is generally accepted as the most stringent test for functionally defining an organizer (Knoetgen et al, 2000; Martyn et al, 2018; Zhu et al, 1999).To validate this hypothesis and, furthermore, to test whether this function of BMP4 is conserved among vertebrates, we utilized a transgenic H9 hESCs with doxycycline-inducible expression of human *BMP4 (iBMP4)-cds* (Luo et al, 2025) for the xenograft experiment. Before transplantation, these DOX-inducible hESCs were treated with 1 μg/mL doxycycline for 24 h. Then, about 20–50 iBMP4 induced human PSCs were grafted into zebrafish animal poles at sphere stage (see "Methods", Fig. 5A). Transplanted human PSCs express human *BMP4*, and BMP4 induced its own expression, as evidenced by the co-expression of *iBMP4-cds* and *iBMP4-intron* (Fig. EV4O). Zebrafish cells surrounding transplanted human PSCs are rapidly observed to be specified as posterior mesoderm, as indicated by the expression of *cdx4* (Fig. EV4N). At 24 hpf, we observed that transplanted human cells induced a secondary caudal axis with sequential somites, resembling that obtained by injecting *bmp4* mRNA (Fig. 5B,C). Whole mount in situ hybridization of *tbxta* revealed that TMP is present at the posterior end, and no axial structures are observed in the secondary axis (Fig. 5D). Transplantation of untreated iBMP4 cells or H9 cells into the animal pole of zebrafish blastula failed to induce the development of secondary tail (Fig. EV4P). Taken together, this functional test demonstrated that iBMP4 human PSCs function as an organizer instructing VCLS formation, and this function of BMP4 is conserved between zebrafish and humans.

## *BMP4* induces caudal cell fates and elongation in human 3D PSC aggregates

To further investigate the organizing capability of *iBMP4* human PSCs, we conjugated an aggregation of *BMP4*-inducible PSCs with an aggregation of normal human PSCs. They fused together quickly

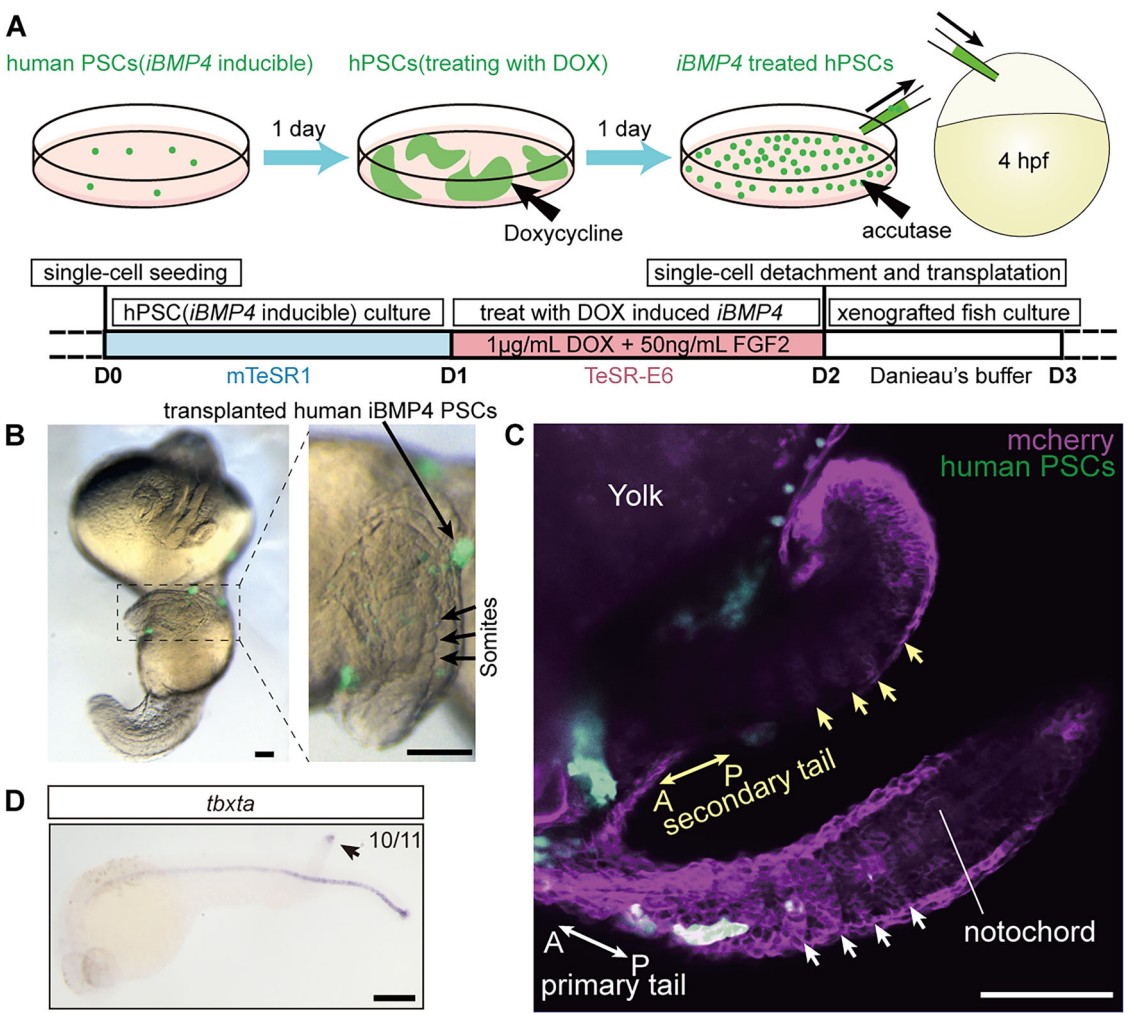

**Figure 5. Xenografting *iBMP4* Human PSCs into Zebrafish Embryos.**

(A) Schematic illustrating the xenograft procedure of *iBMP4*-treated hPSC into zebrafish embryos. (B) Bright field images showing the secondary axis induced by *iBMP4* hPSCs labeled with GFP. Long black arrow indicates hPSC clones. Short black arrows highlight the somites in the secondary axis. (C) Confocal microscopy image of a zebrafish showing both the primary and secondary axes post-xenograft. Host zebrafish cells and transplanted *iBMP4* hPSCs are labeled with mCherry and GFP, respectively. Arrows indicate the A–P polarity and the somites in the secondary and primary axes, respectively. Notochord was not detected in the secondary axis. (D) WISH of *tbxta* in the xenografted embryo at 24 hpf. Arrowhead indicates the ventral–caudal organizer in the secondary axis. Each experiment was performed for at least three independent replicates (technical replicates). Embryos with morphologically observed ectopic induced tissue were used for in situ hybridization of *tbxta*. Scale bars:100 μm. Source data are available online for this figure.

and formed a 3D embryoid body (EB) at day 2 (Fig. 6A). Upon DOX induction, a BMP signaling gradient was established from the end producing BMP4 to the opposite end (Fig. 6C,D). When the EBs were cultured in mTeSR1 medium, they exhibited ordered arrangements of markers of trophectoderm (TE, CDX2$^+$), endoderm (SOX17 + ), mesoderm (T + ) and ectoderm (SOX2 + ) from the BMP4 expressing end to the opposite end upon *BMP4* induction, resembling the 2D micropatterned gastruloids (Martyn et al, 2018) (Fig. EV6G), however, they did not undergo significant morphogenetic movements and maintained a ovoid-like morphology. In contrast, when the EBs were cultured in E6 medium, only non-anterior mesoderm was induced, as no CDX2, CER1 or SOX17 expression was detected (Fig. EV6H) and the EB could elongate progressively, reaching a length of about 0.5 mm after 5 days of development, with segments observable at this stage

(Figs. 6B and EV6A), while EB without DOX induction stayed spheroidal at day 5 (Fig. EV6F). The elongation of EB was arrested at day 6 (Fig. 6A,B). Further assessment of the expression patterns of key markers in *BMP4*-induced EB revealed similarities to vertebrate VCLS. We observed NMP-like cells (TBXT$^+$ and SOX2$^+$) at day 4, and segmented somitic mesoderm (MEOX1$^+$) at day 5 in BMP4-induced EBs (Fig. 6E,F). In the vertebrate embryo, MESP2 (Whittock et al, 2004) is expressed in the anterior PSM and newly formed somites, and the expression of TBX18 (Kraus et al, 2001) is restricted to the anterior halves of the somites. In the *BMP4*-induced human EBs, the region expressing MESP2 was broad and opposed to the ectodermal region expressing SOX2, suggesting the presence of the PSM (Fig. 6G). The expression of TBX18 was evident, although not in a striped pattern (Fig. 6H), indicating that somites are formed.

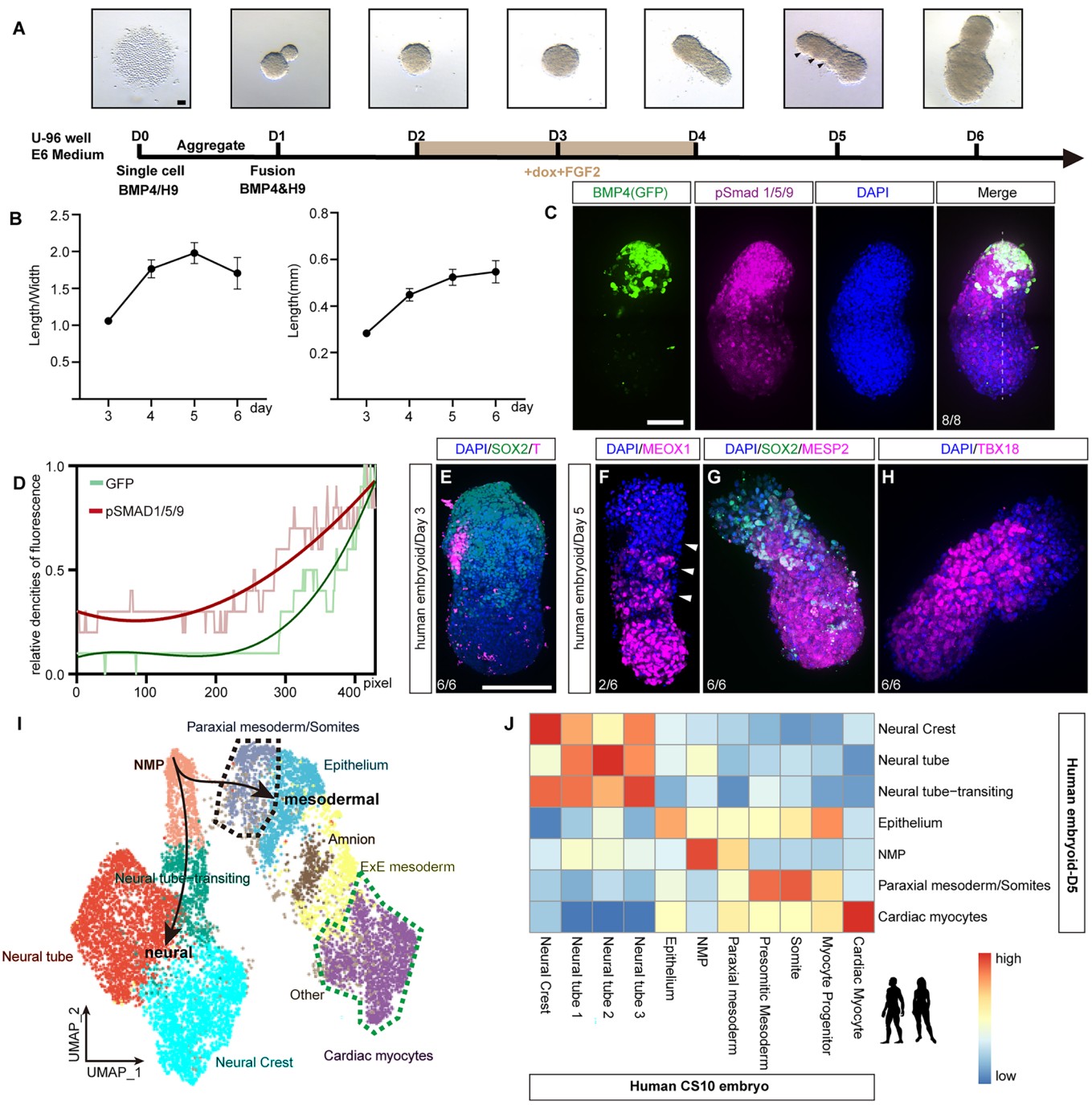

**Figure 6. Construction of human ventral–caudal-like structure (VCLS) with *BMP4*.**

(A) Representative images depicting the method used to induce human VCLS with BMP4. (B) Quantification analysis of the length (right) and length/width ratio (left) of VCLS. Data are presented as mean ± SD of three independent experiments (biological replicates). (C) Immunofluorescence staining of phosphorylated Smad1/5/9 and co-stained with DAPI in VCLS. The *BMP4* transgenic cells are labeled with GFP. (D) Quantification of pSmad1/5/9 and GFP signals along the dashed line shown in (C), normalized to their respective peak signals. (E–H) Immunofluorescence staining for T and SOX2 (E), MEOX1 (F), MESP2 and SOX2 (G), TBX18 and T (H) with DAPI co-stained in VCLS. (I) UMAP plot showing the cell clusters in VCLS at day 5, with cells colored according to cell type. Black lines with arrows depict potential differentiation trajectories from NMP-like cells towards neural or mesodermal lineages. (J) Heatmap showing the Pearson correlation between gene expression profiles in the human embryo and VCLS. Each experiment was performed for at least three independent replicates (technical replicates). Scale bars: 100 μm. Source data are available online for this figure.

To further analyze the cellular and molecular features of *BMP4*-induced human EBs, we performed scRNA-seq analysis. At day 5, we identified cell clusters such as NMP-like, paraxial mesoderm/somites, neural tube, and cardiac myocytes that resemble the caudal counterparts in vivo (Fig. 6I,J; Dataset EV4). For example, the paraxial mesoderm/somites cell cluster in the EB displays high expressions of *MEOX1* and *CYP1B1*, and the cardiac myocytes cell cluster displays high expressions of *MYOCD* and *FN1*, similar to the situation in the human embryo at Carnegie stages 10 (Zeng et al, 2023) (Fig. EV6B–E). At day 6, we also identified NMP-like that expresses *SOX2* and *NKX1-2*, paraxial mesoderm/somites that express *MEOX1* and *CYP1B1*, and cardiac myocytes that express *MYOCD* and *FN1*, similar to the situation in Monkey embryo at Carnegie stages 8–11 (Zhai et al, 2022) (Fig. EV7A–E). However, the overall paraxial mesoderm/somites cell fates were greatly diminished and transcriptionally resembled epithelium (Fig. EV7A,F), suggesting that somitogenesis was not normally progressed from day 5 to day 6.

Taken together, the above observations imply that *BMP4*-induced EBs could exhibit cellular and molecular features of human caudal structures, similar to the VCLS in zebrafish. Further modifications to culture conditions may improve the progress of somitogenesis in this model system.

# Discussion

In evolution and embryonic development, key regulators such as morphogens and transcription factors are repeatedly utilized across various locations and developmental stages (Gilmour et al, 2017; Langdon and Mullins, 2011). It is well established that BMP4 is critical for dorsal-ventral patterning, tailbud development, and organogenesis, with its functions conserved across a variety of species (Bier and De Robertis, 2015). In this study, we have demonstrated that *bmp4* overexpression at the animal pole of zebrafish robustly induces the formation of a partial secondary axis with caudal-like structure, and after explantation, the resulting structure exhibited anterior–posterior (A-P) patterned caudal tissues but lacked anterior neural ectoderm, axial mesoderm, and endoderm, which was defined as VCLS. Agathon et al and our previous work showed that *bmp2b/bmp4* in combination with *ndr1*, rather that *bmp2b* alone, could induce secondary tails in vivo, which were organized by the tail organizer specified by triple stimulation of Nodal, Wnt, and BMP (Agathon et al, 2003). We demonstrated that zebrafish *bmp4* alone, at an appropriate dosage and without the assistance of Nodal, is capable of providing the triple stimulation of Nodal, Wnt, and BMP, leading to the formation of a ventral–caudal organizing center, both in vivo and in vitro. These findings are consistent with previous in vivo loss-of-function studies of BMP4 (Stickney et al, 2007; Winnier et al, 1995).

Our previous work demonstrated that opposing gradients of BMP and Nodal signaling in zebrafish can induce a complete body axis (Xu et al, 2014). However, the precise mechanistic underpinnings remained unclear—particularly the specific cell types and spatial patterns induced by Nodal or BMP alone. In our recently published study, we showed that Nodal signaling alone is sufficient to drive axial mesendoderm development and orchestrate head patterning (Cheng et al, 2023). In the current study, we found that BMP4 alone can recapitulate the functional role of a ventral–caudal organizer, inducing the development and morphogenesis of VCLS

that lack anterior neural ectoderm, axial mesoderm, and endoderm —tissues typically induced by Nodal signaling (Cheng et al, 2023)— in both zebrafish and humans. The current study and our previous study further demonstrated that Nodal and BMP are minimal factors for completed body axis induction (Xu et al, 2014). These studies suggest that embryonic tissues can be induced in vitro as modular units along a defined axis.

Using single-cell RNA sequencing analysis and combined with HCR and in situ hybridization for spatial characterization, we showed that this model system displays molecular features characteristic of ventral–caudal-like structure, organized orderly in the anterior–posterior axis, and parallels the developmental trajectories of its in vivo counterparts (Farrell et al, 2018). At the beginning of gastrulation, we identified two cell clusters, NDMI and NDM, correspond to the ventral margin and lateral margin in vivo, which have been demonstrated to have organizing potential (Fauny et al, 2009). We did not find cell clusters correspond to the dorsal margin (organizer) in the Bmp4 explant. At 24 hpf, Bmp4 explants developed somites and some lateral plate mesoderm derivatives orderly arranged anterior–posteriorly but lack axial mesoderm and head structures, resembling the vertebrate ventral–caudal region, and thus we designate this model system as a ventral–caudal-like structure (VCLS). The structure of VCLS is very much like that of highly ventralized zebrafish mutants like *ichabod* (Kelly et al, 2000) and *huluwa* (Yan et al, 2018), which also cannot form the organizer at the beginning of gastrulation.

Notably, analysis of morphogenetic cell movements in this VCLS revealed remarkable similarities to embryonic gastrulation and tail elongation processes observed in vivo: during gastrulation, internalization was prominent, driving the precursors of paraxial mesoderm and lateral plate mesoderm (LPM) anteriorly, thereby establishing the A-P axis of the VCLS. Intriguingly, a transient blastopore was observed at the posterior of the VCLS—a feature not typically visible in zebrafish embryos due to the presence of the yolk cell (Serrano Nájera and Weijer, 2023). As dramatic elongation commenced, we observed several classical morphogenetic cell movements, including the irregular movement of TMP cells, their integration into the anterior PSM, and the migration of neural cells towards the TMP. It is noteworthy that cell proliferation and convergence and extension in the formed somites and PSM likely contribute to the elongation of the anterior part of the VCLS, as similarly described in vivo (Steventon et al, 2016; Thomson et al, 2021).

The concept of "organizer" was first introduced by Hans Spemann and Hilde Mangold in 1924 (Spemann and Mangold, 1924). It traditionally refers to a region at the dorsal blastopore of the embryo that can induce a secondary axis when transplanted to the ventral side of a host embryo. Subsequent studies identified additional embryonic regions with organizing properties, such as the tail organizer in zebrafish, which arises from the ventral blastoderm margin and can induce a secondary tail when grafted to the animal pole (Agathon et al, 2003; Martinez Arias and Steventon, 2018). In fact, any portion of the blastoderm margin in zebrafish embryo is potentiated to organize structures whose AP features is determined by the margin's position along the dorsoventral axis (Fauny et al, 2009). The blastoderm margin in VCLS (Fig. 4D) is composed of two cell clusters: one cell cluster (NDM) that stays in the posterior end of VCLS and contributes to TMP corresponds to the most ventral margin in the embryo; the other cell cluster (NDMI) that internalizes and contributes to paraxial mesoderm and LPM corresponds to the lateral margin of embryonic blastoderm (Figs. 2E and EV3B). Considering that the trunk components in VCLS

are limited at 24 hpf, the organizing potential of NMDI is compromised compared to its compartments in vivo. Thus, we name the whole blastoderm margin in VCLS "ventral–caudal organizer" that roughly corresponds to the non-dorsal margin of the embryo. Xenografting of *BMP4*-treated human PSCs to the zebrafish animal pole led to the formation of a secondary axis with ventral–caudal tissues, demonstrating the function of "ventral–caudal organizer".

When cultured under 2D micropatterned conditions in mTeSR medium supplemented with BMP4, human embryonic stem cells consistently differentiate into gastruloids, which exhibit radial arrangements of three germ layers and extraembryonic layers (Deglincerti et al, 2016; Warmflash et al, 2014). But this model did not display any morphogenetic movements. Aggregates of mouse embryonic stem cells treated with BMP4 function as a dorsal organizer, resulting in the development of embryoids with three germ layers (Xu et al, 2021). Our xenograft assay indicates that it may be feasible to construct human ventral–caudal-like structures by BMP4-induced ventral–caudal organizer. We build a 3D embryoid body by conjugating *BMP4*-inducible PSCs with an aggregation of normal human PSCs. While EBs exhibit ordered arrangements of three germ layers and TE-like cells when cultured in mTeSR upon *BMP4* induction, those cultured in E6 medium only specify non-anterior mesoderm, suggesting that BMP4 functions in a context-dependent manner in in vitro cultured system.

In our current model, scRNA-sequencing analysis identified caudal cell fates, such as neural mesodermal progenitor-like cells (NMP-like) and somites, in human PSC aggregates. However, after 5 days of development, the elongation and segmentation processes ceased in this human VCLS. Further modification of culture conditions or incorporation of additional signaling factors may be tested to better recapitulate human somitogenesis.

Overall, our work demonstrates that BMP4 can robustly induce the ventral–caudal organizer, a function that is highly conserved across different species. To validate this crucial role of BMP4, we employed a variety of experimental systems, including the in vivo overexpression of *bmp4* at the animal pole in zebrafish, zebrafish explant systems, xenografting of human pluripotent stem cells (PSCs) into zebrafish embryos, and 3D aggregates of human PSCs. We believe that by combining single-cell transcriptomics and live imaging techniques, the research platforms established in this study cannot only be effectively used to study the role of morphogens and other key molecules in embryonic development but also to facilitate research such as cell-cell communications, morphogenetic cell movements, as well as the construction of embryoids and organoids in vitro.

# Methods

### Reagents and tools table

| Reagent/ resource | Reference or source | Identifier or catalog number |
|---|---|---|
| **Experimental models** | | |
| AB | Dr. Sean Megason's lab (Department of Systems Biology Harvard Medical School) | N/A |
| Tg(actb2::H2B-EGFP) | Dr. Peng Xia's lab (Life Sciences Institute, Zhejiang University) | N/A |

| Reagent/ resource | Reference or source | Identifier or catalog number |
|---|---|---|
| Tg(actb2::mem-mCherry) | Dr. Sean Megason's lab (Department of Systems Biology Harvard Medical School) | N/A |
| Tg (aldh1a2: H2B-RFP) | This study | N/A |
| iBMP4 Human PSCs | Luo et al, 2025 | N/A |
| H9 human pluripotent stem cells | WiCell | WA09 |
| **Recombinant DNA** | | |
| pCS2+-bmp4 | This study | N/A |
| pCS2+-bmp2b | This study | N/A |
| **Oligonucleotides and other sequence-based reagents** | | |
| *wnt8a* probe forward primer | This study | ATGGATCATCTGTGCAGCAAG |
| *wnt8a* probe reverse primer | This study | TCATCCACGACGCGTGTGTTG |
| *ndr2* probe forward primer | This study | GGAGTGTTCGGAAAGCACAC |
| *ndr2* probe reverse primer | This study | GCATGTCCTCGTGGTGTCTC |
| *sox2* probe forward primer | This study | TTGTGGGCACAACAGGACCTAAGAG |
| *sox2* probe reverse primer | This study | AGTTGTACTGGAGCGCGCTCATGTC |
| *myod1* probe forward primer | This study | GATGGAGTTGTCGGATATCC |
| *myod1* probe reverse primer | This study | TTCTGCTGATCTGCTGAGTC |
| *gbx1* probe forward primer | This study | CATTTGGGTGATTTGAAGGC |
| *gbx1* probe reverse primer | This study | TCGCCTGATCTGTTGTTGAC |
| *fgf8a* probe forward primer | This study | CACCTTCATGGACAGCTCGG |
| *fgf8a* probe reverse primer | This study | TGTTTATCTCCAGAGAGTTC |
| *cdx4* probe forward primer | This study | CCAAGTGATATTTGGCTTGA |
| *cdx4* probe reverse primer | This study | ATATTTGAAGAGCCTCCAGC |
| *tnnt2c* probe forward primer | This study | CCTCTCTTCTCTAAACCCTCAGAAG |
| *tnnt2c* probe reverse primer | This study | AATGAGCTTGAGACTTGCATGCGAC |
| *olig2* probe forward primer | This study | CCTGCTACCGGCAATATCGACATCC |
| *olig2* probe reverse primer | This study | TCAGTCCTATAGTCGAGGGCTGAGG |
| *cldn3d* probe forward primer | This study | ACCAGGACAGAACATTCCAGCCGTC |
| *cldn3d* probe reverse primer | This study | GTGGAGTGACAATGGACAAAACC |
| *hoxc13b* probe forward primer | This study | GGCGGACACACTGATTTACGTG |
| *hoxc13b* probe reverse primer | This study | CGATTCATGTGTGAATGTCTTCTGG |
| *foxd3* probe forward primer | This study | CAGTGGGAAAAGAGACCCAAGC |

| Reagent/ resource | Reference or source | Identifier or catalog number |
|---|---|---|
| *foxd3* probe reverse primer | This study | TTTGATAATCGACGCGGTGCTC |
| **Antibodies** | | |
| pSmad1/5/9 | Cell Signaling Technology | 13820 T |
| TBXT | R&D | AF2085 |
| TBXT | Abcam | ab209665 |
| SOX2 | R&D | AF2018 |
| MEOX1 | Biorbyt | orb618083 |
| MESP2 | Biorbyt | orb669191 |
| TBX18 | Biorbyt | orb671710 |
| TBX18 | R&D | MAB63371-SP |
| Anti-Digoxigenin-AP | Roche | 11093274910 |
| **Chemicals and other reagents** | | |
| SB505124 | MedChemExpress | HY-13521 |
| IWP-L6 | MedChemExpress | HY-15825 |
| SU5402 | MedChemExpress | HY-10407 |
| DAB Substrate kit | abcam | ab64238 |
| HCR RNA-FISH probe sets | Molecular Instruments | N/A |
| HCR probe hybridization buffer | Molecular Instruments | BPH0524 |
| HCR probe wash buffer | Molecular Instruments | BPW02424 |
| Amplification buffer | Molecular Instruments | BAM02824 |
| HCR Amplifiers | Molecular Instruments | N/A |
| Dextran, Alexa Fluor™ 647; 10,000 MW, Anionic, Fixable | ThermoFisher Scientific | D22914 |
| Dextran, Fluorescein, 10,000 MW, Anionic, Lysine Fixable (Fluoro-Emerald) | ThermoFisher Scientific | D1820 |
| Doxycycline | selleck | S5159 |
| Human FGF-2 protein | R&D | 233-FB-025/CF |
| mTeSR1 | STEMCELL | 85850 |
| TeSR-E6 | STEMCELL | 05946 |
| **Other** | | |
| Single-cell RNA-seq data of human CS10 embryos | Zeng et al, 2023 | https://www.ncbi.nlm.nih.gov/geo/query/acc.cgi?acc=GSE155121 |
| Single-cell RNA-seq data of monkey CS8-11 embryos | Zhai et al, 2022 | https://www.ncbi.nlm.nih.gov/geo/query/acc.cgi?acc=GSM5466487 |

## Animals

Zebrafish wide-type AB line and Tg(*actb2:H2B-GFP*), Tg(*aldh1a2:H2B-RFP*) transgenic lines are used in this study. Embryos were dechorionated by Pronase, raised in 0.3× Danieau's buffer,

kept at 28.5 °C, and staged according to Kimmel et al (Kimmel et al, 1995). The Institutional Review Board of Zhejiang University and the Animal Ethics Committee of the School of Medicine, Zhejiang University, approved the experimental protocol. The protocol number is ZJU20220375.

## Cell lines

The WA09 human embryonic stem cells (hESCs, H9, WiCell) and H9 hESCs with doxycycline-inducible expression of *BMP4* (*iBMP4*) were used in this research (Luo et al, 2025). The iBMP4 hESCs were generated as follows: The transposon plasmid containing a tetracycline (TRE) inducible *BMP4* cassette (PB-Tre-BMP4) and constantly expressing GFP was constructed. To generate iBMP4 hESCs, the PB-Tre-BMP4 and PB-blue plasmids were co-transfected into H9 using the P3 Primary Cell 96-well Nucleofector Kit (Lonza) and the 4D Nucleofector×Unit (Lonza) according to the manufacturer's instructions. Cells after electroporation were recovered in mTeSR1 medium supplemented with ROCK inhibitor Y276932 (10 μM; Selleckchem). 24 h later, these cells were cultured in fresh mTeSR1 medium until reaching approximately 70% confluency and then were digested with Accutase (A6964, Sigma-Aldrich) at 37 °C for 5-8 min to obtain a single-cell suspension. Five hundred cells were then seeded in six-well plates and cultivated until forming colonies.

## Injection of *bmp4* mRNA into zebrafish embryos and construction of Bmp4 explants

*bmp4*-injected zebrafish embryos were generated by injecting *bmp4* (8–10 pg) mRNA into one animal pole blastomere at the 128-cell stage. The injected embryos were raised at standard conditions until reaching the desired stages. Bmp4 explants were generated by cutting off the animal pole region of *bmp4*-injected embryos at 512-cell stage in a Petri dish coated with 2% agarose gel, using a syringe needle (27 G SUNGSHIM MEDICAL CO., LTD). Then the explants were cultured in Dulbecco's modified Eagle's medium (DMEM/F-12 with 10 mM HEPES, 1× MEM with non-essential amino acids, 7 mM CaCl2, 1 mM sodium pyruvate, 50 μg/mL gentamycin, 100 μM 2-mercaptoethanol, 1× antibiotic-antimycotic, 10% serum replacement) and kept at 28.5 °C.

## Construction of human caudal-like structures

Human H9 hESCs and iBMP4 hESCs were cultured on Matrigel-coated 12-well plates in mTeSR1 medium. Cells were grown for 48–72 h until reaching an approximate 70% confluence, when they are appropriate for the formation of an embryoid body. The following induction process utilized a conditional medium (TeSR-E6 medium supplemented with 0.5% N2 and 0.5% B27 to support cell growth).

On day 0, both wild-type H9 cells and iBMP4 cells were dissociated using Accutase and subsequently seeded into 96-well U-bottom low-attachment plates. The seeding density was set at 1300 cells per well for H9 cells and 400 cells per well for iBMP4 hESCs, each in 50 μL of aggregation medium. This medium was further fortified with 10 μM of the ROCK inhibitor Y27632.

After a 24-h incubation period, the cells formed spherical aggregates in their respective wells. Subsequently, one aggregate from the H9 well and one from the iBMP4 well were manually transferred into the same well with 80 μL of initial conditional

medium to promote fusion. During days 2 and 4, the conditional medium was replaced with 80 μL induction medium (conditional medium supplemented with 75 ng/mL Doxycycline and 50 ng/mL FGF2) to induce *BMP4* expression. Starting from day 4, the induction medium was replaced with 80 μL conditional medium for continued growth and development.

## Immunostaining

Samples were fixed by 4% paraformaldehyde at 4 °C overnight. Incubate the samples with 150 mM Tris-HCl (PH 9.0) at 70 °C for 15 min to retrieve antigen. Treat samples with blocking buffer (10% FBS, 0.1% Triton X-100, 1× PBS) for 2 h, then incubate overnight at 4 °C with antibody diluted in blocking buffer. Anti-pSmad1/5/9 antibody (13820, Cell Signaling Technologies, 1:800), anti-SOX2 antibody (AF2018, R&D, 1:50), anti-MEOX1 antibody (orb618083, Biorbyt, 1:50), anti-MESP2 antibody (orb669191, Biorbyt, 1:50) and anti-TBX18 antibody (MAB63371, R&D, 1:50) was used for the immunostaining experiments. Samples were washed six times with PBSTr (0.1% Triton X-100, 1× PBS) to remove the primary antibody. Treat samples with blocking buffer for 1 h, then with secondary antibody diluted in blocking buffer (1:500) until desired signal intensity is achieved. DAPI co-staining was achieved by adding 0.1 μg/mL DAPI with secondary antibody. Wash samples six times with PBSTr to remove secondary antibody.

## Live imaging

Samples were transferred to a glass-bottom culture dish (Cellvis, D35-10-0-N), properly oriented in one drop of 0.1% low-melting agarose, and covered with the culture medium. The samples were then photographed using OLYMPUS CSU-W1 confocal microscope every 3 min.

## Whole mount in situ hybridization (WISH)

Probes were labeled with digoxigenin-11-UTP (Roche Diagnostics), and the substrate was NBT/BCIP. Samples were fixed with 4% (w/v) paraformaldehyde in DEPC-treated PBS at 4 °C overnight. Dehydrate the samples with 100% methanol at −20 °C overnight. Rehydrate samples by washing with successive dilutions of Methanol. Hybridize the samples with a hybridization mix containing 30–50 ng probe at 70 °C overnight. Gradually exchange the hybridization mix to 2× SSC and then incubate in 0.2× SSC at 70 °C for two times. Replace 0.2× SSC to PBST by successive dilutions of 0.2× SSC. Incubate samples in blocking buffer for 2 h and then in blocking buffer containing 1/5000 AP-anti-DIG antibody at 4 °C overnight. Remove the antibody by extensive washes of PBST. Incubate the samples in 1 mL staining solution (alkaline tris buffer, 225 μg/mL NBT, 175 μg/mL BCIP) until desired signal density was achieved. Block the labeling by washing with PBST (0.1% Tween-20, 1× PBS) and store the samples in glycerol.

## In situ hybridization chain reaction (HCR)

HCR staining was performed as described previously (Choi et al, 2018). Samples were fixed with 4% (w/v) paraformaldehyde in DEPC-treated PBS at 4 °C overnight. Proteinase K treatment is recommended before incubation of probe sets. Samples were incubated with 2 pmol of each HCR probe set in 100 μL probe hybridization buffer at 37 °C for

12–16 h. Wash with probe wash buffer at 37 °C for four times and with 5× SSCT at room temperature for two times to stop hybridization. Pre-amplify samples with amplification buffer at room temperature for 30 min, then incubate samples with 30 pmol snap-cooled hairpin h1 and h2 for 4 h at room temperature in a dark drawer. Stop amplification by washing with 5× SSCT (DAPI co-staining was achieved by adding 0.1 μg/mL DAPI) several times. Fluorescent hairpins, buffers, and DNA probes were purchased from Molecular Technologies. Fluorescent signals were detected by OLYMPUS CSU-W1 confocal microscope.

## Xenograft assay

We used transgenic H9 hESCs with doxycycline-inducible expression of human *BMP4* (hereafter referred to as iBMP4 cells) for the xenograft experiment. When iBMP4 cells were around 80% confluent, cells were dissociated using Accutase, washed in DMEM/F12, centrifuged at 180× *g* for 3 min, and resuspended in mTeSR1 supplemented with 10 μM ROCK inhibitor Y27632. Cell numbers were determined using an automated cell counter, and a calculated volume of single-cell suspension containing about $2 \times 10^5$ cells were added to a Matrigel pre-treated well of a 12-well plate. 24 h after single-cell seeding, the culture medium was replaced with TeSR-E6 containing 1 μg/mL Doxycycline and 50 ng/mL recombinant Human FGF2, for a 24-h treatment. Following treatment with Doxycycline, *iBMP4* cells were digested into single cells as described above and resuspended in 50 μL TeSR-E6. About 20–50 resuspended *iBMP4* cells were then transplanted to the animal pole of 4 hpf zebrafish embryos. The zebrafish embryos with induced secondary axis at 24 hpf were selected for imaging analysis and WISH.

## Processing of single-cell RNA-seq data

Raw sequencing reads were processed using the CellRanger (Zheng et al, 2017) pipeline (10x Genomics, version 3.0.2) with default parameters. The resulting filtered expression matrix was utilized for downstream analysis, with the zebrafish reference genome GRCz11 or the human reference genome GRCh38. Quality control and cell clustering were performed using the Seurat (Hao et al, 2021) package (version 4.0.3). Initially, a Seurat object was created with parameters min.cells = 3 and min.features = 200 by *CreateSeuratObject* function. Subsequently, the *subset* function was employed to remove cells based on appropriate parameters for each sample. Normalization of gene expression for each cell was performed using the *NormalizeData* function, and log-transformed expression counts were obtained. Highly variable genes were detected using the *FindVariableGenes* function, and the top 2000 genes with the highest variability were selected for downstream analysis. The data was scaled by running the *ScaleData* function. Linear dimensional reduction was performed by *RunPCA* function. Principal components were determined using the *JackStraw* and *ScoreJackStraw* functions. Clustering analysis was conducted using the *FindNeighbors* and *FindClusters* functions. The resolution parameter in the *FindClusters* function was set between 1 and 2. Differentially expressed markers for each cluster were identified using the *FindAllMarkers* function with default parameters. Finally, the cell clusters were annotated based on Zebrafish Information Network (Bmp4 explant data, https://www.zfin.org/) or other published

studies (Farrell et al, 2018; Wagner et al, 2018; Zeng et al, 2023; Zhai et al, 2022).

## Integration of single-cell RNA-seq data

To integrate the Seurat objects of the Bmp4 explant of different timepoints or with wild-type embryonic dataset, each Seurat object was normalized by *NormalizeData* function. And then, the *FindVariableFeatures* function was used to detect the variable genes in each sample, the nfeatures was set to 2000. Next, select features that are repeatedly variable across all datasets that need to be integrated by running *SelectIntegrationFeatures* function. The identified features were used to run *ScaleData* and *RunPCA* for each object. Then, *FindIntegrationAnchors* function was used to determine the "anchors" of these objects. Finally, the integrated dataset was obtained by running *IntegrateData* function. After that, the default assay was set to 'integrated', and the standard workflow for visualization and clustering was executed, including the following steps: *ScaleData*, *RunPCA*, *RunUMAP*, *FindNeighbors*, and *FindClusters*. The identified cell clusters in the integrated Seurat object were visualized on UMAP (Becht and McInnes, 2018) plot.

## Construction of a single-cell trajectory for Bmp4 explants

Three different methods were employed to conduct single-cell RNA-seq trajectory analysis. Firstly, URD (Farrell et al, 2018) was employed to reconstruct the developmental trajectories of Bmp4 explant cells, spanning the time range from 6 to 24 hpf. The analysis was conducted following the instructions of the previous study (Farrell et al, 2018). In this analysis, early time point cells (6 hpf) were designated as the root, while late time point cells (24 hpf) were assigned as the tips of the trajectory. The tip clusters were annotated based on the differentially expressed markers within each cluster. To construct the branching tree, we utilized the *buildTree* function in URD with the following parameters: save.all.breakpoint.info = T, divergence.method = "preference", cells.per.pseudotime.bin = 50, bins.per.pseudotime.window = 5, and p.thresh = 0.01. The branchpoint preference plot for TMP and PSM was generated using the *branchpointPreferenceLayout* function from URD. Subsequently, stage, pseudotime information, and gene expression were visualized on the branchpoint using the *plotBranchpoint* function. For identifying the differential expression (DE) genes along the trajectories, we utilized the *geneCascadeProcess* function with a moving window average (moving.window = 5, cells.per.window = 15). This smoothing process enabled us to obtain a more refined representation of gene expression dynamics. The smoothed expression of each gene was fitted using an impulse model.

Next, NMP cells, as well as neural derivatives and mesodermal derivatives, were selected for cell differentiation analysis using monocle 3 (Cao et al, 2019) with default parameters. This analysis aimed to demonstrate the bipotent differentiation trajectory of NMP cells. Similarly, cells from the TMP, PSM, and somite populations were selected for analysis using monocle 3. This analysis allowed for the identification of different gene modules along the TMP-PSM trajectory and the PSM-somite trajectory. *graph_test* function was used to identify DE genes along the trajectories and *find_gene_modules* function was employed to detect related gene modules that regulate cell differentiation along a specific trajectory.

Lastly, a K-Nearest Neighbors (KNN) algorithm-based method was employed to construct a single-cell developmental trajectory.

The scRNA-seq data from adjacent time points were integrated using Harmony (Korsunsky et al, 2019) with default settings. Then the KNN algorithm was used to select the 10 nearest neighbors in the earlier time point for each spot of the latter time point. The most frequent clusters of the 10 nearest spots were regarded as the source states that the latter spots would develop from. If two or more clusters had the same frequencies, the cluster of the nearest spot was used. The procedure was repeated on each pair of adjacent time points from 6 to 24 hpf, and a connection was only reserved when there were more than 20% cells of the same cluster in the latter time point developing from a cluster in the earlier time point.

## Comparison with in vivo datasets

The scRNA-seq datasets of Human CS10 embryos (Zeng et al, 2023) and Monkey CS8-CS11 embryos (Zhai et al, 2022) were used as references for comparison. The gene expression of each cluster in the human embryoids and the reference datasets was aggregated into a pseudo-bulk. The marker genes of each cluster were selected, and a correlation test was performed using the marker gene expression in the pseudo bulk to assess the cell type similarities between the human embryoids and the references. The correlations of cell types between the human embryoids and the references were calculated as Pearson correlation coefficients and visualized using a heatmap.

## Gene set enrichment analysis

The genes from gene modules along the TMP-PSM trajectory or PSM-somite trajectory were used to perform gene set enrichment analysis. The gene list was uploaded to the Metascape (Zhou et al, 2019) website (http://metascape.org), and the results were downloaded following the execution of Express Analysis.

## Tracking cell movements in Bmp4 explants

Imaris software (version 9.7) was utilized to track the cell movements in Bmp4 explants, or the embryos injected with *bmp4* mRNA. The Spot plugin from Imaris was employed to label the cells and analyze cell trajectories. Spot detection diameter was set at 6–15 µm (Figs. EV5G,H and 4B, 13.5 µm; EV5J,K, 6.5 µm). Subsequently, candidate spots were manually selected, filtering out spots that met the appropriate quality threshold, for further analysis. The Autoregressive Motion algorithm was employed to track the movement of each cell, with Max Gap Site set to 3. The movement dataset, consisting of cell coordinates at all time points, was extracted from Imaris and analyzed in R. It was observed that all cells exhibited migration in the same direction. This migration pattern, referred to as the observed trajectory (OT), was primarily attributed to the elongation of the entire explant. To analyze the cell trajectory specifically within the Bmp4 explant, termed the migration trajectory (MT), the elongation trajectory (ET) was first derived from the OT by studying the movements of EVL cells (designated as anchor cells) that were presumed to remain stationary within the bmp explant. Three anchor cells (EVL cells) were selected, and their central coordinate trajectory was defined as ET. Finally, the MT of the selected cells was obtained by subtracting the ET from the OT. The trajectories were visualized by R package plot3D (https://CRAN.R-project.org/package=plot3D).

## Quantification of cell movement directions and the changes of neighbor cells

We first identified internalizing cells using the SPOT module of Imaris and obtained the 3D coordinates of each cell. To calculate the direction of cell movement across different timepoints, we projected the positional changes of cells onto a two-dimensional coordinate system, thereby determining the directional movement of each cell between timepoints. For analyzing changes in neighboring cells, we first defined neighboring cells as those within a 15 μm radius of a given cell, then identified the neighbors for each cell at each timepoint. For any two selected timepoints, each cell has its own set of neighboring cells. We defined a "rate" as the number of shared neighbors between the two timepoints divided by the number of neighbors at the initial timepoint. Based on this defined rate, we quantified the changes in neighboring cells for all cells between any two timepoints.

## Data availability

The data generated in this study are deposited in the Gene Expression Omnibus (GEO). The single-cell RNA sequencing data are under the accession number GSE270989. Any custom code and data are available from the authors upon request. All analyses are based on previously published code and software.

The source data of this paper are collected in the following database record: biostudies:S-SCDT-10_1038-S44318-025-00643-6.

## Peer review information

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

## Acknowledgements

We thank Dr. Bernard Thisse (1959-2021) and Dr. Christine Thisse at the University of Virginia for their helpful comments and suggestions. We also thank Dr. Min-Xin Guan, Dr. Xiao-Hang Yang, Dr. Hong-Qing Liang and the members of the Laboratory of Development and Organogenesis (LDO) at Zhejiang University for helpful suggestions and discussions. We thank Ying-Niang Li from the zebrafish core facility at Zhejiang University School of Medicine for her technical support. This work was supported by grants from the National Scientific Foundation of China (31900576, 32300677, 32300688) and the Chinese National Key Research and Development Project (2022YFA1103102, 2019YFA0802402).

## Author contributions

**Yan-Yi Xing**: Formal analysis; Funding acquisition; Investigation; Writing—original draft. **Ying Huang**: Formal analysis; Investigation. **Tao Cheng**: Data curation; Formal analysis; Supervision; Funding acquisition; Investigation; Writing—review and editing. **Yi-Meng Tian**: Investigation. **Yang Dong**: Funding acquisition; Investigation. **Zi-Xin Jin**: Investigation. **Hai-Rong Pu**: Investigation. **Tao Luo**: Investigation. **Xiang Liu**: Investigation. **Hao-Nan Shen**: Investigation. **Jing Mo**: Investigation. **Jun Ma**: Validation. **Jun-Feng Ji**: Validation. **Peng-Fei Xu**: Conceptualization; Supervision; Funding acquisition; Validation; Project administration; Writing—review and editing.

Source data underlying figure panels in this paper may have individual authorship assigned. Where available, figure panel/source data authorship is listed in the following database record: biostudies:S-SCDT-10_1038-S44318-025-00643-6.

## Disclosure and competing interests statement

The authors declare no competing interests.

# Expanded View Figures

**Figure EV1.  Characterization of *bmp4* injected zebrafish embryos and explants, related to Fig. 1.**

(**A**) Immunofluorescence staining for phosphorylated Smad1/5/9 in embryos injected with *bmp4 + RFP* mRNA (left) and *RFP* mRNA alone (right) at 6 hpf. The injection site was determined by red fluorescence. Signal quantifications along the ventro-dorsal axis are displayed beside the merged images. (**B**) Immunofluorescence staining for pSmad1/5/9 in explants injected with *bmp4 + RFP* mRNA (left) and *RFP* mRNA alone (right) at 10 hpf. Signal quantifications along the anterior–posterior axis are shown next to the merged images. Clones of *bmp4* injected blastomere would stay at the injected end when they differentiated into enveloping layer cells. But when they differentiate into deep cells, they would involute and migrate anteriorly. Signal intensities of pSmad1/5/9 and RFP were normalized first by the DAPI signal and then by their mean signal intensity. (**C**) Whole mount in situ hybridization (WISH) of *id1* (left) HCR co-staining of *id1*, *tbxta* and *tbx6* (middle) in Bmp4 explants at 14-18 hpf, with injection sites oriented to the right. Signal quantifications along the white dashed line are presented to the right. (**D**) WISH of *eve1, fgf8a, sox32, cdx4, chrd* in Bmp4 explants at 6 hpf, with injection sites oriented to the right. (**E**) WISH of *tbxta, cdx4* in Bmp4 explants treated with Nodal inhibitor (SB505124, 50 μM), Wnt inhibitor (IWP-L6,25 μM) or FGF inhibitor (SU5402, 25 μM) at 6 hpf, with injection sites oriented to the right. Representative images of treated or untreated Bmp4 explants at 24 hpf are displayed. (**F**) WISH of *wnt8a, ndr2*, and *tbxta* in uninjected, 20 pg *bmp2b* mRNA injected, 0.5 pg *bmp4* mRNA injected explants or embryos at 6 hpf. (**G**) Expression patterns of *fgf8a, sox17*, and *eve1* revealed by WISH in *bmp4*-injected embryos at 6 hpf. Views from the animal pole of embryos are displayed, with descendants of the *bmp4*-injected blastomere labeled by DAB staining of GFP (left and middle images). (**H**) WISH of *sox32* in uninjected, 8 pg or 20 pg *bmp4* mRNA injected explants at 10 hpf. (**I**) Expression pattern of *tbxta* revealed by WISH in *bmp4*-injected embryos at 12 hpf. (**J**) WISH for *tbxta, egr2b, sox19a, olig2, foxd3, shha, tnnt2c, myod1, cldn3d, hoxc13b* in *bmp4*-injected embryos 24 hpf. (**K**) WISH for *egr2b, gata6, shha, her1, fn1b*, and *foxd3* in Bmp4 explants at 24 hpf. Each experiment was performed for at least three independent replicates (technical replicates). Scale bars: 100 μm.

▶

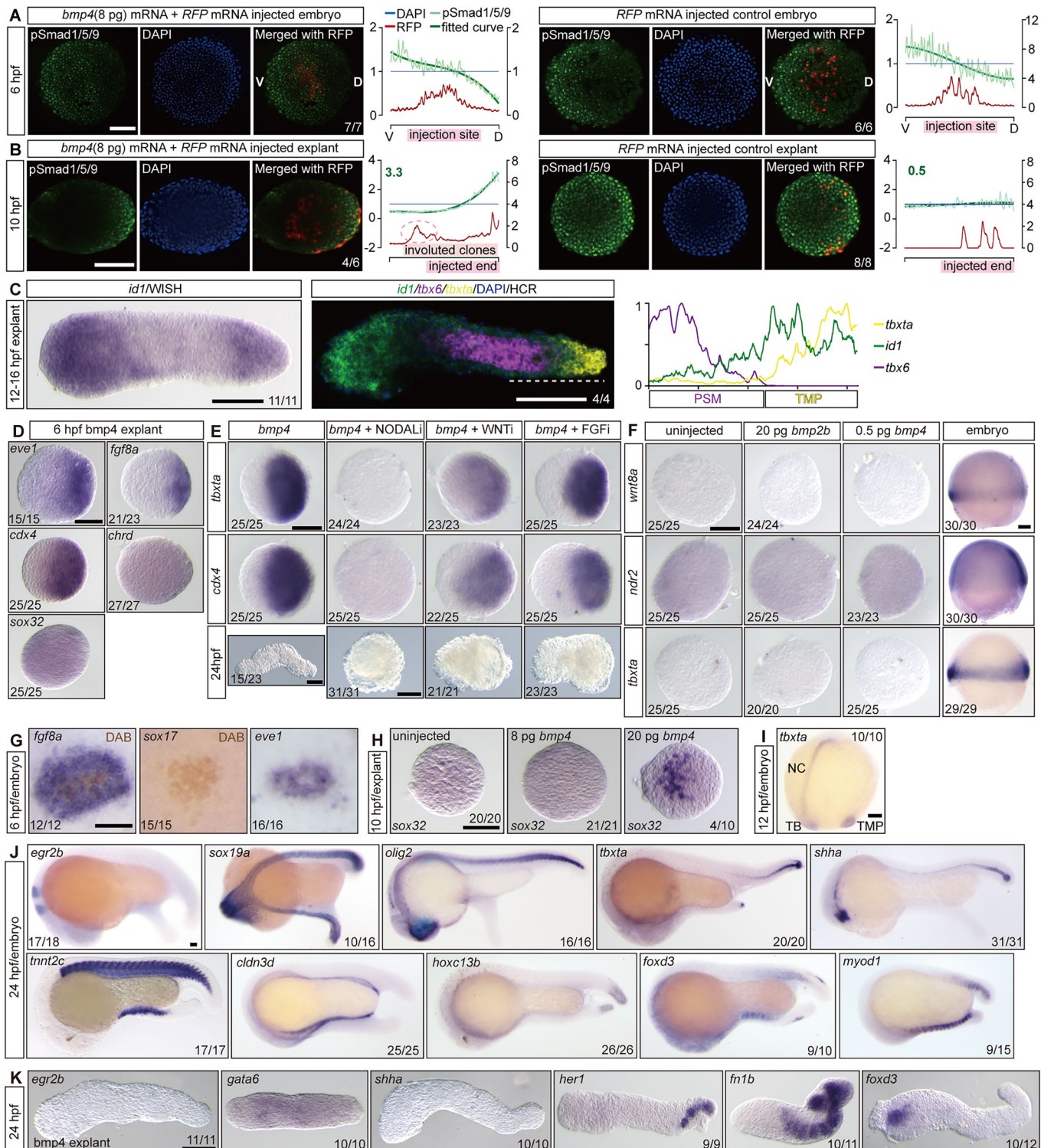

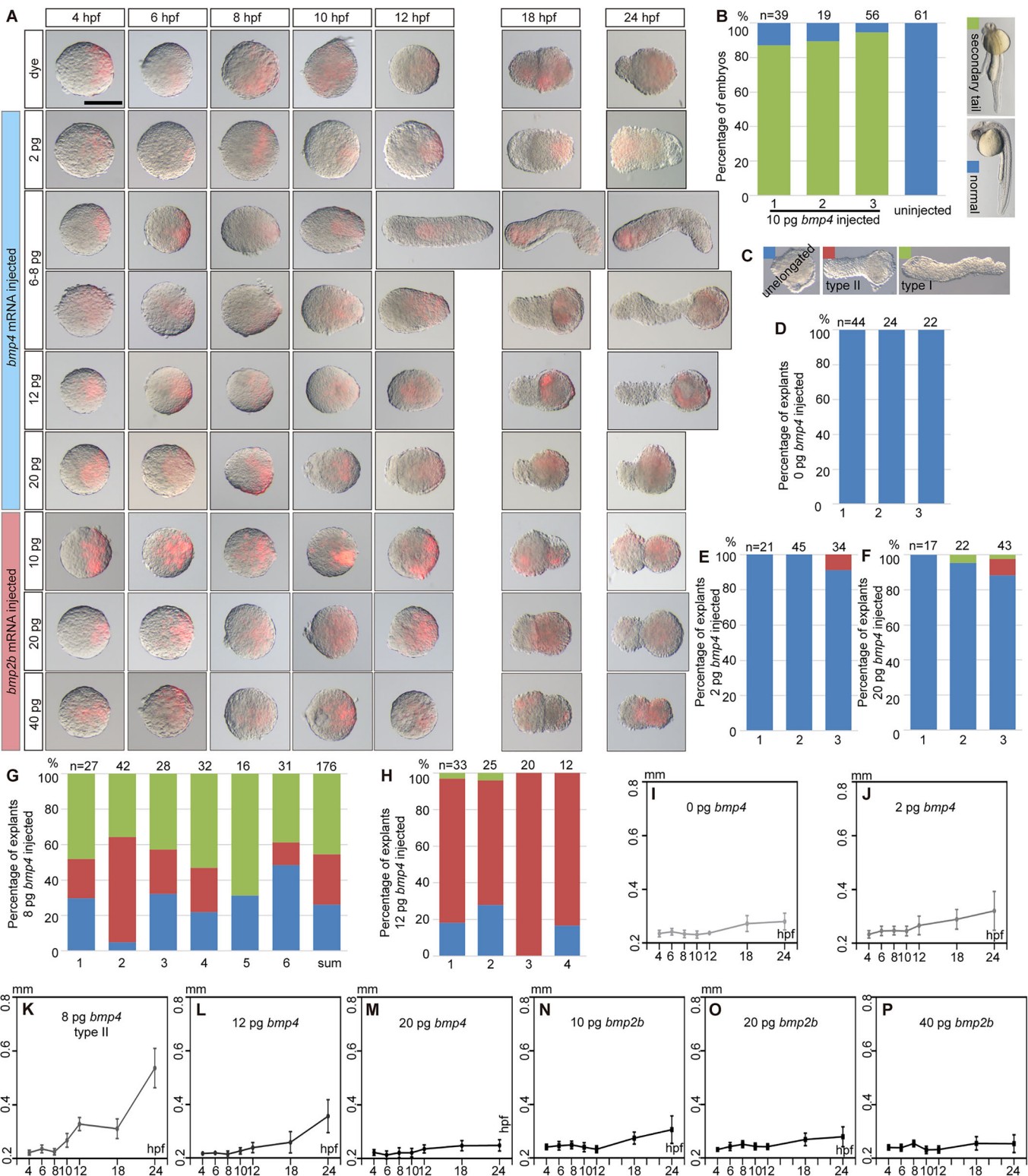

**Figure EV2.    Titration of optimal *bmp4* dosage for inducing caudal structures in zebrafish explants, related to Fig. 1.**

(A) Representative images of explants induced by different doses of *bmp4* or *bmp2b* at indicated stages. The injection sites are oriented towards the right. (B) Statistical analysis of *bmp4* injected embryos with or without secondary tail. Analysis based on three independent replicated experiments. (C) Representative images showing the morphologies of *bmp4* injected explants at 24 hpf: unelongated, elongated but without somites (type II) and elongated with somites (type I) are shown. (D–H) Statistical analysis of different morphological outcomes (as defined in (C)) for explants injected with 0 pg (D), 2 pg (E), 8 pg (F, only type II explants analyzed), 12 pg (G) and 20 pg (H) *bmp4* mRNA. Numbers on the top of each column indicate the total number of explants analyzed. At least three independent experiments were done for each dose of *bmp4*. (I–P) Statistical analysis detailing the length measurements over time for explants injected with 0 pg (I), 2 pg (J), 8 pg (K, type II without somites), 12 pg (L), 20 pg (M) *bmp4* mRNA or 10 pg (N), 20 pg (O), 40 pg (P) *bmp2b* mRNA. Each experiment was performed for at least three independent replicates (technical replicates). Scale bars: 200 μm.

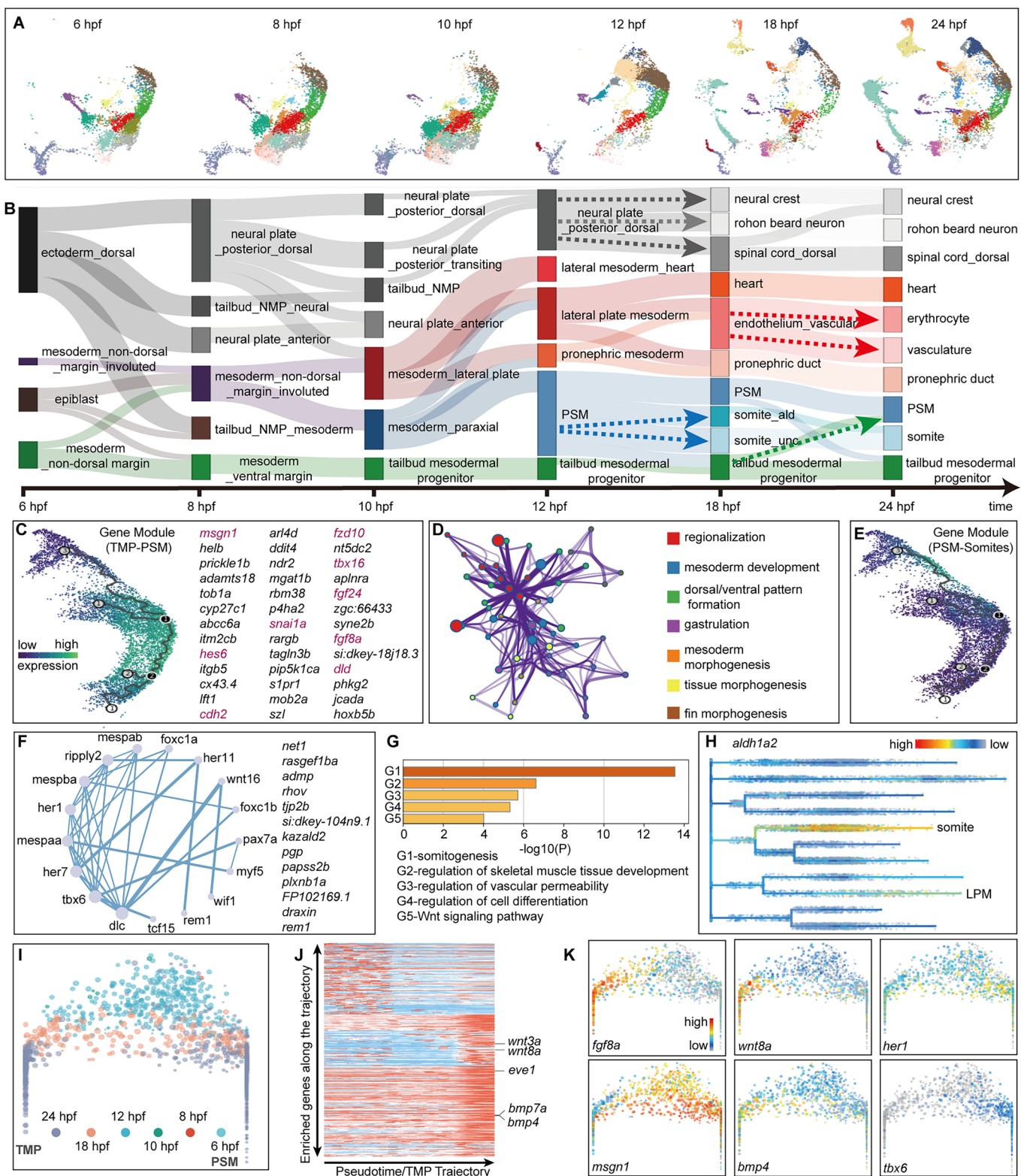

◀  **Figure EV3.   Single-cell RNA sequencing analysis of Bmp4 explant, related to Fig. 2.**

(**A**) UMAP plot showing the cell clusters in Bmp4 explants at 6 developmental stages (6 hpf, 8 hpf, 10 hpf, 12 hpf, 18 hpf and 24 hpf). Cells are colored by different cell types. Schematic of Bmp4 explants at indicated developmental stage are shown on the top. (**B**) Sankey plot showing inferred relationships between cell states across the development of Bmp4 explant. Gray, blue, green and red arrows highlight the differentiation of neural plate to neural crest/neural tube/Rohan beard neuron, PSM to somites, TMP to PSM and vascular endothelium to vasculature/erythrocyte, respectively. (**C**) The single-cell trajectory of TMP, PSM and Somites was analyzed using Monocle 3 and visualized by UMAP. Cells are color-coded based on the expression levels of gene module of TMP-PSM, which represents the co-upregulated genes in the trajectory from TMP to PSM. The genes in this module are listed on the right, and genes highlighted in purple indicate that they have been reported to be relevant to somite formation. (**D**) Network layout displays the enriched Gene Ontology (GO) terms for gene module of TMP-PSM (**C**). The size of each node is scaled according to the number of input genes associated with that term. The color of the nodes corresponds to their cluster identity, which is labeled on the right side. The terms are connected by edges, where the thickness of the edge indicates the similarity score (score > 0.3) between the terms. (**E**) The single-cell trajectory of TMP, PSM and Somites. Cells are color-coded based on the expression levels of gene module of PSM-Somites. (**F**) The genes in the gene module of PSM-Somites. Several genes reported to show interactions are displayed in a circular network layout (left). (**G**) Heatmap showing the enriched GO terms using the genes from (**F**). (**H**) URD trajectory of Bmp4 explant showing the expression of *aldh1a2*, somite and LPM were labeled. (**I**) URD branchpoint plots of TMP and PSM development are generated using scRNA-seq data from Bmp4 explants. The plots depict pseudotime on the y-axis and the random walk visitation preference from TMP to PSM domains on the x-axis. Cells are colored by developmental stages. (**J**) Heatmap showing the expression of enriched genes along the TMP trajectory (pseudotime, x-axis). Several genes related to the TMP, including *wnt3a*, *wnt8a*, *eve1*, *bmp7a*, and *bmp4*, are labeled on the right. (**K**) The expressions of *fgf8a*, *wnt8a*, *msgn1*, *bmp4*, *her1* and *tbx6* are showed on URD branchpoint plots of TMP and PSM.

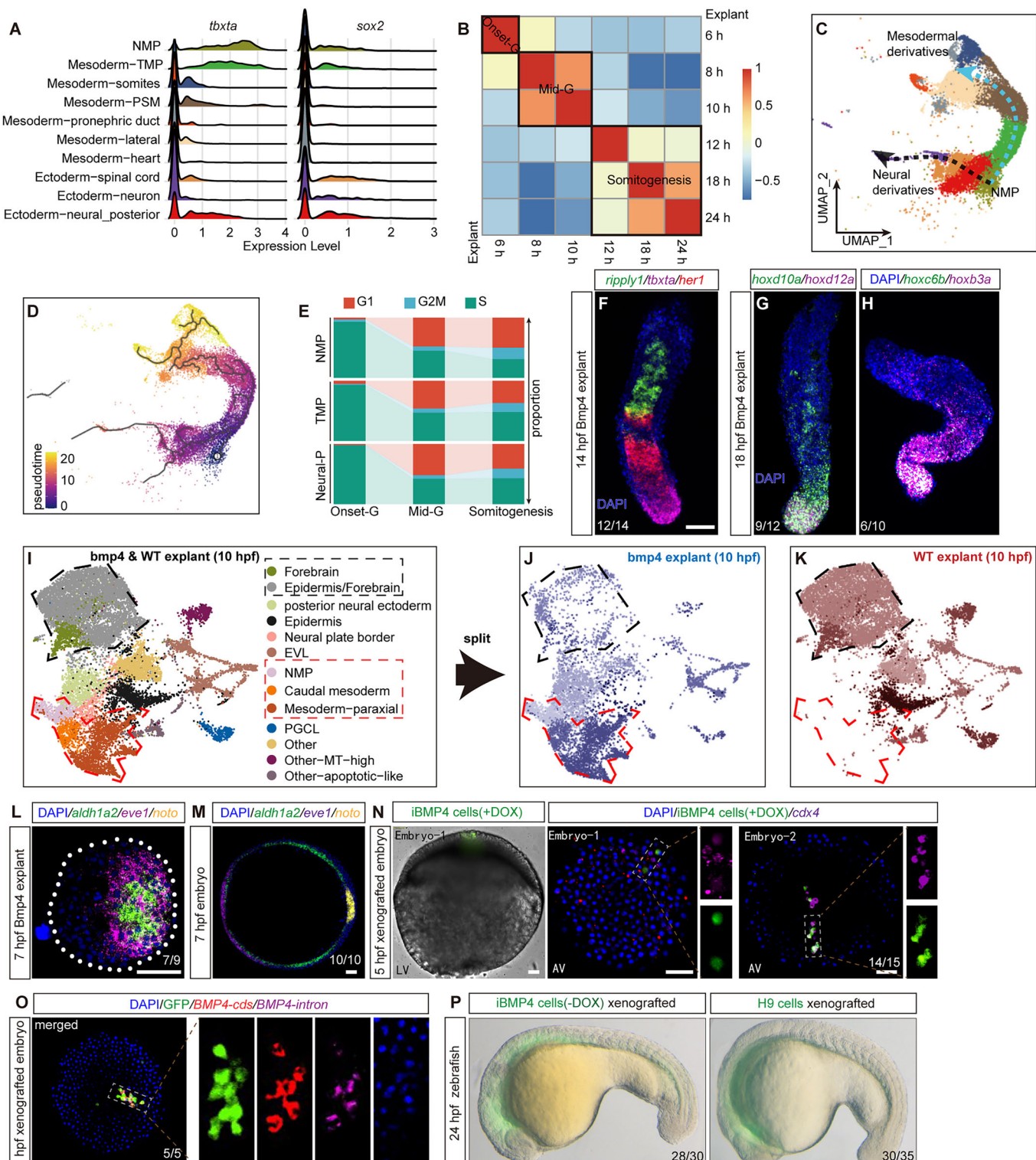

◀ **Figure EV4.  Analyzing NMP-like development in Bmp4 explant, related to Fig. 2.**

(**A**) Ridge plot showing the expression of *tbxta* and *sox2* in each cell cluster (including NMP-like, mesodermal derivatives and neural derivatives). (**B**) Heatmap showing the transcriptional similarity between Bmp4 explants at 6 developmental timepoints. We defined three developmental stages: Onset-G (6 hpf), Mid-G (8–10 hpf) and somitogenesis (12–24 hpf). (**C**) UMAP plot displaying cell clusters related to NMP-like, mesodermal derivatives, and neural derivatives. The blue dashed line with an arrow indicates the potential differentiation trajectory from NMP-like to mesodermal derivatives. The black dashed line with an arrow indicates potential differentiation trajectory from NMP-like to neural derivatives. (**D**) UMAP plot showing the inferred trajectories from NMP-like cells to mesodermal derivatives or neural derivatives. Cells are colored by pseudotime. (**E**) Ribbon plot showing the proportions of the computational cell cycle stages (G1, G2M and S) in NMP-like, TMP and Neural-P cells during development. (**F–H**) HCR co-staining of *ripply1*, *tbxta* and *her1* (**F**), *hoxd10a* and *hoxd12a* (**G**), *hoxc6b* and *hoxb3a* (**H**) in Bmp4 explants at 14 hpf (**F**),18 hpf (**G**, **H**). DAPI was co-stained (**F–H**). (**I**) UMAP plots showing integrated single-cell RNA sequencing (scRNA-seq) datasets of Bmp4 explants and WT uninjected explants at 10 hpf (**I**). (**J**, **K**) UMAP plots showing scRNA-seq datasets of Bmp4 explants (**J**) or WT uninjected explants (**K**) separated from I. Red and black dashed circle indicate caudal cell fates and anterior neural cell fates respectively. (**L**, **M**) HCR co-staining of *aldh1a2*, *eve1* and *noto* in Bmp4 explants (**L**) and zebrafish embryos (**M**) at 7 hpf. (**N**) HCR staining of *cdx4* in xenografted zebrafish embryos at 5 hpf. Green fluorescence indicates grafted iBMP4-induced human PSCs. Representative merged images depicting the location of grafted iBMP4-induced human PSCs in a zebrafish embryo at 5 hpf. The left panel (lateral view) shows the graft site, while the middle panel (animal pole view) of the same embryo reveals the expression of *cdx4* (magenta) in relation to the grafted human cells (green). (**O**) HCR staining of BMP4-cds and BMP4-intron in xenografted zebrafish embryos at 6 hpf. Grafted human cells are labeled with GFP. (**P**) Merged images showing untreated iBMP4 cells grafted (left, labeled with GFP) or H9 grafted (right, labeled with RFP) zebrafish larvae at 24 hpf. Each experiment was performed for at least three independent replicates (technical replicates). Scale bar: 50 μm.

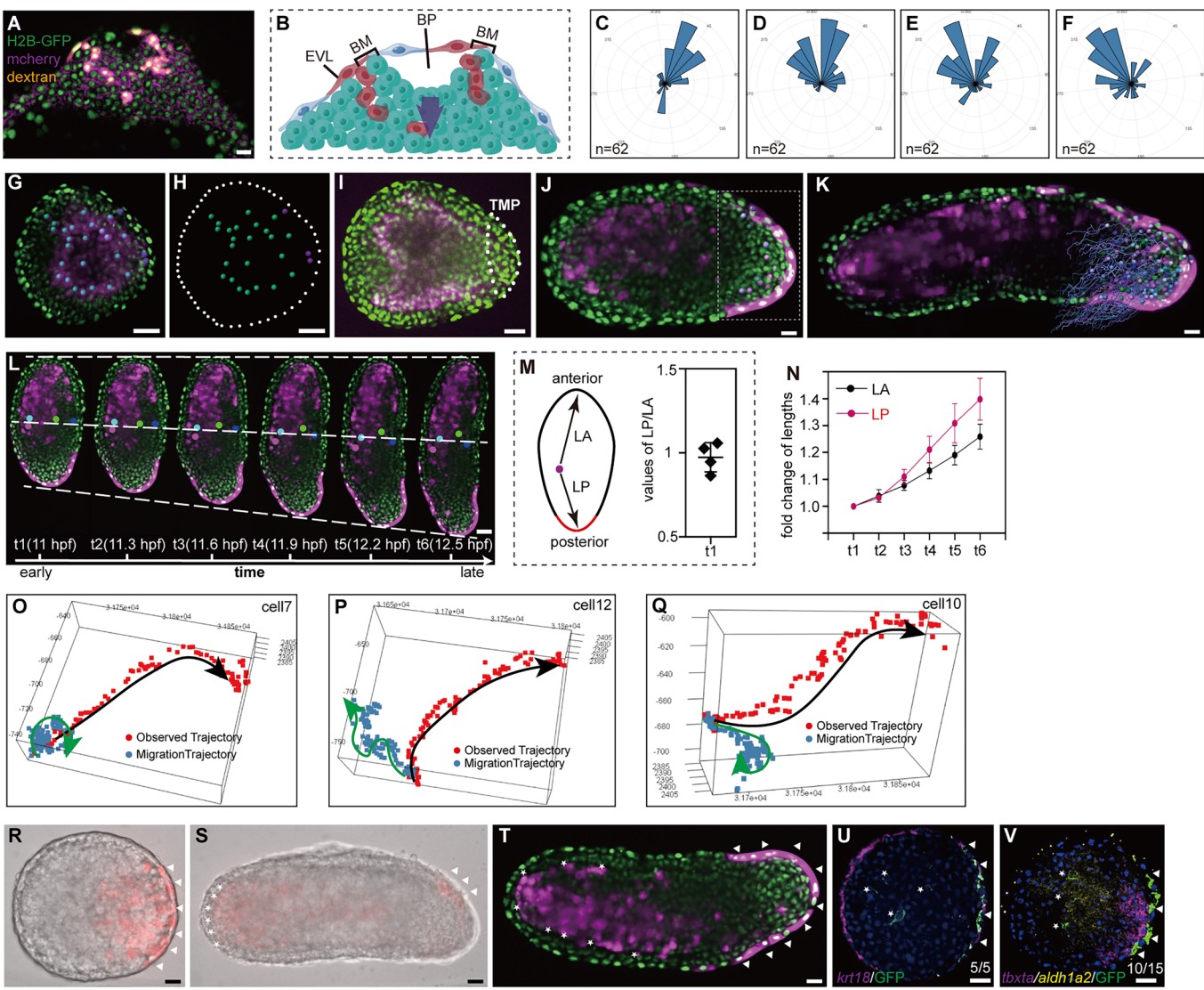

**Figure EV5. Cell movements in *bmp4* injected embryos and explants, related to Fig. 4.**

(A) Confocal image of the protrusion in a *bmp4* injected embryo at 8 hpf. Cell nuclei, cell membrane and descendants of *bmp4* mRNA injected blastomere are labeled by H2B-GFP, mcherry and dextran fluorescein(purple), respectively. (B) Schematic summary of structures near *bmp4* injection site in the embryo, based on (A). (C–F) Rose-plots show the movement direction of internalizing cells at different timepoints, from early (C) to late (F) timepoints. (G–I) Representative images of gastrulation cell movements in a Bmp4 explant generated from Tg (*aldh1a2: H2B-RFP*) transgenic line at the middle (G, H) and end (I) of gastrulation, according to Movies EV6 and EV7. Cells are colored based on the conditions of their movement, as described in Fig. 4B'. Dashed lines outline the boundaries of the embryonic blastoderm (H) and explant (I). (H) Cell movement trajectories in the posterior part of Bmp4 explant (dashed rectangle region) are analyzed at the beginning of segmentation stage. (I) Cell movement trajectories of all 13 tracked cells. (J–L) Comparison of observed raw trajectories (red spots and black arrows) to the computed migration trajectories (green spots and green arrows). These trajectories illustrate cell movements within the Bmp4 explant, with details on the trajectory analysis strategy outlined in Fig. 4H. Scale bars: 20 μm. (M) Measurement of distances from one spot to the anterior or posterior end of the explant, denoted as LA or PA respectively. It is noteworthy that the blastomere injected with bmp4 mRNA contributes both to EVL cells that stay at the injection site and deep cells that migrate to the anterior later. So, we use the red fluorescence labeled EVL cells to determine the injection site and posterior end. The values of LP divided by LA are all around 1, indicating central positioning of the tracked spots. (N) Quantification of the elongation of the anterior and posterior halves of Bmp4 explant over time, with lengths of LP/LA at times t2-6 normalized to those at t1. (O–Q) The observed raw trajectories (red spots and black arrows) compared to the migration trajectories (cell movements within Bmp4 explant, green spots and green arrows) obtained by the strategy described (details see methods) in Fig. 4H of cell 7 (O), 12 (P), 10 (Q). (R, S) Merged images showing the morphology and distributions of *bmp4* injected cell clones in Bmp4 explant at 5 hpf (R) and 12 hpf (S). White triangles indicate the EVL cells. (T) Merged image showing the morphology and *bmp4* injected cell clones in Bmp4 explant at 12 hpf. GFP signal: nucleus (H2B-GFP), RFP signal: *bmp4* injected clones. White triangles indicate the EVL cells. (U, V) HCR staining of *krt18* (U), *tbxta&aldh1a2* (V) in Bmp4 explants at 9 hpf. GFP signals indicate the *bmp4* injected clones. White triangles indicate the EVL cells, white stars indicate the internalized *bmp4* injected clones. Each experiment was performed for at least three independent replicates (technical replicates). Scale bar: 50 μm (G, H, L), 20 μm (A, I, J, K, R–V).

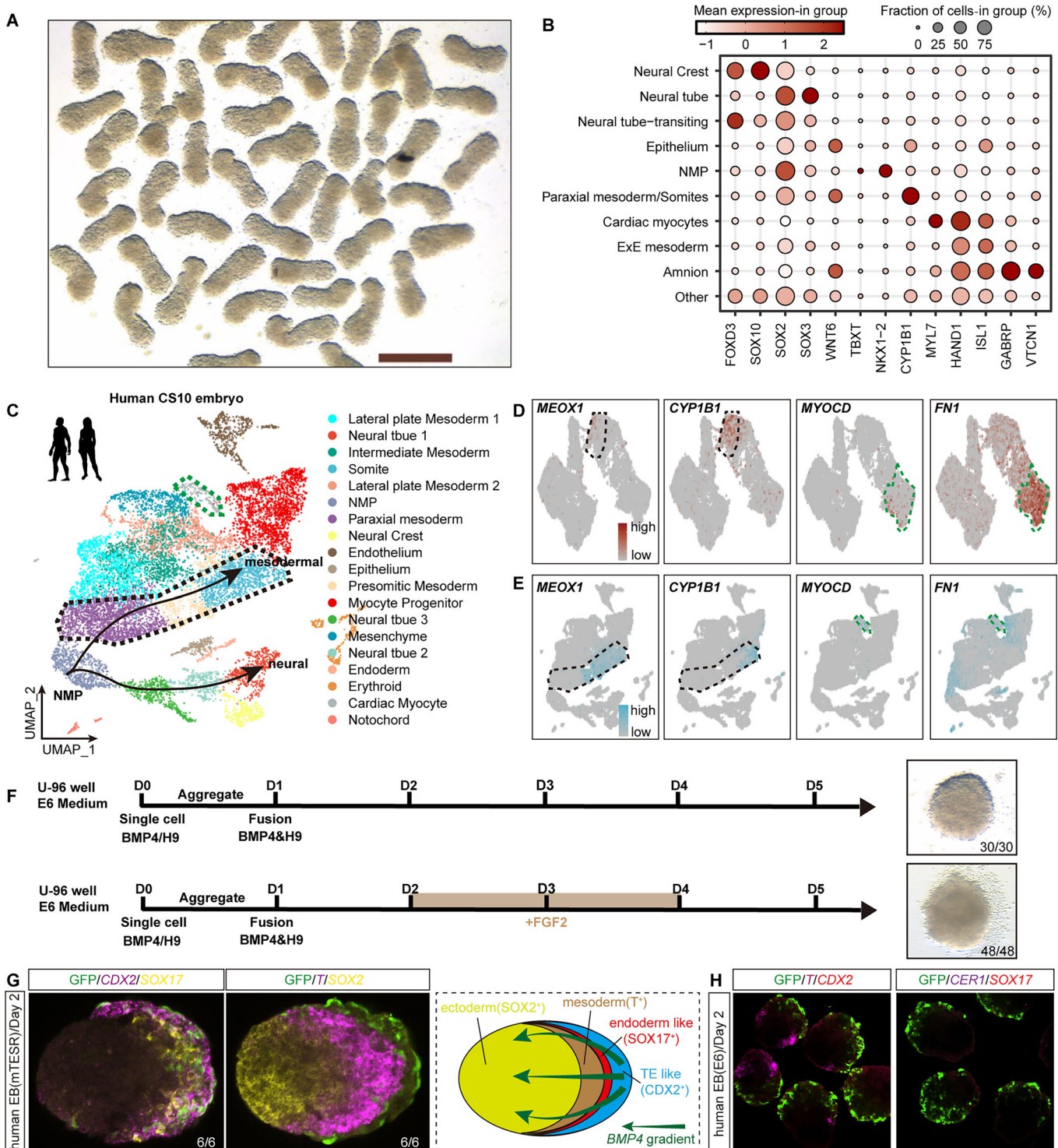

◄   **Figure EV6.   scRNA-seq analysis of human VCLS at day 5, related to Fig. 6.**

(A) Bright field image of VCLS at day 5, illustrating typical morphology. (B) Dot plot showing the average expression levels of the specified marker genes in each cell cluster. (C) UMAP plot showing the cell clusters in human CS10 embryo, with cells colored by different cell types. Black lines with arrows highlight the potential differentiation trajectories from NMP-like cells to neural or mesodermal lineages. (D) UMAP plots showing the expression patterns of *MEOX1, CYP1B1, MYOCD* and *FN1* in VCLS at day 5. These genes are key markers for paraxial mesoderm/somites and cardiac myocytes, which are highlighted with black and green circles, respectively. (E) Comparative UMAP plots showing the expression of *MEOX1, CYP1B1, MYOCD* and *FN1* in human CS10 embryo. (F) Bright field images of control human embryoid at day 5 induced without dox and FGF2 (upper panel) or only with FGF2 (lower panel). (G) HCR co-staining of *CDX2/SOX17*(left) and *T/SOX2*(middle) in EBs cultured in mTeSR1 at day 2. GFP indicates the location of *BMP4*-producing cells. The pattern of different germ layers is summarized to the right. (H) HCR co-staining of *CDX2/T*(left) and *CER1/ SOX17*(middle) in EBs cultured in E6 at day 2. GFP indicates the location of *BMP4*-producing cells. Each experiment was performed for at least three independent replicates (technical replicates). Scale bars: 500 μm.

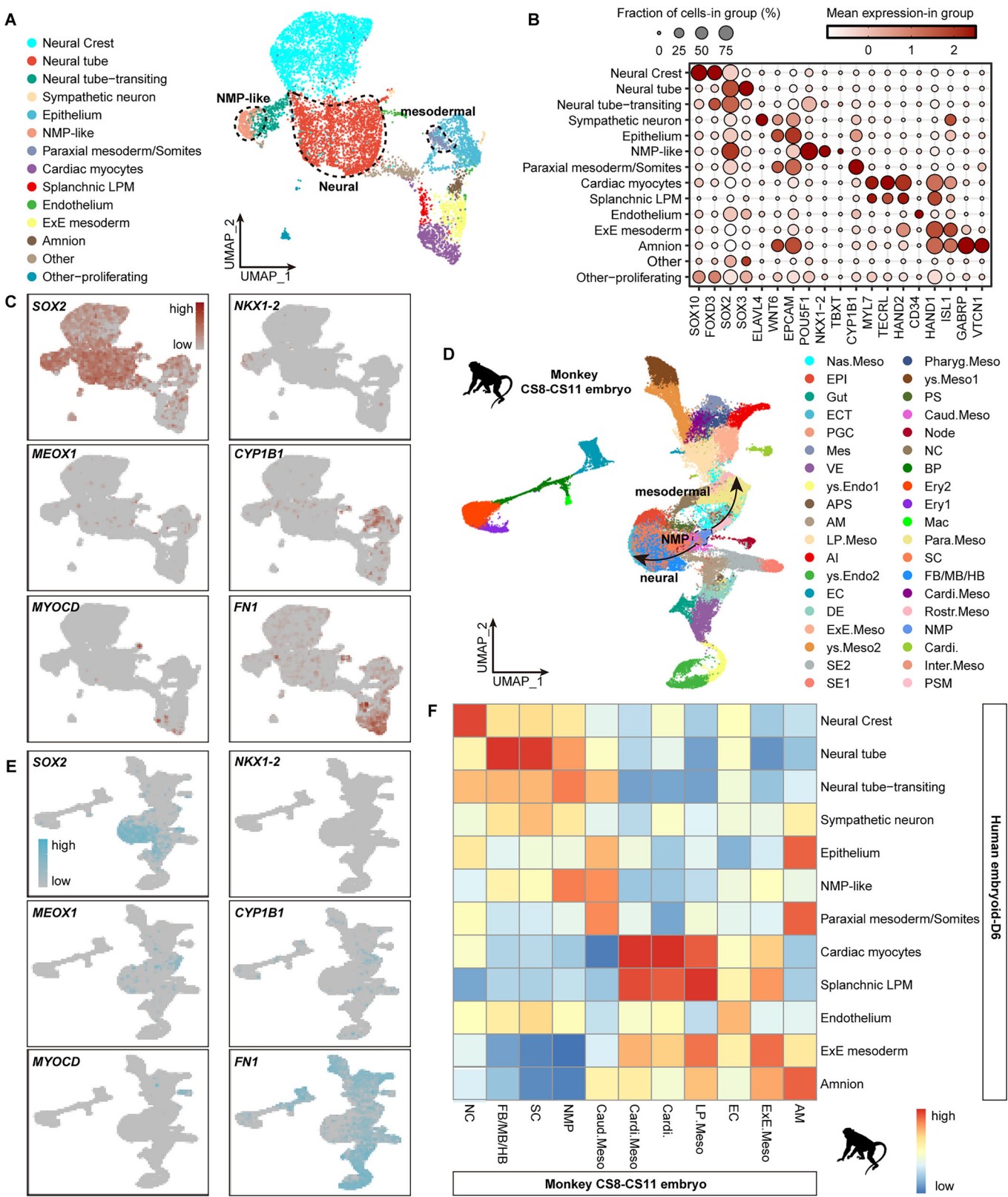

**Figure EV7.   scRNA-seq analysis of human VCLS at day 6, related to Fig. 6.**

(A) UMAP plot showing the cell clusters in VCLS at day 6, cells are colored by different cell types. (B) Dot plot showing the averaged expression levels of indicated marker genes of each cell cluster. (C) UMAP plots showing the expression of *SOX2, NKX1-2, MEOX1, CYP1B1, MYOCD* and *FN1* in VCLS at day 6. (D) UMAP plot showing the cell clusters in CS8-11 monkey embryo, cells are colored by different cell clusters. Two black lines with arrows highlight the potential differential trajectories of NMP cells (to neural lineage or mesodermal lineage) (E) UMAP plots showing the expression of *SOX2, NKX1-2, MEOX1, CYP1B1, MYOCD* and *FN1* in monkey embryo. (F) Heatmap showing the Pearson correlation of related cell clusters between monkey embryo and VCLS.

