## [Peer Review File · The EMBO Journal]

BMP4 initiates and patterns ventral-caudal structures in zebrafish and human pluripotent stem cell aggregates

Yan-Yi Xing, Ying Huang, Tao Cheng, Yi-Meng Tian, Yang Dong, Zi-Xin Jin, Hai-Rong Pu, Tao Luo, Xiang Liu, Hao-Nan Shen, Jing Mo, Jun Ma, Jun-Feng Ji, and Peng-Fei Xu

Corresponding authors: Peng-Fei Xu (pengfei_xu@zju.edu.cn) , Tao Cheng (chengtao2ldo@zju.edu.cn)

Review Timeline:

Transferred from Review Commons:	10th Jul 25
Editorial Decision:	3rd Sep 25
Revision Received:	21st Oct 25
Accepted:	30th Oct 25

Editor: Ieva Gailite

Transaction Report:

Review
COMMONS

This manuscript was transferred to The EMBO Journal following peer review at Review Commons.

Review #1**1. Evidence, reproducibility and clarity:****Evidence, reproducibility and clarity (Required)******Summary:****

Here, Xing and colleagues investigate signals that specify the caudal gastrula organizer using a combination of zebrafish embryos, zebrafish explants, and human stem cells. They show that injection of mRNA encoding BMP ligands into a single zebrafish blastomere is sufficient to induce a secondary tail within intact embryos, and to generate a tail-like structure when these blastomeres are explanted and cultured in isolation. They use single cell RNA-sequencing to determine that these tail-like structures contain predominantly paraxial mesoderm and tailbud derivatives and are largely devoid of neural and axial mesoderm tissues. Hybridization chain reaction demonstrates that these tissues are patterned along the anterior posterior axis, including proper regionalization of trunk and tail somites and lateral/intermediate mesoderm derivatives. These structures also undergo axis extension-like morphogenetic movements. Treatment of human pluripotent stem cells with BMP4 alone similarly induces caudal organizer activity, promoting axial elongation of 3D cell aggregates and inducing a secondary tail when transplanted into zebrafish embryos. However, these findings are largely contradictory to a body of published work, including that of the corresponding author. These previous studies should be acknowledged and the discrepancies need to be addressed. Finally, no samples numbers are presented for the data presented herein, making it unclear how representative any of the data shown are.

****Major comments:****

1. The major finding that BMP alone can induce a tail both in vivo and in explants is at odds with results from multiple previous studies, including those by the corresponding author. For example, it was shown in Fauny et al, 2009 and Xu et al, 2014 that injection of BMP RNAs into a single zebrafish blastomere did not give rise to a secondary axis while co-injection of BMP and Nodal did. This led to the conclusion that BMP and Nodal together are necessary for formation of the tail organizer, but BMP alone is not sufficient. The fact that these discrepancies are unremarked upon here is quite surprising.
2. It was further shown in Xu et al, 2014 that uninjected zebrafish explants express high levels of BMP ligands, so it is unclear why, if BMP is sufficient for explant extension (as shown in the current study), uninjected explants do not extend without additional BMP.

3. The manuscript fails to acknowledge previous work by the Arias, Veenvliet, Kageyama, van Oudenaarden, Shendure, Warmflash, and other labs that investigated the role of BMP in mammalian gastruloid development. These studies show that BMP alone is insufficient for axis elongation but is sufficient for paraxial, lateral, and intermediate mesoderm formation. In general, the current manuscript ignores similar and related work done in the past and fails to present the study in the larger context of the field. Remarkably, this includes a failure to acknowledge (again) the corresponding author's own work showing that BMP treatment of mouse PSCs produces an organizer-like population that induces gastruloid elongation. The current study essentially recapitulates this experiment with human cells, limiting its novelty.

4. Neither sample nor independent trial numbers are presented for any of the experiments. It is therefore unclear whether the images presented are representative, or how common the reported results are. In a rare example when a sample number is reported, it is for 13 individual cells tracked within a single explant. This is far too few cells from far too few explants to draw any conclusions about how these cells behave.

2. Significance:

Significance (Required)

Embryonic organizers are a fascinating and incompletely understood aspect of early development. Previous work from the corresponding author and their colleagues made huge advances in this area in the past 15+ years by demonstrating that ratios of BMP and Nodal morphogen signaling are sufficient to pattern the entire embryonic axis by inducing a continuum of head-to-tail organizer activity. The current manuscript contradicts this previous work, demonstrating that BMP alone induces the tail organizer. This seems therefore to be a significant departure from our current understanding of organizer formation. As such, the current study must be placed in context with this previous work and the noted discrepancies discussed to better understand why similar results were not observed in prior experiments. If the reason for these inconsistencies can be identified, this shift in understanding would be quite interesting to many developmental and stem cells biologists. This assessment is based on my 15+ years of experience studying morphogenesis during gastrulation and neurulation in multiple vertebrate species, including zebrafish. However, I do not have in-depth knowledge of single cell RNA-seq analysis methods.

3. How much time do you estimate the authors will need to complete the suggested revisions:

Estimated time to Complete Revisions (Required)

(Decision Recommendation)

Between 3 and 6 months

4. Review Commons values the work of reviewers and encourages them to get credit for their work. Select 'Yes' below to register your reviewing activity at Web of Science Reviewer Recognition Service (formerly Publons); note that the content of your review will not be visible on Web of Science.

Yes

Review #2

1. Evidence, reproducibility and clarity:

Evidence, reproducibility and clarity (Required)

Xing et al., report a series of interesting experiments in zebrafish, hESCs and hESC/zebrafish chimeras in support of the thesis that BMP4 alone is sufficient to induce the caudal vertebrate embryonic organizer. They propose that this organizer replicates morphogenesis and key differentiation pathways observed during normal vertebrate development and that this role of BMP is conserved from zebrafish to human.

The key observations include, first, injection of RNA encoding BMP4 in the animal blastomeres of the zebrafish embryo induces ectopic tailbud-like structure and later formation of posterior structures resembling tail and possessing neuromesodermal progenitors, somites, blood, pronephros, and neural tissues but critically lacking notochord and endoderm. Further, Bmp4 activates key signaling pathways pivotal for early development, including Nodal, Wnt and FGF. Another interesting observation is that explants of the animal poles of embryos injected with moderate doses of BMP RNA underwent elongation and formed somites, had neural tissues and exhibited anteroposterior polarity. Further, BMP4 treated hESCs when transplanted into the animal pole of zebrafish embryo induce secondary tail-like structures. Finally, an aggregation of BMP4-inducible PSCs with an aggregation of normal human PSCs led to formation of embryoids that exhibited anteroposterior polarity and expressed genes equivalent to those observed in ectopic tail-like structures induced by BMP4 in zebrafish. The authors performed thorough analysis of cell fates with single-cell RNA-seq, fluorescent in situ hybridization and antibody stainings that showed induced cell types and their spatial arrangement over time. This is interesting work that should be of interest to the fast-

advancing field of vertebrate and human embryogenesis. However, there are concerns about the current interpretations of these experiments and that the authors conclusions are not supported by the results. Moreover, the significance of this work would be elevated by properly placing this work in the context of decades of developmental genetics and embryology.

****Main concerns****

1. The main conclusion of the authors in the Discussion and at the end of several sections is:

- Line 392 "Here, for the first time, we demonstrate that BMP4 alone is sufficient to induce caudal-like structures of a vertebrate embryo both in vivo and in vitro.

- Line 217 "Collectively, these findings demonstrate that Bmp4 explant exhibit morphological and

molecular characteristics of the posterior embryo and undergo a somitogenesis progression akin to that observed in vivo.

- Line 281 "Taken together, these results demonstrate that Bmp4 explant exhibits structural and

molecular features similar to those of the vertebrate caudal region, and thus we designate our model system as a caudal-like structure (CLS).

2. This conclusion of "tail-like" "caudal-like" structure implies that these structures have the full complement of cell types present in the vertebrate tail. This not supported by the presented data, as the authors have demonstrated that these induced structures in zebrafish and hPSC embryoids lack axial mesoderm and endoderm." Moreover, Line 177 "It is noteworthy that the mesodermal and neural domains specified in Bmp4 explant are radially symmetrical viewed from the sagittal section, suggesting the lack of left-right axis". The presented results are still interesting and instructive for understanding "the minimal signals required for building vertebrate embryo". However, claiming that BMP is a caudal organizer and can induce tails or tail-like structures, is inconsistent with the presented data and would confuse rather than advance the field.

Indeed, presenting these results and discussing them in the context of the previous studies would significantly improve this manuscript and its significance. The structures induced by BMP4 observed here resemble tail-like structures of highly ventralized zebrafish mutants like ichabod (b-catenin 2 Kelly et al., Weinberg lab, 2000) that form somites but lack axial mesoderm.

The author emphasize that neuromesodermal progenitors described by Martin and

Kimelman are present in the induced structures; however, Martin and Kimelman described dorsal and ventral mesodermal precursors in the tailbud that give rise to notochord, floor plate and hypochord (not mentioned in the current manuscript). Clearly, BMP4 induces incomplete ventral organizer and this should be clearly acknowledged.

Relevant to this, in the introduction the relevant previous work is not appropriately included: Lines 59-60 "However, the minimal conditions that can induce an organizer capable of inducing and patterning the caudal part of the vertebrate embryo have not been addressed." Whereas the authors cite the work by Bernard and Christine Thisse and acknowledge their advice, they do not introduce their work addressing this point. The manuscript by Thisse & Thisse "Construction of a vertebrate embryo from two opposing morphogen gradients (Science, 2014), addressed exactly this point "the minimal conditions and factors that would be sufficient to organize the embryo". They went on to show by injecting BMP and Nodal mRNAs into the animal pole of the zebrafish embryo such manipulation generated ectopic axes with BMP inducing tail tissues. Importantly, these ectopic axes, or embryoids developing from blastoderm explants after injection of BMP and Nodal RNA had the three germ layers and also notochord. As mentioned above the posterior structures induced by BMP injections described in this manuscript lack notochord and endoderm. This is consistent with Thisse's conclusions that "high BMP/low Nodal is needed to induce complete secondary axis including tail with notochord.

3. Lines 51-52 "The developmental trajectory of the tailbud and the critical signals regulating its formation remain unclear". This is not correct, the authors should cite the classical lineage tracing studies from the Kimmel laboratory that provided evidence that tailbud in zebrafish largely arises from cells located in the ventrolateral region of the blastoderm margin (zebrafish blastopore) (Kimmel et al., 1990). Moreover, Solnica-Krezel lab demonstrated that high BMP levels in the ventral margin inhibit dorsal convergence movements while promoting epibolic movements towards the tailbud, and that mutants with elevated BMP signaling have larger tailbuds and tails at the expense of head and trunk regions (Myers et al., 2002; Gonzalez et al., 2000).

4. It will be also important to discuss what happens in vertebrate embryos when BMP signaling is reduced or absent. Work from Mary Mullins lab in zebrafish on *bmp2b*, *bmp7* and many other mutants and of Brigid Hogan in the mouse on *bmp4* show that BMP signaling is required for tail development and in its absence embryos form with tail truncations. Myers et al., 2002 showed in zebrafish that this is because cells at the ventral blastoderm margin instead of moving to the tailbud engage in convergence and extension movements. This body of work shows that BMP4 is required for tailbud formation, as well as for specification of ventral cells like blood etc. This corresponds well with the authors work addressing what BMP4 is sufficient for.

5. The statement "In zebrafish, tailbud formation occurs at the blastoderm margin convergence and fuses at the end of gastrulation" needs to be revised. First, the prospective tailbud tissues, from the ventral and dorsal part of the blastoderm margin are brought together largely by epibolic movements that spread the blastoderm around the yolk (Kimmel et al., 1996). Contrary to the authors' statement, ventral mesoderm does not converge dorsally but moves towards the vegetal pole in epibolic movements (Myers et al., 2002). The prospective tailbud tissues undergo subduction as described by Kanki and Ho,
6. Line 123 "Upon excision, the explant rapidly became a spherical shape, and established a BMP signaling gradient". This is well illustrated with pSmad1,5 staining at the spherical shape. It would be important to analyze pSmad1,5 staining at the elongation stages.
7. The section describing morphogenetic movements is very superficial, yet the authors make sweeping conclusions about high cell movements in the induces structures resemble cell movements in vivo. Lines 294-293 "During the gastrulation stage, cells near the injection site converged toward the anterior of the explant and involuted outside-in very quickly (Figure 4B, supplementary movies S2, S3). In zebrafish or in mammalian gastrulae, mesodermal cells do not "involute" (this is tissue-based movement like in frogs) but rather ingress as individuals. Demonstrating this would require additional data.
8. Lines "332-333 Taken together, these findings suggest that the Bmp4 explant recapitulates not only the structural and molecular characteristics of caudal structures but also the morphogenetic progresses of gastrulation and tail elongation observed in vivo". This conclusion is not supported by the data presented in Figure 4 where altered distances between cells over time are consistent with tissue elongation, but do not illuminate underlying cell behaviors. The conclusions should be significantly toned down or this section removed from the manuscript.
9. The experiment, which BMP4 treated human PSCs were grafted into zebrafish animal poles at sphere stage leading to induction of posterior-like structures is interesting. Please provide more information about how long hPSCs were treated with BMP4 and at what concentration? Importantly, as BMP signaling acts via a positive feedback loop, do these hPSCs express human BMP4 gene? What would suggest that this experiment is simply equivalent to injecting bmp4 RNA?
10. The observation that injection of bmp4 RNA into animal pole of zebrafish embryos induces wnt, nodal and fgf gene expression is significant and should be discussed. After all, this is the cascade proposed to induce mesendoderm in mouse embryos and human gastruloids. It would be interesting to discuss similarities and differences between the authors experiments and hPSC-based 2D micropatterned gastruloids (Warmflash et al., 2014; Minn et al., 2020) in which germ layers are induced by BMP4.

Minor points:

1. Lines 43-45 "Initially, lineage specification during gastrulation results in the formation of three germ layers, where processes like involution, convergence and extension shape and elongate these layers". Please rephrase, "elongation" is part of "shaping", but what gastrulation also achieves is rearranging the positions of the induced germ layers in the embryo (internalization). Also "involution" is a specific way of internalization observed in amphibian embryos.
2. Gene and RNA names are not consistently written in italics.
3. Figure 1G. What does green fluorescence label?
4. Figure 5C "notocord" correct to "notochord"

2. Significance:

Significance (Required)

The work is significant as it addresses the fundamental inductive process underlying the formation of the vertebrate body plan during embryogenesis, including activity of the embryonic organizer, what are the minimal signaling requirements for the posterior body organizer. The authors performed interesting and original experiments in zebrafish and hPSC-based embryoids that complement and advance the current knowledge. When revised this work will advance our understanding of the sufficiency of BMP4 signaling for aspects of posterior body development and its evolutionary conservation.

3. How much time do you estimate the authors will need to complete the suggested revisions:

Estimated time to Complete Revisions (Required)

(Decision Recommendation)

Between 1 and 3 months

Yes

Review #3

1. Evidence, reproducibility and clarity:

Evidence, reproducibility and clarity (Required)

****Summary:****

In this study, the authors demonstrate that BMP4 overexpression is sufficient to induce caudal-like structures in zebrafish embryos, zebrafish explants, and human embryoid bodies. Transplantation of BMP4-treated human cells into the animal pole of zebrafish embryos also successfully triggered the formation of a secondary axis. These caudal structures closely resembled embryonic caudal regions at the molecular level, aligned with appropriate positions along the anterior-posterior axis, and exhibited similar cell movement dynamics. Based on these findings, the authors propose that BMP alone is sufficient to induce a caudal organizer in otherwise naïve ectoderm. Notably, the ectopic axes lack anterior neural tissue and axial mesoderm, though the reasons for this remain unexplained.

****Major comments:****

While the results are intriguing and provide a model that recapitulates aspects of posterior axis formation, two significant issues remain unresolved.

First, grafting BMP-expressing cells into the early zebrafish blastoderm activates the Nodal, Wnt, and FGF pathways as early as 6 hours post-fertilization (hpf), well before any morphogenetic effects are apparent (the bulge forms at 8 hpf). Since these pathways, along with BMP, are known to contribute to embryo patterning and tissue formation and are capable of inducing ectopic axes, it is unclear which pathway(s) are responsible for axis induction in these experiments. This raises questions about the bold claim that BMP alone is sufficient to establish a posterior organizer.

Second, the BMP-expressing cells are either not consistently shown in the experiments or appear to scatter quickly within the new posterior structure, as observed in zebrafish embryos and explants (Figs. 1B and 4B) and in hPSC-transplanted fish embryos (Fig. 5C). Few BMP-secreting cells seem to remain in the caudal region of the induced axis. The paper's central claim is therefore not fully supported by the reported observations. Alternatively, it is possible that BMP-secreting cells initiate a signaling cascade early in development, setting up a caudal organizer-like region that subsequently functions without continued BMP activity.

To strengthen the main claim, it is critical to provide compelling evidence of BMP-producing cells contributing to the late-stage caudal structure, akin to the results shown in

hPSC embryoid fusions (Fig. 5C). This would clarify whether BMP alone maintains a posterior organizer or merely initiates downstream signaling events that sustain the structure.

****Minor comments:****

Can the authors track more cells? 13 cells seem not enough to claim that 'the Bmp4 explant recapitulates not only the structural and molecular characteristics of caudal structures but also the morphogenetic progresses of gastrulation and tail elongation observed in vivo'.

2. Significance:

Significance (Required)

This study is impressive, employing a powerful combination of model systems: zebrafish embryos, embryonic explants, human pluripotent stem cell xenografts, and 3D human stem cell aggregates. The single-cell RNA analyses are particularly insightful, shedding light on the embryonic origin and identity of the induced caudal tissues. When combined with HCR in situ hybridization for spatial characterization, these methods provide a robust framework for understanding the molecular basis of caudal tissue induction.

However, the characterization of tissue and cell behavior remains preliminary, and I strongly encourage the authors to enhance this aspect of the study. Employing quantitative imaging tools to analyze their systems in greater detail would not only substantiate their claim that "the BMP4 explant recapitulates not only the structural and molecular characteristics of caudal structures but also the morphogenetic processes of gastrulation and tail elongation observed in vivo," but also significantly expand the scope of the study. This improvement would attract a broader audience and elevate the impact of this work.

3. How much time do you estimate the authors will need to complete the suggested revisions:

Estimated time to Complete Revisions (Required)

(Decision Recommendation)

Between 3 and 6 months

4. Review Commons values the work of reviewers and encourages them to get credit for their work. Select 'Yes' below to register your reviewing activity at Web of Science

Reviewer Recognition Service (formerly Publons); note that the content of your review will not be visible on Web of Science.

Yes

Full Revision

Manuscript number: RC-2024-02776

Corresponding author(s): Tao Cheng, Peng-Fei Xu

1. General Statements

We sincerely thank the editor and reviewers for their time, effort, and insightful comments on our manuscript. We are grateful for the constructive feedbacks and positive evaluations, which have helped us improve the clarity, rigor, and overall quality of our study. In response to the reviewers' suggestions, we have undertaken a thorough revision of the manuscript. We believe that the revised version significantly enhances the presentation and scientific strength of our work.

All substantial changes have been highlighted in yellow in the revised manuscript for ease of reference. And major revisions are summarized below:

- **Functional divergence of BMP ligands:** resolved discrepancies between prior studies (*Fauny et al., 2009; Xu et al., 2014*) by demonstrating that zebrafish Bmp4 uniquely induces posterior mesoderm and activates Wnt/Nodal/FGF pathways. Bmp2b lacks this capability.
- **Mechanism of *bmp4*-induced morphogenesis:** Bmp4 initiates caudal mesoderm development requires Nodal/Wnt/FGF cooperation for explant elongation. Endogenous/low-dose Bmp4 fails to induce elongation due to insufficient Wnt/Nodal activation and absence of a Bmp4 gradient.
- **Refined morphogenetic claims:** revised conclusions about *bmp4* explant morphogenesis, including: detailed analysis of internalization movements and tail elongation processes; adjusted description of caudal organizer per reviewer's suggestion.
- **Lineage tracing:** *bmp4*-injected blastomeres contribute to lateral plate mesoderm and enveloping layer.
- **Xenograft experiments:** DOX-induced iBMP4 triggers endogenous human *BMP4* expression (suggesting a positive feedback loop).
- **Human gastruloid comparisons:** 3D ventral-caudal like structures (VCLS) (BMP4-induced) resemble 2D micropatterned gastruloids in mTESR medium. In E6 medium, BMP4 exclusively induces posterior mesoderm differentiation.
- **Enhanced methodological rigor:** added exact sample sizes (N) and independent trial numbers for all experiments.

Reviewer #1 (Evidence, reproducibility and clarity (Required)):

Summary:

Here, Xing and colleagues investigate signals that specify the caudal gastrula organizer using a combination of zebrafish embryos, zebrafish explants, and human stem cells. They show that injection of mRNA encoding BMP ligands into a single zebrafish blastomere is sufficient to induce a secondary tail within intact embryos, and to generate a tail-like structure when these blastomeres are explanted and cultured in isolation. They use single cell RNA-sequencing to determine that these tail-like structures contain predominantly paraxial mesoderm and tailbud derivatives and are largely devoid of neural and axial mesoderm tissues. Hybridization chain reaction demonstrates that these tissues are patterned along the anterior posterior axis, including proper regionalization of trunk and tail somites and lateral/intermediate mesoderm derivatives. These structures also undergo axis extension-like morphogenetic movements. Treatment of human pluripotent stem cells with BMP4 alone similarly induces caudal organizer activity, promoting axial elongation of 3D cell aggregates and inducing a secondary tail when transplanted into zebrafish embryos. However, these findings are largely contradictory to a body of published work, including that of the corresponding author. These previous studies should be acknowledged and the discrepancies need to be addressed. Finally, no samples numbers are presented for the data presented herein, making it unclear how representative any of the data shown are.

Response:

We sincerely thank Reviewer #1 for the thoughtful comments and valuable suggestions. In response, we have conducted additional experiments and revised the manuscript accordingly to address the discrepancies highlighted between our current work and previous studies. Furthermore, we have provided the sample numbers for all data presented.

Major comments:

1. The major finding that BMP alone can induce a tail both in vivo and in explants is at odds with results from multiple previous studies, including those by the corresponding author. For example, it was shown in Fauny et al, 2009 and Xu et al, 2014 that injection of BMP RNAs into a single zebrafish blastomere did not give rise to a secondary axis while co-injection of BMP

and Nodal did. This led to the conclusion that BMP and Nodal together are necessary for formation of the tail organizer, but BMP alone is not sufficient. The fact that these discrepancies are unremarked upon here is quite surprising.

Response:

We thank Reviewer #1 for highlighting this important issue and apologize for the inadequacies in our original description.

In the revised manuscript, we have added new experiments to clarify the distinct properties of the two BMP ligands, Bmp2b and Bmp4, which help explain the discrepancies between our current findings and previous reports (Please see Lines 178-201, 491-503). Specifically, we show that zebrafish Bmp4 can induce mesoderm formation and activate key signaling pathways, including Wnt, Nodal, and FGF (**Figure 1D and Figure S2D**), in a manner consistent with observations in mouse¹ and human² BMP4 studies. In contrast, Bmp2b, which was used in the studies by Fauny et al., 2009, and Xu et al., 2014, does not induce mesoderm formation, secondary tail structures, or activation of Wnt/Nodal signaling under similar conditions (**Figures 1A, 1D, S1A, S1N-S1P, S2D and S2F**).

These results explain why Fauny et al. and Xu et al. reported that Bmp2b required co-activation with Nodal to induce secondary tails, whereas in our current study, Bmp4 alone is sufficient to drive ventral-caudal like structures (VCLS) formation.

To further address this point, we performed explant assays by injecting different doses of *bmp2b* and *bmp4* mRNAs. We found that only an appropriate concentration of *bmp4*, but not *bmp2b*, could induce the formation of ventral-caudal structures through the activation of Wnt, Nodal, and FGF signaling pathways. These additional data have been included in the revised manuscript (**Figures 1D, S1I-S1P, and S2D**).

2. It was further shown in Xu et al, 2014 that uninjected zebrafish explants express high levels of BMP ligands, so it is unclear why, if BMP is sufficient for explant extension (as shown in the current study), uninjected explants do not extend without additional BMP.

Response:

We thank Reviewer #1 for raising this important question. As the reviewer pointed out, uninjected zebrafish animal pole explants fail to undergo significant morphogenesis or cell fate specification, yielding only epidermal and forebrain cells, despite inheriting high levels of maternally provided BMP ligands^{3,4}.

We believe there are two primary reasons for this observation. First, maternally inherited

BMP levels are likely insufficient to initiate morphogenesis and lineage differentiation. In the revised manuscript, we generated explants by injecting varying doses of *bmp4* mRNA. We found that only an optimal concentration of *bmp4* could induce ventral/caudal structure formation, likely through the activation of Wnt, Nodal, and FGF signaling pathways (**Figures 1D, S1I–S1P and S2D**). Second, the spatial gradient of BMP signaling plays a critical role. Uninjected zebrafish animal pole explants exhibit homogeneous BMP signaling and lack the necessary gradient to properly pattern the embryo⁴.

3. The manuscript fails to acknowledge previous work by the Arias, Veenliet, Kageyama, van Oudenaarden, Shendure, Warmflash, and other labs that investigated the role of BMP in mammalian gastruloid development. These studies show that BMP alone is insufficient for axis elongation but is sufficient for paraxial, lateral, and intermediate mesoderm formation. In general, the current manuscript ignores similar and related work done in the past and fails to present the study in the larger context of the field. Remarkably, this includes a failure to acknowledge (again) the corresponding author's own work showing that BMP treatment of mouse PSCs produces an organizer-like population that induces gastruloid elongation. The current study essentially recapitulates this experiment with human cells, limiting its novelty.

Response: We thank the reviewer#1 for pointing out this important issue. We appreciated that the reviewer reminded us those important works regarding to the role of BMP signaling in mammalian gastruloid development. We have acknowledged these works in appropriate context of our revised manuscript (Refs 21, 22, 23, 24, 25, 26, 27). As the Reviewer #2 and Reviewer #3 indicated, our current work emphasized that BMP4 alone is sufficient to induce/initiate the formation of the ventral-caudal vertebrate embryonic organizer, and that this BMP4-mediated role is evolutionarily conserved across vertebrates, from zebrafish to humans. Our study redefines the molecular basis of the tail/caudal organizer⁵. In our revised manuscript, we generated explants by injecting varying doses of *bmp4* and *bmp2b* mRNA, and characterized these explants. We found that only an optimal concentration of *bmp4* could induce ventral-caudal like structures (VCLS), likely via activation of Wnt, Nodal, and FGF signaling pathways (**Figures 1D, S1I–S1P, and S2D**).

4. Neither sample nor independent trial numbers are presented for any of the experiments. It is therefore unclear whether the images presented are representative, or how common the

Full Revision

reported results are. In a rare example when a sample number is reported, it is for 13 individual cells tracked within a single explant. This is far too few cells from far too few explants to draw any conclusions about how these cells behave.

Response: We thank the reviewer#1 for raising this important issue and apologize for our omissions. We have added the exact number of biological and technical replicates into corresponded figures and the figure legends. In our revised manuscript, we performed further analyses on the cell movements and morphogenesis of *bmp4*-injected explants and embryos. we tracked and quantified the cell movement trajectories of hundreds of individual cells in the explants (**Figures 4C-4I and S5A-S5F**).

Reviewer #1 (Significance (Required)):

Embryonic organizers are a fascinating and incompletely understood aspect of early development. Previous work from the corresponding author and their colleagues made huge advances in this area in the past 15+ years by demonstrating that ratios of BMP and Nodal morphogen signaling are sufficient to pattern the entire embryonic axis by inducing a continuum of head-to-tail organizer activity. The current manuscript contradicts this previous work, demonstrating that BMP alone induces the tail organizer. This seems therefore to be a significant departure from our current understanding of organizer formation. As such, the current study must be placed in context with this previous work and the noted discrepancies discussed to better understand why similar results were not observed in prior experiments. If the reason for these inconsistencies can be identified, this shift in understanding would be quite interesting to many developmental and stem cells biologists. This assessment is based on my 15+ years of experience studying morphogenesis during gastrulation and neurulation in multiple vertebrate species, including zebrafish. However, I do not have in-depth knowledge of single cell RNA-seq analysis methods.

Response: We sincerely thank the reviewer for recognizing our works, and greatly appreciate the constructive suggestions from the reviewer. Our previous work demonstrated that opposing gradients of BMP and Nodal signaling in zebrafish can induce a complete body axis⁴. However, the precise mechanistic underpinnings remained unclear. Such as: what specific cell types and spatial patterns are induced by Nodal or BMP alone? In our recently published study³, we demonstrated that Nodal signaling alone is sufficient to drive axial mesendoderm development

Full Revision

and orchestrate the patterning of head. This current work highlights BMP4 alone can recapitulate the functional role of the ventral-caudal organizer and induce the development and morphogenesis of ventral-caudal like tissues in both zebrafish and humans probably through activating Wnt, Nodal, and FGF signaling pathways. Collectively, this work deepens our understanding of BMP4's conserved regulatory roles across vertebrate species, including zebrafish, mice, and humans. We have incorporated this information into discussion section in our revised manuscript (Please see Lines 491-516). We thank the reviewer #1 again for the constructive suggestions and professional comments. We believe our manuscript has been significantly improved after addressing the concerns raised by the reviewer.

Reviewer #2 (Evidence, reproducibility and clarity (Required)):

Xing et al., report a series of interesting experiments in zebrafish, hESCs and hESC/zebrafish chimeras in support of the thesis that BMP4 alone is sufficient to induce the caudal vertebrate embryonic organizer. They propose that this organizer replicates morphogenesis and key differentiation pathways observed during normal vertebrate development and that this role of BMP is conserved from zebrafish to human.

The key observations include, first, injection of RNA encoding BMP4 in the animal blastomeres of the zebrafish embryo induces ectopic tailbud-like structure and later formation of posterior structures resembling tail and possessing neuromesodermal progenitors, somites, blood, pronephros, and neural tissues but critically lacking notochord and endoderm. Further, Bmp4 activates key signaling pathways pivotal for early development, including Nodal, Wnt and FGF. Another interesting observation is that explants of the animal poles of embryos injected with moderate doses of BMP RNA underwent elongation and formed somites, had neural tissues and exhibited anteroposterior polarity. Further, BMP4 treated hESCs when transplanted into the animal pole of zebrafish embryo induce secondary tail-like structures. Finally, an aggregation of BMP4-inducible PSCs with an aggregation of normal human PSCs led to formation of embryoids that exhibited anteroposterior polarity and expressed genes equivalent to those observed in ectopic tail-like structures induced by BMP4 in zebrafish. The authors performed thorough analysis of cell fates with single-cell RNA-seq, fluorescent in situ hybridization and antibody stainings that showed induced cell types and their spatial arrangement over time. This

is interesting work that should be of interest to the fast-advancing field of vertebrate and human embryogenesis. However, there are concerns about the current interpretations of these experiments and that the authors conclusions are not supported by the results. Moreover, the significance of this work would be elevated by properly placing this work in the context of decades of developmental genetics and embryology.

Response: We thank the reviewer for recognizing our work. We greatly appreciate the reviewer's professional and valuable comments, and we are thankful for the constructive suggestions that helped us improving our manuscript. As suggested by the reviewer, we have revised our manuscript and supplemented additional experiments and analyses which have been summarized in General Statements section of this response letter. We hope our responses have addressed the concerns raised by the reviewer.

Main concerns

1. The main conclusion of the authors in the Discussion and at the end of several sections is:
 - Line 392 "Here, for the first time, we demonstrate that BMP4 alone is sufficient to induce caudal-like structures of a vertebrate embryo both in vivo and in vitro.
 - Line 217 "Collectively, these findings demonstrate that Bmp4 explant exhibit morphological and molecular characteristics of the posterior embryo and undergo a somitogenesis progression akin to that observed in vivo.
 - Line 281 "Taken together, these results demonstrate that Bmp4 explant exhibits structural and molecular features similar to those of the vertebrate caudal region, and thus we designate our model system as a caudal-like structure (CLS).
2. This conclusion of "tail-like" "caudal-like" structure implies that these structures have the full complement of cell types present in the vertebrate tail. This not supported by the presented data, as the authors have demonstrated that these induced structures in zebrafish and hPSC embryoids lack axial mesoderm and endoderm." Moreover, Line 177 "It is noteworthy that the mesodermal and neural domains specified in Bmp4 explant are radially symmetrical viewed from the sagittal section, suggesting the lack of left-right axis" .The presented results are still interesting and instructive for understanding "the minimal signals required for building vertebrate embryo". However, claiming that BMP is a caudal organizer and can induce tails or tail-like structures, is inconsistent with the presented data and would confuse rather than advance the field.

Full Revision

Indeed, presenting these results and discussing them in the context of the previous studies would significantly improve this manuscript and its significance. The structures induced by BMP4 observed here resemble tail-like structures of highly ventralized zebrafish mutants like *ichabod* (*b-catenin 2* Kelly et al., Weinberg lab, 2000) that form somites but lack axial mesoderm.

Response: We thank the reviewer for highlighting this important issue and providing insightful suggestions. We fully acknowledge that BMP4 does not induce a complete tail structure. Instead, it generates a ventralized caudal structure that exhibits molecular and morphological similarities to the caudal region of the embryo but lacks axial mesoderm and endoderm. Therefore, it is inadequate to claim that BMP functions as a caudal-tail organizer.

Prompted by the reviewer's comments, we have revised our terminology from "caudal-tail organizer" to "ventral-caudal organizer" throughout the manuscript and the title, to avoid confusion.

The author emphasize that neuromesodermal progenitors described by Martin and Kimelman are present in the induced structures; however, Martin and Kimelman described dorsal and ventral mesodermal precursors in the tailbud that give rise to notochord, floor plate and hypochord (not mentioned in the current manuscript). Clearly, BMP4 induces incomplete ventral organizer and this should be clearly acknowledged.

Response: We thank the reviewer for raising this important issue. Neuromesodermal progenitors (NMPs) are a population of bipotent cells capable of giving rise to both spinal cord and paraxial mesoderm derivatives, characterized by the co-expression of SOX2 and TBXT. In our study, we identified a population of cells co-expressing SOX2 and TBXT in both zebrafish explants and *BMP4*-induced human embryoids. Accordingly, we initially defined this cluster of cells as neuromesodermal progenitors (NMPs).

As the reviewer noted, "Martin and Kimelman described dorsal and ventral mesodermal precursors in the tailbud that give rise to the notochord, floor plate, and hypochord"—structures that are absent in our explants and embryoids. It remains an interesting but unresolved question why SOX2/TBXT double-positive cells cannot give rise to these dorsal tissues in our model. This raises the possibility that NMPs comprise distinct subclusters derived from dorsal or ventral origins, or that their differentiation depends on dorsal–ventral microenvironmental cues⁶.

To avoid an imprecise definition, we have revised our terminology from "NMPs" to "NMP-

Full Revision

like cells” throughout the manuscript. We have also added the suggested discussion to acknowledge that *BMP4*-induced incomplete ventral organizer that do not give rise to notochord, floor plate, or hypochord (Please see Lines 491-516).

Relevant to this, in the introduction the relevant previous work is not appropriately included: Lines 59-60 "However, the minimal conditions that can induce an organizer capable of inducing and patterning the caudal part of the vertebrate embryo have not been addressed." Whereas the authors cite the work by Bernard and Christine Thisse and acknowledge their advice, they do not introduce their work addressing this point. The manuscript by Thisse & Thisse "Construction of a vertebrate embryo from two opposing morphogen gradients (Science, 2014), addressed exactly this point "the minimal conditions and factors that would be sufficient to organize the embryo". They went on to show by injecting BMP and Nodal mRNAs into the animal pole of the zebrafish embryo such manipulation generated ectopic axes with BMP inducing tail tissues. Importantly, these ectopic axes, or embryoids developing from blastoderm explants after injection of BMP and Nodal RNA had the three germ layers and also notochord. As mentioned above the posterior structures induced by BMP injections described in this manuscript lack notochord and endoderm. This is consistent with Thisse's conclusions that "high BMP/low Nodal is needed to induce complete secondary axis including tail with notochord.

Response: We thank the Reviewer for highlighting this important issue and apologize for the earlier omission. As the reviewer noted, our previous work demonstrated that two opposing morphogen gradients of BMP and Nodal can instruct the formation of the entire embryonic axis both *in vivo* and *in vitro*⁴. However, important questions remain to be addressed, such as which specific cell types and spatial patterns are induced by Nodal and BMP alone.

In our recently published study³, we showed that Nodal signaling alone is sufficient to drive axial mesendoderm development and orchestrate head patterning. In the current study, we demonstrate that BMP4 alone—but not BMP2b—can recapitulate the functional role of the ventral-caudal organizer, inducing the development and morphogenesis of ventral-caudal tissues in both zebrafish and humans, likely through activation of Wnt, Nodal, and FGF signaling pathways (**Figures 1A, 1D, S1A, S1N-S1P, S2D and S2F**).

Together, these findings enhance our understanding of the conserved roles of BMP4 across vertebrates, including zebrafish, mice, and humans. We have incorporated this discussion into the revised manuscript (Please see Lines 178-201, 491-503).

Full Revision

3. Lines 51-52 "The developmental trajectory of the tailbud and the critical signals regulating its formation remain unclear". This is not correct, the authors should cite the classical lineage tracing studies from the Kimmel laboratory that provided evidence that tailbud in zebrafish largely arises from cells located in the ventrolateral region of the blastoderm margin (zebrafish blastopore) (Kimmel et al., 1990). Moreover, Solnica-Krezel lab demonstrated that high BMP levels in the ventral margin inhibit dorsal convergence movements while promoting epibolic movements towards the tailbud, and that mutants with elevated BMP signaling have larger tailbuds and tails at the expense of head and trunk regions (Myers et al., 2002; Gonzalez et al., 2000).

Response: We thank the reviewer#2 for pointing out this issue and reminding us those pioneer works regarding to tailbud development. As the reviewer suggested, we have cited these works and deleted this sentence. (Please see Lines 66-71).

4. It will be also important to discuss what happens in vertebrate embryos when BMP signaling is reduced or absent. Work from Mary Mullins lab in zebrafish on *bmp2b*, *bmp7* and many other mutants and of Brigid Hogan in the mouse on *bmp4* show that BMP signaling is required for tail development and in its absence embryos form with tail truncations. Myers et al., 2002 showed in zebrafish that this is because cells at the ventral blastoderm margin instead of moving to the tailbud engage in convergence and extension movements. This body of work shows that BMP4 is required for tailbud formation, as well as for specification of ventral cells like blood etc. This corresponds well with the authors work addressing what BMP4 is sufficient for.

Response: We thank the reviewer for the insightful suggestions. We appreciate the reviewer for highlighting these important studies, according to the suggestions, we have incorporated these works into the introduction and discussion part of the revised manuscript. (Please see Lines 94-101, and 491-503)

5. The statement "In zebrafish, tailbud formation occurs at the blastoderm margin convergence and fuses at the end of gastrulation" needs to be revised. First, the prospective tailbud tissues, from the ventral and dorsal part of the blastoderm margin are brought together largely by epibolic movements that spread the blastoderm around the yolk (Kimmel et al., 1996). Contrary to the authors' statement, ventral mesoderm does not converge dorsally but moves towards the vegetal pole in epibolic movements (Myers et al., 2002). The prospective tailbud tissues undergo subduction as described by Kanki and Ho,

Response: We thank the reviewer#2 for pointing out this issue and reminding us those important works regarding to the formation of tailbud in zebrafish. As the reviewer suggested, we have modified our description in the revised manuscript accordingly. (Please see Lines 66-71)

6. Line 123 "Upon excision, the explant rapidly became a spherical shape, and established a BMP signaling gradient". This is well illustrated with pSmad1,5 staining at the spherical shape. It would be important to analyze pSmad1,5 staining at the elongation stages.

Response: We thank the reviewer#2 for the helpful suggestion. As suggested, we performed pSmad1/5/9 staining on *bmp4*-injected explants at 10 hpf, a stage at which the explants had undergone significant elongation. We observed a clear BMP signaling gradient extending from the injection site to the opposite side (**Figure S2B**). Additionally, we performed HCR staining for *id1* (BMP direct downstream target)⁷, *tbx6*, and *tbxta*, which mark BMP signaling, pre-somitic mesoderm, and caudal organizer respectively, in *bmp4* explants at 12-18 hpf, and found that BMP signaling forms a gradient that is high in ventral-caudal organizer and low in pre-somitic mesoderm (**Figure S2C**).

7. The section describing morphogenetic movements is very superficial, yet the authors make sweeping conclusions about high cell movements in the induces structures resemble cell movements in vivo. Lines 294-293 "During the gastrulation stage, cells near the injection site converged toward the anterior of the explant and involuted outside-in very quickly (Figure 4B, supplementary movies S2, S3). In zebrafish or in mammalian gastrulae, mesodermal cells do not "involute" (this is tissue-based movement like in frogs) but rather ingress as individuals. Demonstrating this would require additional data.

Response: We thank the reviewer#2 for raising this important question. To address this question, we performed additional analyses on tracking cell movements from *bmp4*-injected explants and embryos. We used Imaris to track the cell movements and quantified the cell movement directions for every cell during gastrulation stage. We observed that the movement directions varied significantly even among neighboring cells (**Figures 4E-4G, S5C-S5F**), supporting ingression rather than involution as the predominant mode of cell movement. We have modified our description in revised manuscript. (Please see Lines 367-374)

8. Lines "332-333 Taken together, these findings suggest that the Bmp4 explant recapitulates not only the structural and molecular characteristics of caudal structures but also the morphogenetic progresses of gastrulation and tail elongation observed in vivo". This conclusion is not supported by the data presented in Figure 4 where altered distances between cells over time are consistent with tissue elongation, but do not illuminate underlying cell behaviors. The conclusions should be significantly toned down or this section removed from the manuscript.

Response: We thank Reviewer #2 for pointing out this issue. We apologize for the earlier imprecise description. To address this concern, we tracked and quantified the movements of additional cells within the tailbud region. Our analysis revealed that the movement patterns of these cells tend to be random (**Figures 4H-4M**, Lines 408-409). As suggested, we have removed these statements in the revised manuscript.

9. The experiment, which BMP4 treated human PSCs were grafted into zebrafish animal poles at sphere stage leading to induction of posterior-like structures is interesting. Please provide more information about how long hPSCs were treated with BMP4 and at what concentration? Importantly, as BMP signaling acts via a positive feedback loop, do these hPSCs express human BMP4 gene? What would suggest that this experiment is simply equivalent to injecting bmp4 RNA?

Response: We thank the reviewer #2 for the question. We used a transgenic H9 hESCs with doxycycline-inducible expression of human *BMP4* (*iBMP4*) CDS for the xenograft experiment. Before transplantation, these DOX inducible hESCs were treated with 1 µg/mL doxycycline for 24 hours. Then about 20-50 *iBMP4* induced cells were transplanted to the animal pole of 4 hpf zebrafish embryos which were then cultured to 24 hpf without DOX induction. To describe the xenograft experiment more clearer, we have modified the schematic illustration in Figure 5A and the description in main text and methods section. (Please see Lines 424-433, 699-700)

As the reviewer indicated, BMP signaling acts via a positive feedback loop, we speculate that there must be some expression of endogenous human *BMP4* in the transplanted human cells. To investigate this, we co-stained the CDS and intron region of BMP4 by HCR on the transplanted human cells in xenografted zebrafish embryos at 5 hpf (**Figure S40**), as the *iBMP4-intron* sequence can only be generated by the endogenous *BMP4* loci. We identified obvious expression of endogenous *BMP4* expression in the transplanted human *iBMP4* cells (**Figure S40**). Based on these experiments, we hypothesize that human *iBMP4* cells may act

as a BMP4 signaling center by establishing a positive feedback loop, and that this signaling mechanism is conserved between zebrafish and humans. Consequently, xenografting human iBMP4 cells into zebrafish embryos could induce the formation of a secondary tail—similar to the effect observed upon injection of *bmp4* RNA.

10. The observation that injection of *bmp4* RNA into animal pole of zebrafish embryos induces *wnt*, *nodal* and *fgf* gene expression is significant and should be discussed. After all, this is the cascade proposed to induce mesendoderm in mouse embryos and human gastruloids. It would be interesting to discuss similarities and differences between the authors experiments and hPSC-based 2D micropatterned gastruloids (Warmflash et al., 2014; Minn et al., 2020) in which germ layers are induced by BMP4.

Response: We thank Reviewer #2 for the insightful comments and constructive suggestions. As noted, human embryonic stem cells cultured under 2D micropatterned conditions and stimulated with BMP4 undergo a gastrulation-like process, characterized by the radially organized emergence of the three germ layers and extraembryonic tissues, as previously described^{2, 8}. Consistent with these observations, we have found that similar spatial patterning can be recapitulated in our 3D embryoid model using iBMP4-expressing cells cultured in mTeSR medium (**Figure S6G**), as demonstrated in a separate, ongoing study. In the present study, we show that iBMP4 induction in E6 medium preferentially drives the formation of caudal mesoderm, rather than anterior mesoderm or endoderm (**Figure S6H**).

These additional data have been incorporated into the revised manuscript to strengthen our conclusions. We also acknowledge that these lineage differences likely result from the presence of TGF- β and FGF pathway components in mTeSR medium (https://cdn.stemcell.com/media/files/pis/10000003789-PIS_03.pdf), underscoring the context-dependent nature of BMP4 signaling in human pluripotent stem cell models.

Minor points:

1. Lines 43-45 "Initially, lineage specification during gastrulation results in the formation of three germ layers, where processes like involution, convergence and extension shape and elongate these layers". Please rephrase, "elongation" is part of "shaping", but what gastrulation also achieves is rearranging the positions of the induced germ layers in the embryo (internalization). Also "involution" is a specific way of internalization observed in amphibian embryos.

Response: We thank the reviewer for reminding us of these misuses of wordings. We have

Full Revision

rephrased the above sentences in our revised manuscript. (Please see Lines 61-62)

2. Gene and RNA names are not consistently written in italics.

Response: We thank the reviewer for pointing out this issue and apologize for our negligence. We have italicized the gene and RNA names accordingly.

3. Figure 1G. What does green fluorescence label?

Response: We thank the reviewer for raising this issue and apologize for our negligence. The green fluorescence in Figure 1G was H2B-GFP which labelled nuclei, and we have added the information in the legends of Figure 1. (Please see Lines 890-891)

4. Figure 5C "notocord" correct to "notochord"

Response: We thank the reviewer for raising this issue and apologize for our omission. We have corrected this word in revised manuscript.

Reviewer #2 (Significance (Required)):

The work is significant as it addresses the fundamental inductive process underlying the formation of the vertebrate body plan during embryogenesis, including activity of the embryonic organizer, what are the minimal signaling requirements for the posterior body organizer. The authors performed interesting and original experiments in zebrafish and hPSC-based embryoids that complement and advance the current knowledge. When revised this work will advance our understanding of the sufficiency of BMP4 signaling for aspects of posterior body development and its evolutionary conservation.

Response: We sincerely thank Reviewer #2 for the positive and insightful feedback, and we greatly appreciate the recognition of our work.

Reviewer #3 (Evidence, reproducibility and clarity (Required)):

Summary:

In this study, the authors demonstrate that BMP4 overexpression is sufficient to induce caudal-like structures in zebrafish embryos, zebrafish explants, and human embryoid bodies.

Full Revision

Transplantation of BMP4-treated human cells into the animal pole of zebrafish embryos also successfully triggered the formation of a secondary axis. These caudal structures closely resembled embryonic caudal regions at the molecular level, aligned with appropriate positions along the anterior-posterior axis, and exhibited similar cell movement dynamics. Based on these findings, the authors propose that BMP alone is sufficient to induce a caudal organizer in otherwise naïve ectoderm. Notably, the ectopic axes lack anterior neural tissue and axial mesoderm, though the reasons for this remain unexplained.

Response: We sincerely appreciate the positive feedbacks from the reviewer. We also thank the reviewer for the professional comments and constructive suggestions. We have carefully revised our manuscript, and we believe our manuscript now shows significant improvements.

Major comments:

While the results are intriguing and provide a model that recapitulates aspects of posterior axis formation, two significant issues remain unresolved.

First, grafting BMP-expressing cells into the early zebrafish blastoderm activates the Nodal, Wnt, and FGF pathways as early as 6 hours post-fertilization (hpf), well before any morphogenetic effects are apparent (the bulge forms at 8 hpf). Since these pathways, along with BMP, are known to contribute to embryo patterning and tissue formation and are capable of inducing ectopic axes, it is unclear which pathway(s) are responsible for axis induction in these experiments. This raises questions about the bold claim that BMP alone is sufficient to establish a posterior organizer.

Response:

We thank the reviewer for pointing out this important issue. To address this question, we treated *bmp4*-injected explants with small-molecule inhibitors targeting the Nodal (SB505124), WNT (IWP-L6), and FGF (SU5402) signaling pathways. We then analyzed explant elongation at 24 hpf and caudal mesoderm specification at 8 hpf. We found that inhibition of Nodal or Wnt signaling prevented Bmp4 from inducing caudal mesoderm specification and blocked explant elongation. In contrast, inhibition of FGF signaling did not significantly affect *bmp4*-induced caudal mesoderm specification but still inhibited explant elongation, possibly due to disruption of the segmentation clock during somitogenesis. (**Figure S2E**).

These findings suggest that while BMP4 is capable of initiating ventral-caudal developmental programs, its effects are mediated through the activation of downstream signaling networks, particularly Nodal and Wnt pathways. We have revised our claim

Full Revision

accordingly to state that BMP4 is sufficient to initiate the posterior developmental process but requires the activation of additional gene regulatory networks to establish posterior organizer activity and elongation dynamics.

Second, the BMP-expressing cells are either not consistently shown in the experiments or appear to scatter quickly within the new posterior structure, as observed in zebrafish embryos and explants (Figs. 1B and 4B) and in hPSC-transplanted fish embryos (Fig. 5C). Few BMP-secreting cells seem to remain in the caudal region of the induced axis. The paper's central claim is therefore not fully supported by the reported observations. Alternatively, it is possible that BMP-secreting cells initiate a signaling cascade early in development, setting up a caudal organizer-like region that subsequently functions without continued BMP activity.

To strengthen the main claim, it is critical to provide compelling evidence of BMP-producing cells contributing to the late-stage caudal structure, akin to the results shown in hPSC embryoid fusions (Fig. 5C). This would clarify whether BMP alone maintains a posterior organizer or merely initiates downstream signaling events that sustain the structure.

Response:

We thank Reviewer #3 for highlighting this important issue and for providing insightful suggestions. We fully agree with the reviewer's perspective that *bmp4* injection or the introduction of BMP-secreting cells initiates a signaling cascade responsible for establishing a ventral-caudal organizer, which subsequently guides the self-organization of the explants into a VCLS. Consequently, clones of *bmp4*-injected blastomeres may contribute to various ventral-caudal tissues.

To address this point, we marked clones derived from *bmp4*-injected blastomeres by co-injecting *bmp4* mRNA and GFP mRNA at the 128-cell stage. We then examined the distribution of these clones using HCR staining at 10 hpf with markers for the tailbud mesoderm progenitor (*tbxta*), pre-somitic mesoderm (*tbx6*), enveloping layer (*krt18*), and lateral plate mesoderm (*aldh1a2*). Our analysis revealed that some *bmp4*-injected clones remained at the injection site and differentiated into enveloping layer cells, while others ingressed and differentiated into lateral plate mesoderm cells (**Figures S5R-S5V**).

We have added a corresponding discussion of these findings in the revised Discussion section of the manuscript (Please see Lines 491-516).

Minor comments:

Full Revision

Can the authors track more cells? 13 cells seem not enough to claim that 'the Bmp4 explant recapitulates not only the structural and molecular characteristics of caudal structures but also the morphogenetic progresses of gastrulation and tail elongation observed in vivo'.

Response: We thank reviewer#3 for pointing out this issue. In our revised manuscript, we have tracked about 100 cells to strength our analysis of tail elongation movements in *bmp4* explant. Additionally, we have removed this statement.

Reviewer #3 (Significance (Required)):

This study is impressive, employing a powerful combination of model systems: zebrafish embryos, embryonic explants, human pluripotent stem cell xenografts, and 3D human stem cell aggregates. The single-cell RNA analyses are particularly insightful, shedding light on the embryonic origin and identity of the induced caudal tissues. When combined with HCR in situ hybridization for spatial characterization, these methods provide a robust framework for understanding the molecular basis of caudal tissue induction.

However, the characterization of tissue and cell behavior remains preliminary, and I strongly encourage the authors to enhance this aspect of the study. Employing quantitative imaging tools to analyze their systems in greater detail would not only substantiate their claim that "the BMP4 explant recapitulates not only the structural and molecular characteristics of caudal structures but also the morphogenetic processes of gastrulation and tail elongation observed in vivo," but also significantly expand the scope of the study. This improvement would attract a broader audience and elevate the impact of this work.

Response: We thank the reviewer for his/her the positive comments and helpful suggestions on our work. As the reviewer suggested, we have carefully revised our manuscript and supplemented additional experiments and analyses. We believe our manuscript now shows significant improvements.

Reference

1. Xu, P.F. et al. Construction of a mammalian embryo model from stem cells organized by a morphogen signalling centre. *Nature communications* **12**, 3277 (2021).
2. Warmflash, A., Sorre, B., Etoc, F., Siggia, E.D. & Brivanlou, A.H. A method to recapitulate early embryonic spatial patterning in human embryonic stem cells. *Nature methods* **11**, 847-854 (2014).

3. Cheng, T. et al. Nodal coordinates the anterior-posterior patterning of germ layers and induces head formation in zebrafish explants. *Cell reports* **42**, 112351 (2023).
4. Xu, P.F., Houssin, N., Ferri-Lagneau, K.F., Thisse, B. & Thisse, C. Construction of a vertebrate embryo from two opposing morphogen gradients. *Science (New York, N.Y.)* **344**, 87-89 (2014).
5. Agathon, A., Thisse, C. & Thisse, B. The molecular nature of the zebrafish tail organizer. *Nature* **424**, 448-452 (2003).
6. Row, R.H., Tsotras, S.R., Goto, H. & Martin, B.L. The zebrafish tailbud contains two independent populations of midline progenitor cells that maintain long-term germ layer plasticity and differentiate in response to local signaling cues. *Development (Cambridge, England)* **143**, 244-254 (2016).
7. Katagiri, T. et al. Identification of a BMP-responsive element in Id1, the gene for inhibition of myogenesis. *Genes Cells* **7**, 949-960 (2002).
8. Deglincerti, A. et al. Self-organization of human embryonic stem cells on micropatterns. *Nature protocols* **11**, 2223-2232 (2016).

Dear Dr. Xu,

Thank you for submitting your revised Review Commons manuscript to The EMBO Journal. We have now received input from all original reviewers, who are satisfied with the revisions and now recommend acceptance of the manuscript after a final minor round of revisions. Their remaining requests can be mainly addressed with textual revisions and improvements of data presentation, while point 2 by reviewer #1 would require inclusion of the appropriate control.

Additionally, there are a few editorial points that need to be addressed before I can extend official acceptance of the manuscript:

1. Please submit a complete author checklist, which you can download from our author guidelines (<https://www.embopress.org/pb-assets/embo-site/EMBO%20Press%20Author%20Checklist-1642513524327.xlsx>). Please insert information in the checklist that is also reflected in the manuscript. The completed author checklist will also be part of the Review Process File.
2. Please remove the significance statement and the list with the additional information from the manuscript.
3. Please re-arrange the manuscript sections and use the following headings: Abstract / Keywords / Introduction / Results / Discussion / Methods / Data Availability / Acknowledgements / Disclosure and Competing Interests Statement / References / Figure Legends / Expanded View Figure Legends.
4. Please remove the figures from the manuscript text and upload the main figures and the supplementary figures as individual, high resolution figure files. The main figure legends should be compiled at the end of the manuscript text, followed by the supplementary figure legends, which should be under the heading "Expanded View Figure Legends". The figure S1 - S6 should be renamed Figure EV1 - Figure EV6.
5. Please merge "Code Availability" with the "Data Availability" section.
6. Please rename Tables S1-4 into Dataset EV1-Dataset EV4. Their legends should be removed from the list of supplementary information in the manuscript text and added to each excel file, in a separate tab/worksheet.
7. Please rename the movies into Movie EV1-EV11 and update the callouts accordingly. The legends should be removed from the manuscript text file and zipped with each movie file. Further information is available here: <https://www.embopress.org/page/journal/14602075/authorguide#expandedview>
8. All Materials and Methods need to be described in the main text using our 'Structured Methods' format. According to this format, the Methods section includes a Reagents and Tools Table (listing key reagents, experimental models, software and relevant equipment and including their sources and relevant identifiers) followed by a Methods and Protocols section describing the methods, ideally using a step-by-step protocol format. The aim is to facilitate adoption of the methodologies across labs. Please download and fill our Reagents and Tools Table template (.docx), which you can find in our author guidelines: <https://www.embopress.org/page/journal/14602075/authorguide#structuredmethods>
When submitting your revised manuscript, please do not include the Reagents and Tools Table in the Methods section of the manuscript but upload it as a separate file choosing the file type "Reagent Table".
An example of a Method paper with Structured Methods can be found here: <https://www.embopress.org/doi/10.15252/msb.20178071>.
9. CRedit has replaced the traditional author contributions section because it offers a systematic, machine-readable author contributions format that allows for more effective research assessment. Please remove the Authors Contributions from the manuscript and use the free text boxes beneath each contributing author's name in our online submission system to add specific details on the author's contribution. More information is available in our guide to authors.
10. Please rename "Declaration of interests" section into "Disclosure and competing interests statement" (further info: <https://www.embopress.org/page/journal/14602075/authorguide#conflictsofinterest>).
11. Please update references according to The EMBO Journal style - it should be alphabetical. Please also remove the DOIs. Please see further information here: <https://www.embopress.org/page/journal/14602075/authorguide#referencesformat>
12. Figure panel 4D is currently not mentioned in the manuscript text - please add the corresponding callout.
13. Please check the order of the figure callouts: Fig. S2 (EV2 in the updated nomenclature) should be called out before Fig. S1 (EV1)
14. In our standard image check, we have found that microscopy and blot panels across the entire figure set appear pixelated under image analysis. This is a common result of converting original 16-bit TIFF images to RGB format for publication. While not a direct cause for integrity concern, it can give the impression of image alteration to critical readers. To avoid any misunderstanding and to ensure compliance with EMBO Press standards, please resubmit the complete figure set at its original data resolution, preferably in TIFF format, to preserve image quality.
15. At EMBO Press we ask authors to provide source data for the main manuscript figures. You will receive a separate email with instructions for providing source data with your revised manuscript, including how to upload and organize the files.
16. Our data editors have flagged the following issues in figure legends that need correcting:
 - Please define the annotated p values ****/**/*/* as well as provide the exact p-values for the same in the legend of figure 4G as appropriate.
 - Please indicate the statistical test used for data analysis in the legend of figure 4G.
 - Please provide information on the number and nature of replicates in the legend of figure 6B.
 - Please define the error bars in the legends of figures 1C, 6B.

17. Papers published in The EMBO Journal are accompanied online by a 'Synopsis' to enhance discoverability of the manuscript. It consists of A) a short (1-2 sentences) summary of the findings and their significance, B) 3-4 bullet points highlighting key results and C) a synopsis image that is 550x300-600 pixels large (width x height, jpeg or png format). You can either show a model or key data in the synopsis image. Please note that the image size is rather small and that text needs to be readable at the final size.

With kind regards,

Ieva

Revision to The EMBO Journal should be submitted online within 90 days, unless an extension has been requested and approved by the editor; please click on the link below to submit the revision online before 2nd Dec 2025:

Link Not Available

Referee #1:

The authors have addressed many of my concerns raised in the initial review. However, some remaining issues should be addressed before publication:

1. In line 191, The authors state that "Inhibition of Nodal or Wnt signaling prevented bmp4 from inducing ... caudal mesoderm", but figure S2E shows clearly that WNTi explants still express caudal mesoderm markers *tbxta* and *cdx4*.
2. Figure 5 shows secondary axis formation upon xenograft of human PSCs into zebrafish embryos, but panel D indicates this experiment was only performed in 5 embryos (during 3 independent replicates). This means that for at least one replicate, only a single embryo was analyzed? Also, no controls are shown for this experiment. Transplantation of BMP4-expressing cells should be compared with uninduced hPSCs to ensure the response is specific to BMP4.
3. Fig. S4N is supposed to show posterior fate acquisition near human PSC xenografts into zebrafish embryos, but this is very difficult to discern from the image provided. Without a bright field image of the embryo, we cannot know what region of the embryo received the transplant and a zoomed-in image would enable visualization of transplanted cells
4. In line 452, the authors state that gastruloids do not form the full spectrum of cell fates induced by BMP4 in E6 media, then go on to say the explants elongate and form segments. Do they mean the explants on E6 media elongate despite the absence of many cell types? Or that mTESR gastruloids elongate? Please clarify the writing in this section.

Minor issues and typographical errors:

1. Figures 1F, S2C, S4L-N, 5C, S5, 6C, S6G: Avoid red-green fluorescence images, which are not accessible to colorblind readers.
2. Line 164: "a neural domain that expressing sox2 was found..."

3. Line 196: "bmp4 could induced the endoderm specification..."
4. Line 230: "none-dorsal margin"
5. Line 255: "-which normally typically the notochord, hypochord..."
6. Line 278: "One NMP cell rarely give rise to..."
7. The word "involution" is still used in Fig. 4, although it was replaced with "ingression / internalization" throughout the manuscript.

Referee #2:

In this revised and significantly improved manuscript Xing et al., report a series of interesting experiments in zebrafish, hESCs and hESC/zebrafish chimeras in support of the thesis that BMP4 alone is sufficient to induce a ventral-caudal vertebrate embryonic organizer. They propose that this organizer replicates some morphogenesis and key differentiation pathways observed during normal vertebrate development and that this role of BMP is conserved from zebrafish to human. The authors made a significant effort to address many of the reviewers' questions and concerns raised about the original submission. The key observations include, first, injection of RNA encoding BMP4 in the animal blastomeres of the zebrafish embryo induces ectopic tailbud-like structure and later formation of posterior structures resembling tail and possessing neuromesodermal progenitors, somites, blood, pronephros, and neural tissues but critically lacking notochord and endoderm. Further, Bmp4 activates key signaling pathways pivotal for early development, including Nodal, Wnt and FGF. Another interesting observation is that explants of the animal poles of embryos injected with moderate doses of BMP RNA underwent elongation and formed somites, had neural tissues and exhibited anteroposterior polarity. Further, BMP4 treated hESCs when transplanted into the animal pole of zebrafish embryo induce secondary tail-like structures. Finally, an aggregation of BMP4-inducible PSCs with an aggregation of normal human PSCs led to formation of embryoids that exhibited anteroposterior polarity and expressed genes equivalent to those observed in ectopic tail-like structures induced by BMP4 in zebrafish. The authors performed thorough analysis of cell fates with single-cell RNA-seq, fluorescent in situ hybridization and antibody stainings that showed induced cell types and their spatial arrangement over time. The revised manuscript is now placing these findings in the context of the previous work, including their own, more clearly stating that they provide evidence for BMP4 activity underlying the ventral gastrula organizer residing in the ventral blastoderm margin of zebrafish gastrula as elegantly shown by the Agathon and Thisses. Demonstration of the conserved activity of BMP4 in hESC-based gastruloids in context-dependent manner strengthens the manuscript. The work is significant as it addresses the fundamental inductive process underlying the formation of the vertebrate body plan during embryogenesis, including activity of the embryonic organizers, what are the minimal signaling requirements for the posterior body organizer.

This is interesting work that should be of interest to the fast-advancing field of vertebrate and human embryo models. This reviewer supports publication when the following few outstanding points are addressed.

Lines 51-53 "we show that this organizer replicates morphogenesis and key differentiation pathways seen in natural development." Should be revised to "replicates aspects of morphogenesis"

Lines 66-68 "In zebrafish, tailbud formation occurs as the ventral and dorsal part of blastoderm margin slip over the yolk and fuses at the end of gastrulation". Needs revising "In zebrafish, tailbud formation occurs as the ventral and dorsal part of blastoderm margin slip over the yolk and fuses at the vegetal pole at the end of gastrulation."

Lines 68-71 Please consider fusing these two sentences "Although... Our understanding..." into one sentence.

Lines 140-141 "Additionally, a continuously dorsalized neural tube extending to the tail tip is observed, marked by the expression of sox19a and olig2 (Figure S2J)." What do the authors mean by "continually dorsalized neural tube". There is no tube observed in the images. Moreover, olig2 is a ventral neural marker. Please revise.

Line 218 "epidermal cells identities" revised "epidermal cell identities"

The authors continue to use the term "involution" for the internalization of mesodermal cells. In zebrafish mesoderm internalizes via ingression (Figure. 4A). Evidence should be presented that the explants, in contrast to embryos undergo "involution".

Line 362-363 "During the gastrulation stage, cells near the injection site converged toward the anterior of the explant ". There is no evidence of convergence movements presented, as convergence entails movements of broadly distributed cells (or tissue) to a more narrow array (tissue). The cell tracking shown in Figure 4B indicates rather random dispersal of cells and translocation away from the injection site. However, these movements do not mimic morphogenetic convergence movements observed in vivo. These movements are reminiscent of mesodermal cells moving away from the blastoderm margin (blastopore) upon internalization, behavior conserved in vertebrate gastrulae.

There are still concerns about the experiments reported as the key evidence that iBMP4 human PSCs function as an organizer instructing VCLS formation. This reviewer does not see appropriate controls. In Figure 5d, images are show of zebrafish embryos with ectopic tail-like structures upon transplantation of H9 hESCs induced to express BMP4. Controls should include H9 cell xenografts and also iBMP4-hESCs not treated with doxycyclin.

Line 492-493 "bmp4 ...induces the formation of a secondary axis with caudal-like structure," consider "a partial/incomplete secondary axis"

Referee #3:

I have read the revised manuscript and all the authors' responses to the reviewers' comments. I believe the authors have addressed the majority of concerns in a satisfactory manner. I appreciate the improvements made in the analysis of cell movements, and I support publication.

Rev_Com_number: RC-2024-02776

New_manu_number: EMBOJ-2025-121857-T

Corr_author: Xu

Title: BMP4 initiates and patterns ventral-caudal structures in zebrafish and human pluripotent stem cells

Answers to Referee Comments

We sincerely thank the editor and reviewers for their insightful comments and constructive suggestions. We have carefully considered all feedback, revised the manuscript thoroughly, and conducted additional control experiments in response to the reviewers' recommendations. And, we also have addressed the editorial points raised by editor. We believe that these revisions have substantially improved the rigor, clarity, and overall significance of the study. Below, we provide a detailed point-by-point response to each comment. For ease of review, all changes in the revised manuscript are highlighted in yellow.

Referee #1:

The authors have addressed many of my concerns raised in the initial review. However, some remaining issues should be addressed before publication:

Response: We thank this reviewer for the recognition of our work and valuable comments.

1. In line 191, The authors state that "Inhibition of Nodal or Wnt signaling prevented bmp4 from inducing ... caudal mesoderm", but figure S2E shows clearly that WNTi explants still express caudal mesoderm markers *tbxta* and *cdx4*.

Response: We thank the reviewer for raising this issue and apologized for the inaccurate description. We have modified our statement (Please see revised manuscript Lines 207-209).

2. Figure 5 shows secondary axis formation upon xenograft of human PSCs into zebrafish embryos, but panel D indicates this experiment was only performed in 5 embryos (during 3 independent replicates). This means that for at least one replicate, only a single embryo was analyzed?

Response: We thank the reviewer for raising this issue. As the reviewer noted, we obtained only 1–2 embryos with well-structured tail tissue from each xenograft of

human PSCs into zebrafish embryos, due to the low efficiency. To increase the sample size, we repeated the xenograft experiments and successfully obtained more samples (Please see Figure R1 below). We have modified the number of xenograft zebrafish embryos accordingly (Please see revised **Figure 5D**).

Figure R1. New supplemented xenograft zebrafish embryos stained with *tbxta*

Also, no controls are shown for this experiment. Transplantation of BMP4-expressing cells should be compared with uninduced hPSCs to ensure the response is specific to BMP4.

Response: We thank the reviewer for raising this important question. To address this issue, as the reviewer suggested, we transplanted human WT H9 cells and iBMP4 cells without DOX treatment into zebrafish embryos at 4 hpf and imaged the embryos at 24 hpf. We found that neither population of human iPSCs induced ectopic structures in zebrafish; instead, the cells were randomly distributed throughout the embryos (Please see revised **Figure EV4P**).

3. Fig. S4N is supposed to show posterior fate acquisition near human PSC xenografts into zebrafish embryos, but this is very difficult to discern from the image provided. Without a bright field image of the embryo, we cannot know what region of the embryo received the transplant and a zoomed-in image would enable visualization of transplanted cells

Response: We thank the reviewer for raising this issue. As suggested by the reviewer, we repeated this experiment and included a bright-field image along with a zoomed-in view to better visualize the transplanted cells. As shown in the images, the human transplanted cells were located in the animal pole of zebrafish embryos (Please see

revised **Figure EV4N**).

4. In line 452, the authors state that gastruloids do not form the full spectrum of cell fates induced by BMP4 in E6 media, then go on to say the explants elongate and form segments. Do they mean the explants on E6 media elongate despite the absence of many cell types? Or that mTESR gastruloids elongate? Please clarify the writing in this section.

Response: We thank the reviewer for raising this issue and apologized for the unclear description. As the reviewer suggested, we have modified our statement in the main text (Please see revised manuscript **Lines 472-476**).

Minor issues and typographical errors:

1. Figures 1F, S2C, S4L-N, 5C, S5, 6C, S6G: Avoid red-green fluorescence images, which are not accessible to colorblind readers.

Response: We thank the reviewer for pointing out this issue. As the reviewer suggested, we have modified our figures accordingly (Please see revised **Figures 1F, EV2C, EV4L-N, 5C, EV5, 6C, EV6G**).

2. Line 164: "a neural domain that expressing sox2 was found..."

Response: We thank the reviewer for pointing out this issue. We have corrected in the revised manuscript (Please see **Line 178**).

3. Line 196: "bmp4 could induced the endoderm specification..."

Response: We thank the reviewer for pointing out this issue. We have corrected this in the revised manuscript (Please see **Line 212**).

4. Line 230: "none-dorsal margin"

Response: We thank the reviewer for pointing out this issue. We have corrected in the revised manuscript (Please see **Line 247**).

5. Line 255: "-which normally typically the notochord, hypochord..."

Response: We thank the reviewer for pointing out this issue. We have corrected in the revised manuscript (Please see **Line 272**).

6. Line 278: "One NMP cell rarely give rise to..."

Response: We thank the reviewer for pointing out this issue. We have corrected in the revised manuscript (Please see **Line 296**).

7. The word "involution" is still used in Fig. 4, although it was replaced with "ingression / internalization" throughout the manuscript.

Response: We thank the reviewer for pointing out this issue. We have corrected in the revised manuscript (Please see revised **Figure 4**).

Referee #2:

In this revised and significantly improved manuscript Xing et al., report a series of interesting experiments in zebrafish, hESCs and hESC/zebrafish chimeras in support of the thesis that BMP4 alone is sufficient to induce a ventral-caudal vertebrate embryonic organizer. They propose that this organizer replicates some morphogenesis and key differentiation pathways observed during normal vertebrate development and that this role of BMP is conserved from zebrafish to human. The authors made a significant effort to address many of the reviewers' questions and concerns raised about the original submission.

The key observations include, first, injection of RNA encoding BMP4 in the animal blastomeres of the zebrafish embryo induces ectopic tailbud-like structure and later formation of posterior structures resembling tail and possessing neuromesodermal progenitors, somites, blood, pronephros, and neural tissues but critically lacking notochord and endoderm. Further, Bmp4 activates key signaling pathways pivotal for

early development, including Nodal, Wnt and FGF. Another interesting observation is that explants of the animal poles of embryos injected with moderate doses of BMP RNA underwent elongation and formed somites, had neural tissues and exhibited anteroposterior polarity. Further, BMP4 treated hESCs when transplanted into the animal pole of zebrafish embryo induce secondary tail-like structures. Finally, an aggregation of BMP4-inducible PSCs with an aggregation of normal human PSCs led to formation of embryoids that exhibited anteroposterior polarity and expressed genes equivalent to those observed in ectopic tail-like structures induced by BMP4 in zebrafish. The authors performed thorough analysis of cell fates with single-cell RNA-seq, fluorescent in situ hybridization and antibody stainings that showed induced cell types and their spatial arrangement over time. The revised manuscript is now placing these findings in the context of the previous work, including their own, more clearly stating that they provide evidence for BMP4 activity underlying the ventral gastrula organizer residing in the ventral blastoderm margin of zebrafish gastrula as elegantly shown by the Agathon and Thisses. Demonstration of the conserved activity of BMP4 in hESC-based gastruloids in context-dependent manner strengthens the manuscript. The work is significant as it addresses the fundamental inductive process underlying the formation of the vertebrate body plan during embryogenesis, including activity of the embryonic organizers, what are the minimal signaling requirements for the posterior body organizer.

This is interesting work that should be of interest to the fast-advancing field of vertebrate and human embryo models. This reviewer supports publication when the following few outstanding points are addressed.

Response: We thank the reviewer for the positive feedback. We truly appreciate the reviewer's professional comments and constructive suggestions on our manuscript. Following the suggestions of the reviewer, we carefully revised our manuscript.

Lines 51-53 "we show that this organizer replicates morphogenesis and key differentiation pathways seen in natural development." Should be revised to "replicates aspects of morphogenesis"

Response: We thank the reviewer for raising this issue. As the editor suggested, we have removed the significance statement section in our revised manuscript.

Lines 66-68 "In zebrafish, tailbud formation occurs as the ventral and dorsal part of

blastoderm margin slip over the yolk and fuses at the end of gastrulation". Needs revising "In zebrafish, tailbud formation occurs as the ventral and dorsal part of blastoderm margin slip over the yolk and fuses at the vegetal pole at the end of gastrulation."

Response: We thank the reviewer for this suggestion. We have modified this sentence accordingly (Please see revised manuscript Lines 67-69).

Lines 68-71 Please consider fusing these two sentences "Although... Our understanding...." into one sentence.

Response: We thank the reviewer for the suggestion; we have revised the sentence as suggested (Please see revised manuscript Lines 71-76).

Lines 140-141 "Additionally, a continuously dorsalized neural tube extending to the tail tip is observed, marked by the expression of sox19a and olig2 (Figure S2J)." What do the authors mean by "continually dorsalized neural tube". There is no tube observed in the images. Moreover, olig2 is a ventral neural marker. Please revise.

Response: We thank the reviewer for pointing out this important issue. We have revised our description as suggested (Please see revised manuscript Lines 153-155).

Line 218 "epidermal cells identities" revised "epidermal cell identities"

Response: Thank the reviewer for the suggestion, we have corrected this as suggested (Please see revised manuscript Line 234).

The authors continue to use the term "involution" for the internalization of mesodermal cells. In zebrafish mesoderm internalizes via ingression (Figure. 4A). Evidence should be presented that the explants, in contrast to embryos undergo "involution".

Response: We thank the reviewer for raising this important issue. As the reviewer suggested, we have modified our Figure accordingly (Please see revised **Figure 4A**).

Line 362-363 "During the gastrulation stage, cells near the injection site converged toward the anterior of the explant ". There is no evidence of convergence movements presented, as convergence entails movements of broadly distributed cells (or tissue)

to a more narrow array (tissue). The cell tracking shown in Figure 4B indicates rather random dispersal of cells and translocation away from the injection site. However, these movements do not mimic morphogenetic convergence movements observed in vivo. These movements are reminiscent of mesodermal cells moving away from the blastoderm margin (blastopore) upon internalization, behavior conserved in vertebrate gastrulae.

Response: We thank the reviewer for pointing out this issue. As suggested, we have modified our description accordingly (Please see revised manuscript Lines 380-382).

There are still concerns about the experiments reported as the key evidence that iBMP4 human PSCs function as an organizer instructing VCLS formation. This reviewer does not see appropriate controls. In Figure 5d, images are show of zebrafish embryos with ectopic tail-like structures upon transplantation of H9 hESCs induced to express BMP4. Controls should include H9 cell xenografts and also iBMP4-hESCs not treated with doxycyclin.

Response: We thank the reviewer for pointing out this important issue. As the reviewer suggested, we transplanted human WT H9 cells and iBMP4 cells without DOX treatment into zebrafish embryos at 4 hpf and imaged the embryos at 24 hpf. We found that neither population of human iPSCs induced ectopic structures in zebrafish; instead, the cells were randomly distributed throughout the embryos (Please see revised **Figure EV4P**).

Line 492-493 "bmp4 ...induces the formation of a secondary axis with caudal-like structure," consider "a partial/incomplete secondary axis"

Response: We thank the reviewer for the suggestion; we have revised the sentence accordingly (Please see revised manuscript Line 516).

Referee #3:

I have read the revised manuscript and all the authors' responses to the reviewers' comments. I believe the authors have addressed the majority of concerns in a satisfactory manner. I appreciate the improvements made in the analysis of cell movements, and I support publication.

Response: We sincerely thank the reviewer for recognizing our efforts in the revision

and for expressing satisfaction with our responses. We greatly appreciate the reviewer's constructive and professional suggestions, which helped us identify weaknesses and substantially improve the manuscript.

Dear Pengfei,

Thank you for your input on the final editorial requests. I am now pleased to inform you that your manuscript has been accepted for publication in The EMBO Journal. Congratulations with a nice study!

Before we forward your manuscript to our publishers, we would like to propose some edits in the manuscript title, abstract and synopsis (please see below and in the attached file). I have also written a short blurb that will accompany the title of your manuscript in our online system. Please take a look and let me know if any corrections are needed.

Title:

BMP4 initiates and patterns ventral-caudal structures in zebrafish and human pluripotent stem cell aggregates

Blurb:

BMP4 induces formation of an organizer that recapitulates key aspects of gastrulation and somitogenesis.

Synopsis:

The vertebrate body's posterior part forms through a complex interplay of signals. This study reveals that BMP4 alone is sufficient to create a "ventral-caudal organizer" that directs the development of posterior structures, pointing to a minimal, evolutionarily conserved signaling for building the posterior body plan across vertebrates.

- Ectopic BMP4 in the zebrafish animal pole induces a secondary axis containing somites and a tailbud but lacking notochord and endoderm.
- These BMP4-induced structures recapitulate key aspects of gastrulation, somitogenesis, and anterior-posterior patterning of ventral-caudal tissues.
- Human pluripotent stem cells expressing BMP4 can organize a secondary caudal axis when transplanted into zebrafish embryos, demonstrating cross-species organizer activity.
- The BMP4 signaling center induces a ventral-caudal-like structure in human 3D pluripotent stem cell aggregates in vitro.

Please note that it is The EMBO Journal policy for the transcript of the editorial process (containing referee reports and your response letters) to be published as an online supplement to each paper. If you should prefer removal of any referee-only figures included in the point-by-point response(s), e.g. because they may still be used for future publication or because they have been reproduced from published work by others, please do let us know immediately via response email.

More information is available here: https://www.embopress.org/transparent-process#Review_Process

If you have any questions, please do not hesitate to contact the Editorial Office or me directly. Thank you for your contribution to The EMBO Journal, and I look forward to your input on the textual changes!

With best wishes,

Ieva

Rev_Com_number: RC-2024-02776
New_manu_number: EMBOJ-2025-121857R
Corr_author: Xu
Title: BMP4 initiates and patterns ventral-caudal structures in zebrafish and human pluripotent stem cells